# Sharper Generalization Guarantees for Asynchronous SGD: Beyond Lipschitzness, Smoothness and Data Homogeneity

**Yufeng Xie** [1]   **Yunwen Lei** [1]

## Abstract

Asynchronous stochastic gradient descent (ASGD) is widely adopted in distributed and federated learning. In this paper, we develop a sharp generalization analysis for ASGD by leveraging the concept of on-average model stability. For convex and smooth objectives, we establish stability and excess risk bounds under minimal assumptions, removing assumptions on Lipschitz continuity, bounded noise, bounded parameter or data domains, while allowing randomly partitioned data and arbitrary delays. Our bounds are optimistic and explicitly characterize the impact of worker participation, recovering the minimax-optimal rate $O(1/\sqrt{mn})$ in balanced regimes where $mn$ denotes the sample size and implying fast rates under low-noise conditions. We further extend the analysis to non-smooth objectives with Hölder-continuous gradients and to heterogeneous data settings via *random* ASGD, obtaining non-vacuous excess risk guarantees in both settings. Experimental results support our theoretical findings.

## 1. Introduction

Stochastic gradient descent (SGD) and its parallel variants have been widely employed for large-scale machine learning tasks, exploiting the decomposable structure of stochastic gradients to improve computational efficiency (Zinkevich et al., 2010; Lian et al., 2015). In distributed systems, gradient computation jobs are assigned to multiple parallelized workers and aggregated by a central server for model updates. Synchronous methods, e.g. mini-batch SGD (Cotter et al., 2011) and local SGD (Stich, 2018), require all workers to synchronize at each iteration, making the training

process vulnerable to stragglers and communication bottlenecks (Mishchenko et al., 2022). Asynchronous SGD (ASGD) alleviates these limitations by allowing workers to compute and communicate independently (Recht et al., 2011; Agarwal & Duchi, 2011). This asynchrony improves system throughput but introduces delayed gradients, since model updates may occur while a worker is still computing its local gradient. Understanding how such delays affect learning performance is therefore a fundamental problem in distributed optimization and federated learning (Dean et al., 2012; Nguyen et al., 2022).

Existing theoretical analyses mainly focus on the convergence performance of ASGD, considering both homogeneous and heterogeneous data settings where each worker can access the whole training data or only a subset of it, with a different distribution (Mishchenko et al., 2022; Koloskova et al., 2022; Islamov et al., 2024). In learning theory, however, convergence alone is insufficient given that the generalization performance on unseen data is equally critical. Despite its practical importance, the generalization behavior of ASGD remains poorly understood, particularly beyond the homogeneous data regime. Early work by Regatti et al. (2019) initiated the stability-based generalization analysis of ASGD via the lens of uniform stability (Hardt et al., 2016), revealing the slight impact of delays on generalization. Following studies (Deng et al., 2024; 2025) leveraged weaker stability measures and derived tighter risk bounds, while Liu et al. (2022) analyzed ASGD with variance-reduction under different memory architectures.

Nevertheless, existing generalization results for ASGD remain limited in three main aspects: restrictive assumptions, simplified settings and pessimistic, delay-sensitive risk bounds. For instance, Deng et al. (2025) relied on the quadratic assumption for the loss function while Deng et al. (2024) assumed that the parameter space is bounded and the loss function is Lipschitz. Moreover, most existing studies (Deng et al., 2025; 2024; Liu et al., 2022) are limited to homogeneous data settings in which all workers have access to the entire dataset. Further, these results fail to attain the minimax-optimal rate of excess risk for convex and smooth problem (Chen et al., 2018). In this work, we provide a fine-grained stability and generalization analysis of ASGD

[1]Department of Mathematics, The University of Hong Kong, HongKong, China. Correspondence to: Yunwen Lei <leiyw@hku.hk>.

*Proceedings of the 43$^{rd}$ International Conference on Machine Learning*, Seoul, South Korea. PMLR 306, 2026. Copyright 2026 by the author(s).

under relaxed assumptions and realistic distributed settings. Our goal is to derive sharp excess risk bounds, revealing the impacts of asynchrony and heterogeneity on generalization performance. The main contributions of this paper are summarized as follows.

• We analyze the on-average model stability and excess risk of ASGD for convex and smooth problems, removing restrictive conditions, such as Lipschitz or quadratic losses. We measure the degree of asynchrony by worker participation, demonstrating that such asynchrony has a provably limited impact on both stability and generalization.

• We derive optimistic and delay-insensitive stability and excess risk bounds. In balanced regimes, we achieve the minimax-optimal excess risk rate $O((mn)^{-\frac{1}{2}})$, where $mn$ is the total sample size. Additionally, our results imply faster rates under low-noise conditions.

• We extend the analysis to non-smooth objectives by replacing smoothness with Hölder continuity of gradients, maintaining optimistic bounds via appropriate step sizes. Focusing on heterogeneous data, we establish the first non-vacuous excess risk guarantees for *random* ASGD without requiring additional assumptions on data heterogeneity, thereby indicating that such assumptions are inessential for generalization analysis of ASGD.

• We conduct experiments for smooth, non-smooth and non-convex problems to validate our theoretical findings.

## 2. Related Work

### 2.1. Asynchronous SGD

The exploration on ASGD can be traced back to the work of Bertsekas & Tsitsiklis (1989). Existing analyses focusing on convergence properties of ASGD have concentrated on identifying the negative impact of gradient delays (Stich & Karimireddy, 2020), and on developing delay-adaptive schemes to mitigate it, such as adaptive step sizes (Koloskova et al., 2022), adaptive query-point for online learning (Aviv et al., 2021), and Picky SGD (Cohen et al., 2021). While Stich & Karimireddy (2020); Arjevani et al. (2020) provided convergence guarantees for ASGD under constant delays, subsequent works relaxed this assumption, establishing bounds that depend on the maximum delay (Stich et al., 2021; Nguyen et al., 2022). Recently, by introducing a virtual iterate technique, Koloskova et al. (2022) obtained convergence rates relying only on the average delays for smooth losses, whereas Mishchenko et al. (2022) derived optimization error bounds in terms of the worker count under arbitrary delays for both convex and non-convex problems. The convergence analysis of ASGD was recently extended to heterogeneous data settings by Islamov et al. (2024); Koloskova et al. (2022); Mishchenko et al.

(2022). In this setting, local data is sampled from a different distribution, which is prevalent in federated learning (Xu et al., 2023; Fraboni et al., 2023).

### 2.2. Stability and Generalization

Algorithmic stability, originally introduced to quantify an algorithm's sensitivity to data perturbations (Rogers & Wagner, 1978), has become a vital tool for generalization analysis (Bousquet & Elisseeff, 2002; Elisseeff et al., 2005). Unlike the uniform convergence analysis (Bartlett & Mendelson, 2002; Vapnik, 2013; Hieu & Ledent, 2025; Ledent et al., 2026), a notable property of stability analysis is that it can imply algorithm-dependent generalization bounds. A seminal work by Hardt et al. (2016) analyzed the generalization of SGD via uniform stability under smoothness and Lipschitzness assumptions. To relax these requirements, Lei & Ying (2020) proposed a weaker stability measure named on-average model stability, which removes the Lipschitz condition and weakens smoothness while retaining optimal excess risk bounds. Several related stability measures, including uniform argument stability (Bassily et al., 2020), stability in gradients (Fan & Lei, 2024; Chen et al., 2026; Zhu et al., 2026), and on-average stability (Shalev-Shwartz et al., 2010), have also been applied to the generalization analysis of stochastic optimization algorithms (Neu et al., 2021; Nikolakakis et al., 2022; Liu et al., 2024; Schliserman & Koren, 2022; Zeng & Lei, 2026; Lei et al., 2025; Zeng & Lei, 2025; Schliserman et al., 2025; Lei et al., 2026) and neural networks (Richards & Kuzborskij, 2021; Taheri & Thrampoulidis, 2024; Ma et al., 2026; Deora et al., 2024).

Several works have studied the stability and generalization of asynchronous SGD. Regatti et al. (2019) derived a uniform stability bound of order $O(T^{\tau_M}/(n\tau_M))$, where $T$, $n$, and $\tau_M$ denote the number of iterations, sample size, and maximum delay, respectively. Liu et al. (2022) proposed a semi-asynchronous variant and obtained a bound of $O(\eta T/n)$ under restrictive assumptions. In the setting of convex quadratic losses and homogeneous data, Deng et al. (2025) established a generalization error bound of $\tilde{O}((T-\tau)/(n\tau))$ for constant delays, with a faster rate of $\tilde{O}(1/n)$ in the strongly convex case. More recently, Deng et al. (2024) employed on-average model stability to derive generalization error and excess risk bounds of $O(1/\bar{\tau} + 1/\sqrt{n})$ and $O(1/\bar{\tau} + 1/n)$, respectively, for convex and smooth problems, where $\bar{\tau}$ denotes the average delay.

## 3. Preliminaries

Let $\mathcal{D}$ denote an unknown distribution defined over the sample space $\mathcal{Z} = \mathcal{X} \times \mathcal{Y}$, where $\mathcal{X}$ and $\mathcal{Y} \subseteq \mathbb{R}$ denote the input and output space, respectively. We consider the

classical supervised learning problem

$$\min_{\mathbf{w} \in \mathcal{W}} F(\mathbf{w}) := \mathbb{E}_{\mathbf{z} \sim \mathcal{D}}[f(\mathbf{w}; \mathbf{z})], \quad (3.1)$$

where $F(\mathbf{w})$ is the population risk, $\mathbf{w} \in \mathcal{W} \subseteq \mathbb{R}^d$ denotes the model parameters, $\mathbb{E}_{\mathbf{z} \sim \mathcal{D}}[\cdot]$ represents the expectation with respect to (w.r.t.) $\mathbf{z}$, and $f(\mathbf{w}; \mathbf{z}) : \mathcal{W} \times \mathcal{Z} \mapsto \mathbb{R}^+$ indicates a non-negative loss function. The population risk minimizer is defined as

$$\mathbf{w}^* = \operatorname{argmin}_{\mathbf{w} \in \mathcal{W}} F(\mathbf{w}). \quad (3.2)$$

Since the distribution $\mathcal{D}$ is unknown, learning is performed by minimizing the empirical risk over training dataset. We consider a distributed setting with $m$ workers and a central server maintaining the model parameters. Each worker $j \in [m] := \{1, \ldots, m\}$ holds a local dataset $S_j = \{\mathbf{z}_{1,j}, \ldots, \mathbf{z}_{n,j}\}$, where $n$ denotes the local sample size and $\mathbf{z}_{i,j} = (\mathbf{x}_{i,j}, y_{i,j})$ are i.i.d. samples drawn from $\mathcal{D}$. Let $S = \{S_1, \ldots, S_m\}$ denote the collection of all local datasets. The empirical risk is defined by

$$F_S(\mathbf{w}) = \frac{1}{mn} \sum_{i=1}^{n} \sum_{j=1}^{m} f(\mathbf{w}; \mathbf{z}_{i,j}). \quad (3.3)$$

Let $A$ denote a randomized learning algorithm, and $A(S)$ denote its output when performed on dataset $S$. We are mainly concerned with the excess risk $\mathbb{E}[F(A(S)) - F(\mathbf{w}^*)]$, which admits the standard decomposition

$$\mathbb{E}\big[F(A(S)) - F(\mathbf{w}^*)\big] =$$
$$\underbrace{\mathbb{E}\big[F(A(S)) - F_S(A(S))\big]}_{\text{generalization error}} + \underbrace{\mathbb{E}\big[F_S(A(S)) - F(\mathbf{w}^*)\big]}_{\text{optimization error}}.$$
$$(3.4)$$

### 3.1. Asynchronous SGD

In this paper, we focus on the Asynchronous SGD (Algorithm 1), with references to prior works (Mishchenko et al., 2022; Koloskova et al., 2022). When performing ASGD, the central server first initializes the model parameters $\mathbf{w}_1$ and subsequently broadcasts it to all $m$ workers. Upon receiving the weights, each worker independently and uniformly draws a sample from its local dataset and computes stochastic gradient. When a worker, indexed as $m_t$, completes its gradient computation at iteration $t$, it immediately sends the gradient $\nabla f(\mathbf{w}_{t-\tau_t}; \mathbf{z}_{i_t, m_t})$ to the central server for model updating, where $\tau_t$ denotes the delay induced by asynchronous updates. The server then performs the update $\mathbf{w}_{t+1} = \mathbf{w}_t - \eta_t \nabla f(\mathbf{w}_{t-\tau_t}; \mathbf{z}_{i_t, m_t})$ and returns the updated parameters to the worker. We denote the worker arrival sequence by $M = \{m_t\}$, which completely determines the delay sequence $\{\tau_t\}$. In general, no structural assumptions are imposed on the delays, which may be deterministic or random. The randomness in Algorithm 1 arises solely from the random sampling of data indices $I = \{i_1, \ldots, i_T\}$.

---

**Algorithm 1** Asynchronous SGD

---

**Input:** Initial weights $\mathbf{w}_1 \in \mathbb{R}^d$, iteration number $T$, learning rate scheme $\{\eta_t\}_{t=1}^T$
**Initialization:** The server assigns $\mathbf{w}_1$ to all workers for gradient computation
**Server**
    **for** $t = 1, \ldots, T$ **do**
        Receive gradient from worker $m_t$: $\nabla f(\mathbf{w}_{t-\tau_t}; \mathbf{z}_{i_t, m_t})$,
        where $i_t$ is sampled from $\{1, \ldots n\}$ uniformly
        Update the model: $\mathbf{w}_{t+1} = \mathbf{w}_t - \eta_t \nabla f(\mathbf{w}_{t-\tau_t}; \mathbf{z}_{i_t, m_t})$
        Assign $\mathbf{w}_{t+1}$ to worker $m_t$ for gradient computation
    **end for**

---

### 3.2. Generalization via Model Stability

On-average model stability, measuring the sensitivity of algorithm's output, $A(S)$, to a slight perturbation of training dataset, $S$, exhibits a close correlation to algorithm's generalization performance (Lei & Ying, 2020). In distributed settings, perturbation occurs within individual sub-dataset, motivating a refined stability notion tailored to local data.

**Definition 3.1** (On-average model stability). Let $S = \{S_1, \ldots, S_m\}$ with $|S_j| = n$ and let $\tilde{S}$ be an independent copy of $S$. Specifically, $S_j = \{\mathbf{z}_{1,j}, \ldots, \mathbf{z}_{n,j}\}$, where $\mathbf{z}_{i,j} \sim \mathcal{D}$. For any $i \in [n]$ and $j \in [m]$, let $S^{(ij)}$ denote the dataset formed from $S$ by replacing $\mathbf{z}_{i,j}$ by $\tilde{\mathbf{z}}_{i,j}$:

$$S^{(ij)} = (S_1, \ldots, S_{j-1}, S_j^{(i)}, S_{j+1}, \ldots, S_m),$$

where $S_j^{(i)} = \{\mathbf{z}_{1,j}, \ldots, \mathbf{z}_{i-1,j}, \tilde{\mathbf{z}}_{i,j}, \mathbf{z}_{i+1,j}, \ldots, \mathbf{z}_{n,j}\}$. An algorithm $A$ is said to be on-average $\epsilon_{stab,j}$-model stable w.r.t. sub-dataset $S_j$ if

$$\mathbb{E}_{S, \widetilde{S}, I}\Big[\frac{1}{n} \sum_{i=1}^{n} \|A(S) - A(S^{(ij)})\|_2^2\Big] \leq \epsilon_{stab,j}^2.$$

We next recall standard regularity conditions including convexity, $L$-smoothness and $(\alpha, L)$-Hölder continuity. Hölder continuity of gradients generalizes smoothness and allows for non-smooth objectives, such as the $q$-norm hinge loss, which has Hölder continuous gradients, yet is not smooth if $q \in [1, 2)$. Indeed, Hölder continuity becomes Lipschitzness when $\alpha = 0$ and recovers the smoothness if $\alpha = 1$.

**Definition 3.2.** Let $f : \mathbb{R}^d \to \mathbb{R}$ be differentiable. Let $\alpha \in [0, 1], L > 0$.

● (Convexity). $f$ is convex if for all $\mathbf{w}, \mathbf{w}'$, $f(\mathbf{w}) \geq f(\mathbf{w}') + \langle \mathbf{w} - \mathbf{w}', \nabla f(\mathbf{w}') \rangle$.

● ($L$-Smoothness). $f$ is $L$-smooth if for all $\mathbf{w}, \mathbf{w}'$, $\|\nabla f(\mathbf{w}) - \nabla f(\mathbf{w}')\|_2 \leq L\|\mathbf{w} - \mathbf{w}'\|_2$.

● (($\alpha, L$)-Hölder continuity). $f$ has an $(\alpha, L)$-Hölder continuous gradient if for all $\mathbf{w}, \mathbf{w}'$,

$$\|\nabla f(\mathbf{w}) - \nabla f(\mathbf{w}')\|_2 \leq L\|\mathbf{w} - \mathbf{w}'\|_2^\alpha.$$

Lemma 3.3, adapted from Lei & Ying (2020), gives a connection between generalization error and on-average model stability. Instead of considering the whole training data, we focus on the empirical risk on a certain local dataset. Here $\mathbb{E}_{S,I}[\cdot]$ means the expectation w.r.t. $S$ and $I$.

**Lemma 3.3.** *Let $S, \tilde{S}$ and $S^{(ij)}$ be constructed as Definition 3.1. Let $A(S)$ be a randomized algorithm performed on $S$. If for any $z$, the function $\mathbf{w} \mapsto f(\mathbf{w}; z)$ is nonnegative and $L$-smooth, then for any $\gamma > 0$ and $j \in \{1, \ldots, m\}$, we have*

$$\mathbb{E}_{S,I}[F(A(S)) - F_{S_j}(A(S))] \leq \frac{L}{\gamma}\mathbb{E}_{S,I}[F_{S_j}(A(S))]$$
$$+ \frac{L+\gamma}{2n}\sum_{i=1}^{n}\mathbb{E}_{S,\tilde{S},I}[\|A(S^{(ij)}) - A(S)\|_2^2]. \quad (3.5)$$

For completeness, the proof of Lemma 3.3 is given in Appendix A.1. To facilitate comprehension, we write $a \lesssim b$ if there exists a constant $c > 0$ such that $a \leq c \cdot b$ and denote $a \asymp b$ if $a \lesssim b$ and $b \lesssim a$. Other most commonly used notations in this paper can be found in Table 3 and Table 4.

# 4. Generalization for Smooth Problems

In this section, we establish generalization guarantees for ASGD (Algorithm 1) on convex and smooth objectives. Our analysis proceeds in two steps. First, we derive the on-average model stability bounds for ASGD that capture worker participation. We then combine these stability estimates with Lemma 3.3 to bound generalization error, which, together with optimization error bounds, establish the final excess risk bounds. All proofs are deferred to Appendix B.

In ASGD's implementation, the worker arrival sequence, $\{m_t\}$, is largely determined by the underlying algorithm architecture, such as job assignment strategy, worker computation and communication speeds and network topology. Different system architectures induce different delay patterns. For instance, the cyclic delayed architecture proposed by Agarwal & Duchi (2011) adopts a pipelined update scheme, leading to a deterministic worker sequence $\{m_t\}$ and constant delays $\tau_t = m - 1$ for all $t \geq m$, where $m$ is the number of workers. To get a fine-grained understanding of generalization under asynchrony, we condition on a realized sequence $\{m_t\}$ and derive stability bounds that depend explicitly on worker participation. This allows our analysis to accommodate a wide range of ASGD architectures without imposing structural assumptions on the delay sequence.

## 4.1. Stability Analysis

The following theorem bounds the on-average model stability of ASGD w.r.t. each local dataset and reveals that optimization benefits the algorithmic stability.

**Theorem 4.1** (Stability of ASGD: smooth case). *Assume*

*for any $\mathbf{z}$, $\mathbf{w} \mapsto f(\mathbf{w}; \mathbf{z})$ is non-negative, convex and $L$-smooth. If $\{\mathbf{w}_t\}$ is generated by Algorithm 1 with $\eta_t$ satisfying $\left((1 + \frac{m}{n})\tau_t + m\right)\eta_t \leq 2/L$, then for any index $j \in [m]$, the algorithm output $A(S) = \mathbf{w}_{T+1}$ is on-average $\epsilon_{stab,j}$-model stable w.r.t. $S_j$, where*

$$\epsilon_{stab,j}^2 \leq \frac{32L}{n}\left(1 + \frac{N_T^M}{n}\right)\sum_{t=1}^{T}\eta_t^2\mathbb{I}_{[m_t=j]}\mathbb{E}_{S,I}[F_{S_{m_t}}(\mathbf{w}_{t-\tau_t})],$$
(4.1)

*and $\mathbb{I}_{[\cdot]}$ is an indicator function, $N_T^M = \max_{j\in[m]}\sum_{t=1}^{T}\mathbb{I}_{[m_t=j]}$ indicates the worker participation.*

*Remark 4.2* (Novelty). A main challenge of the stability and generalization analysis of ASGD lies in the bias induced by delays, which is absent in vanilla SGD. Most of the existing analyses either impose restrictive assumptions (e.g., constant (Deng et al., 2025) or bounded delay (Regatti et al., 2019)), or derive stability bounds that explicitly depend on delay magnitudes. For example, Deng et al. (2024) study convex and smooth objectives assuming data homogeneity and obtain an on-average stability bound of $O(1/\bar{\tau})$ when $n \asymp T$ and $\eta_t \asymp 1/(\bar{\tau}\sqrt{T})$. Here, $\bar{\tau} = \frac{1}{T}\sum_{t=1}^{T}\tau_t$ denotes the average delay. Notably, this bound is non-vacuous only for large average delay.

In contrast, Theorem 4.1 yields a stability bound that depends only on worker participation and empirical risks. Averaging Eq. (4.1) over $j \in [m]$ gives

$$\frac{1}{m}\sum_{j=1}^{m}\epsilon_{stab,j}^2 \leq \frac{32L}{mn}\left(1 + \frac{N_T^M}{n}\right)\sum_{t=1}^{T}\eta_t^2\mathbb{E}_{S,I}[F_{S_{m_t}}(\mathbf{w}_{t-\tau_t})],$$

where we have used $\sum_{j=1}^{m}\mathbb{I}_{[m_t=j]} = 1$. If the step size $\eta_t$ further satisfies the conditions of Lemma 4.4, combining this bound with the optimization guarantee implies

$$\frac{1}{m}\sum_{j=1}^{m}\epsilon_{stab,j}^2 \lesssim \frac{\eta_M}{mn}\left(1 + \frac{N_T^M}{n}\right)\left(\|\mathbf{w}_1 - \mathbf{w}^*\|^2 + F(\mathbf{w}^*)\sum_{t=1}^{T}\eta_t\right),$$
(4.2)

where $\eta_M$ is the maximum step size. The bound given in Eq. (4.2) scales as $O((n + N_T^M)/(mn^2))$ if $\eta_t \asymp 1/\sqrt{T}$. Taking $T \asymp mn$, the above bound is at worst of $O(1/n)$, showing that the stability bound is robust to delays. This improvement is largely driven by a better control over the accumulated biases from delayed updates (see the *second step* in Appendix B.1), which can be entirely eliminated with a suitable step size. Moreover, the analysis naturally extends to heterogeneous settings where $\mathbf{z}_{i,j} \sim \mathcal{D}_j$ with $\mathcal{D}_j \neq \mathcal{D}_{j'}$ for $j \neq j'$, showing that data heterogeneity does not inherently degrade the stability of ASGD.

*Remark 4.3* (Comparison). When $m = 1$, ASGD reduces to vanilla SGD and Eq. (4.1) recovers the classical stability bound (Lei & Ying, 2020):

$$\epsilon_{stab}^2 \leq \frac{32L}{n}\left(1 + \frac{T}{n}\right)\sum_{t=1}^{T}\eta_t^2\mathbb{E}_{S,I}[F_S(\mathbf{w}_t)],$$

*Table 1.* Stability bounds for convex and smooth problems. $G$-Lipschitzness; $Q$-quadratic loss; $H$-homogeneous data; $B$-boundedness; $\mathcal{R} = \sum_{t=1}^{T} \eta_t^2 \mathbb{E}[F_{S_{m_t}}(\mathbf{w}_{t-\tau_t})]$, where $\mathcal{R} = O(\eta + T\eta^2 + T\eta/n)$ when the constant step size satisfies the conditions of Lemma 4.4. Assume the dataset $S$ has size $mn$. Here, by homogeneous data we mean $S_1 = S_2 = \cdots = S_m$. The symbol $(^*)$ denotes that the original $\ell_2$ stability is converted into $\ell_1$ on-average model stability for comparison. Analysis in (Regatti et al., 2019) does not use the convexity assumption, and Liu et al. (2022) assumes strong convexity.

| Reference | Assumptions | | | | Step size | Stability |
|---|---|---|---|---|---|---|
| | $G$ | $Q$ | $H$ | $B$ | | |
| Regatti et al. (2019) | ✓ | × | × | Delay | $\eta_t \lesssim 1/t$ | $T^{\tau_M}/(mn\tau_M)$ |
| Liu et al. (2022) | ✓ | ✓ | ✓ | Delay | $\eta_t \lesssim 1$ | $\eta T/(mn)$ |
| Deng et al. (2025) | × | ✓ | ✓ | Delay, data, noise | $\eta_t \lesssim 1/\tau_M$ | $(T-\tau_M)/(mn\tau_M)$ |
| Deng et al. (2024) | × | × | ✓ | Parameter | $\eta_t \lesssim 1$ | $\sqrt{\eta + T\eta^2}(1+\sqrt{T/mn})/\sqrt{mn}+\eta\sqrt{\sum\tau_t}$ * |
| Thm. 4.1 | × | × | × | × | $\eta_t \lesssim 1/(\tau_t+m)$ | $\sqrt{\mathcal{R}}(1+\sqrt{N_T^M/n})/\sqrt{mn}$ * |

where $\epsilon_{stab}^2 := \mathbb{E}\big[\frac{1}{n}\sum_{i=1}^{n}\|A(S) - A(S^{(i)})\|_2^2\big]$. Under a similar data setting, Regatti et al. (2019) consider the following variant of ASGD:

$$\mathbf{w}_{t+1} = \mathbf{w}_t - \frac{\eta_t}{m}\sum_{j=1}^{m}\nabla f(\mathbf{w}_{t-\tau(j,t)}; \mathbf{z}_{i_t,j}),$$

where $\tau(j,t)$ means the delay of $j$-th worker at time step $t$, and get a uniform stability bound of $O(T^{\tau_M}/(n\tau_M))$ under smoothness, and Lipschitz assumptions, where $\tau_M = \max_{t\in[T]}\{\tau_t\}$. Compared to ours, this bound becomes loose when $\tau_M$ is large. For a homogeneous data setting with $S_1 = S_2 = \cdots = S_m$, Theorem 4.1 implies

$$\epsilon_{stab}^2 \leq \frac{32L}{n}\Big(1+\frac{T}{n}\Big)\sum_{t=1}^{T}\eta_t^2\mathbb{E}_{S,I}\big[F_S(\mathbf{w}_{t-\tau_t})\big]. \quad (4.3)$$

Deng et al. (2025) study ASGD with the update rule

$$\mathbf{w}_{t+1} = \mathbf{w}_t - \eta_t\nabla f\big(\mathbf{w}_{t-\tau}; \mathbf{z}_{i_t}\big) \quad (4.4)$$

under convex, smooth, and quadratic loss functions. Under a bounded data distribution $\big\|\frac{1}{n}\sum_{i=1}^{n}y_i\mathbf{x}_i\big\| \leq r$, and a bounded gradient noise assumption $\mathbb{E}\big[\|\nabla f(\mathbf{w}_{t-\tau_t}; \mathbf{z}_{i_t}) - \nabla F_S(\mathbf{w}_{t-\tau_t})\|^2\big] \leq \sigma^2$ for some $r, \sigma \in \mathbb{R}^+$, the following on-average stability bound is derived:

$$\frac{1}{n}\sum_{i=1}^{n}\mathbb{E}\Big[f\big(A(S^{(i)}); \mathbf{z}_i\big) - f\big(A(S); \mathbf{z}_i\big)\Big] = \widetilde{O}\Big(\frac{T-\tau}{n\tau}\Big).$$

Here, the notation $\widetilde{O}(\cdot)$ suppresses constant and polylogarithmic factors of $\tau$. Consider a cyclic delayed architecture where $\tau = m - 1$, the above bound scales as $\widetilde{O}((T-m)/(mn))$, which is meaningful only when the total number of iterations $T$ is moderate. Another recent work by Deng et al. (2024) investigates the same ASGD

as Eq. (4.4) and consider non-constant delays, denoted by $\{\tau_t\}$. Under assumptions of convexity, smoothness, and bounded parameter space (i.e., there exists $r > 0$ such that $\|\mathbf{w} - \mathbf{v}\| \leq r$ for all $\mathbf{w}, \mathbf{v} \in \Omega$), they derive an on-average model stability bound as follows.

$$\epsilon_{stab}^2 \lesssim \frac{L}{n}\Big(1+\frac{T}{n}\Big)\Big(\eta_1\|\mathbf{w}_1-\mathbf{w}^*\|^2+(Lr^2+F(\mathbf{w}^*))\sum_{t=1}^{T}\eta_t^2\Big) + L^2r^2\sum_{t=1}^{T}\sum_{l=1}^{\tau_t}\eta_t\eta_{t-l},$$

which is at best of $O(1/\bar{\tau})$ when $n \asymp T$ and $\eta_t \asymp 1/(\bar{\tau}\sqrt{T})$, and only applies to a setting with a large $\bar{\tau}$. Compared with prior works, our analysis removes the Lipschitzness, bounded-noise, bounded-data, and bounded-parameter assumptions, and yields a tight stability bound in Eq. (4.3) that is robust to delays and data heterogeneity. We summarize the comparison in Table 1.

### 4.2. Excess Risk Analysis

The following lemma gives the convergence rate of ASGD.

**Lemma 4.4.** *Under the assumptions of Theorem 4.1, if the step size satisfies* $(\max\{\tau_t + m - 1, 1\})\eta_t \leq 1/(4L)$, *then*

$$\sum_{t=1}^{T}\eta_t\mathbb{E}_{S,I}[F_{S_{m_t}}(\mathbf{w}_{t-\tau_t}) - F(\mathbf{w}^*)]$$
$$\leq \frac{1}{2}\|\mathbf{w}_1 - \mathbf{w}^*\|^2 + \sum_{t=1}^{T}\big(\frac{1}{2n} + 2L\eta_t\big)\eta_t F(\mathbf{w}^*). \quad (4.5)$$

*Remark* 4.5. Delay-adaptive step size of the form $(a + \tau_t)\eta_t \lesssim 1/L$ is standard for achieving convergence (Islamov et al., 2024; Koloskova et al., 2022). For instance, Mishchenko et al. (2022) utilize a closely related condition $\eta_t \lesssim \min\{1/(L\tau_t), 1/(Lm), 1/\sqrt{T}\}$.

*Table 2.* Excess risk bounds for convex problems. $S$-smoothness; $N$-low noise and $H$-homogeneous data. For simplicity, we assume the dataset $S$ has size $mn$ and let $N_T^M \asymp T/m$ and $m \leq n$. We denote $\bar{\tau}_T = \sum_{t=1}^{T} \tau_t/T$ and assume an $(\alpha, L)$-Hölder continuous gradient in the non-smooth setting. The symbol $(^*)$ denotes that the given computation cost is only required when $\alpha < \frac{1}{2}$. For the case of $\alpha \geq \frac{1}{2}$, $T \asymp m^2 n$ suffices.

| Reference | Assumptions | | | Delay pattern | Iteration | Excess Risk |
|---|---|---|---|---|---|---|
| | $S$ | $N$ | $H$ | | | |
| Deng et al. (2024) | ✓ | ✓ | ✓ | $\bar{\tau}_T \leq T^{1/4}$ | $T \asymp mn$ | $1/(mn) + 1/\bar{\tau}_T$ |
| Deng et al. (2024) | ✗ | ✓ | ✓ | $\bar{\tau}_T \leq T^{\min\{1/3, \alpha/(3-2\alpha)\}}$ | $T \asymp mn$ | $\bar{\tau}_T^{-\frac{1}{2}} + (mn)^{-\frac{1+\alpha}{2}}$ |
| Cor. 4.8 | ✓ | ✗ | ✓ | Arbitrary | $T \asymp mn$ | $1/\sqrt{mn}$ |
| Cor. 4.8 | ✓ | ✓ | ✓ | Arbitrary | $T \asymp m^2 n$ | $(\|\mathbf{w}_1 - \mathbf{w}^*\|^2 + 1)/(mn)$ |
| Thm. 5.4 | ✗ | ✗ | ✓ | $\hat{\tau}_M \asymp m$ | $T \asymp m^{\frac{3}{1+\alpha}} n^{\frac{2-\alpha}{1+\alpha}}$ $^*$ | $1/\sqrt{mn}$ |
| Thm. 6.2 | ✓ | ✗ | ✗ | $\tau_M \lesssim \sqrt{T}$ | $T \asymp mn$ | $1/\sqrt{mn}$ |

Combining stability bounds (Theorem 4.1) and convergence rates (Lemma 4.4) via the generalization-via-stability framework (Lemma 3.3), we get our main result on excess risks.

**Theorem 4.6** (Excess risk: smooth case). *Assume for any* $\mathbf{z}$, $\mathbf{w} \mapsto f(\mathbf{w}; \mathbf{z})$ *is non-negative, $L$-smooth and convex. Let* $\{\mathbf{w}_t\}$ *be produced by Algorithm 1. If the step size* $\{\eta_t\}$ *satisfies* $((1 + \frac{m}{n})\tau_t + m)\eta_t \leq 1/(4L)$, *then for any* $\gamma > 0$,

$$\mathbb{E}_{S,I}\big[F(A(S))\big] - F(\mathbf{w}^*) \qquad (4.6)$$
$$\lesssim \frac{1}{\sum_{t=1}^{T} \eta_t}\Big(1 + (1+\gamma)\frac{N_T^M}{n}\big(1 + \frac{N_T^M}{n}\big)\eta_M^2\Big)\|\mathbf{w}_1 - \mathbf{w}^*\|^2$$
$$+ \frac{F(\mathbf{w}^*)}{\sum_{t=1}^{T} \eta_t}\sum_{t=1}^{T} \eta_t\Big(\frac{1}{\gamma} + (1+\gamma)\frac{N_T^M}{n}\big(1 + \frac{N_T^M}{n}\big)\eta_M^2 + \frac{1}{n} + \eta_t\Big),$$

*where we denote*

$$A(S) = \frac{\sum_{t=1}^{T} \eta_t \mathbf{w}_{t-\tau_t}}{\sum_{t=1}^{T} \eta_t}, \quad \eta_M = \max_{t \in [T]}\{\eta_t\}. \qquad (4.7)$$

*Remark* 4.7 (Novelty). Most existing generalization analyses on ASGD assume each local worker can access the whole training dataset $S$ (Deng et al., 2024; Liu et al., 2022). Instead, we consider a more general setting where each worker can only access a subset of training data as considered in Regatti et al. (2019). In this case, consider the randomness of $i_t$, $\nabla f(\mathbf{w}_{t-\tau_t}; \mathbf{z}_{i_t, m_t})$ is an unbiased estimator to $\nabla F_{S_{m_t}}(\mathbf{w}_{t-\tau_t})$ but in general $\nabla F_{S_{m_t}}(\mathbf{w}_{t-\tau_t}) \neq \nabla F_S(\mathbf{w}_{t-\tau_t})$. This mismatch introduces a bias that is absent in homogeneous data settings. Nevertheless, we show that this bias does not destroy the generalization performance of ASGD by introducing a novel error decomposition, replacing the standard one in Eq. (3.4). For clarity, we assume that $\eta_t = \eta$. Let $A(S)$ be defined in Eq. (4.7). The convexity of

$f$ implies

$$\mathbb{E}_{S,I}[F(A(S)) - F(\mathbf{w}^*)] \leq \frac{1}{T}\sum_{t=1}^{T} \mathbb{E}_{S,I}[F(\mathbf{w}_{t-\tau_t}) - F(\mathbf{w}^*)].$$

Introduce the following decomposition:

$$\mathbb{E}_{S,I}\big[F(\mathbf{w}_{t-\tau_t}) - F(\mathbf{w}^*)\big]$$
$$= \underbrace{\mathbb{E}\big[F(\mathbf{w}_{t-\tau_t}) - F_{S_{m_t}}(\mathbf{w}_{t-\tau_t})\big]}_{:=g_t} + \underbrace{\mathbb{E}\big[F_{S_{m_t}}(\mathbf{w}_{t-\tau_t}) - F(\mathbf{w}^*)\big]}_{:=o_t}.$$

The optimization error term, $\frac{1}{T}\sum_{t=1}^{T} o_t$, can be controlled by Lemma 4.4. We therefore focus on the generalization error term $\frac{1}{T}\sum_{t=1}^{T} g_t$. Applying Lemma 3.3 and Theorem 4.1 with $j = m_t$ and $A(S) = \mathbf{w}_{t-\tau_t}$ yields

$$\frac{1}{T}\sum_{t=1}^{T} g_t \lesssim \frac{1}{\gamma T}\sum_{t=1}^{T} \mathbb{E}_{S,I}[F_{S_{m_t}}(\mathbf{w}_{t-\tau_t})]$$
$$+ \frac{\gamma\eta^2}{nT}\Big(1 + \frac{N_T^M}{n}\Big)\sum_{t=1}^{T}\sum_{k=1}^{t-\tau_t-1} \mathbb{I}_{[m_k = m_t]}\mathbb{E}_{S,I}[F_{S_{m_k}}(\mathbf{w}_{k-\tau_k})].$$

Recalling that $N_T^M = \max_{j \in [m]}\sum_{t=1}^{T} \mathbb{I}_{[m_t = j]}$ and using the non-negativity of $f$, we further bound

$$\sum_{t=1}^{T}\sum_{k=1}^{t-\tau_t-1} \mathbb{I}_{[m_k = m_t]}\mathbb{E}_{S,I}\big[F_{S_{m_k}}(\mathbf{w}_{k-\tau_k})\big] \leq \sum_{t=1}^{T}\sum_{k=1}^{T} \mathbb{I}_{[m_k = m_t]}$$
$$\times \mathbb{E}_{S,I}\big[F_{S_{m_k}}(\mathbf{w}_{k-\tau_k})\big] \leq N_T^M \sum_{k=1}^{T} \mathbb{E}_{S,I}\big[F_{S_{m_k}}(\mathbf{w}_{k-\tau_k})\big],$$

which can be controlled via Lemma 4.4. Based on the above discussion, we can get excess risk bounds without evaluating the full empirical risk $F_S(A(S))$.

**Corollary 4.8.** *Let assumptions of Theorem 4.6 hold. Assume $m \leq n$. Let $\{\mathbf{w}_t\}$ be produced by Algorithm 1 and $A(S) = \frac{1}{T}\sum_{t=1}^{T}\mathbf{w}_{t-\tau_t}$.*

• *(Constant $\eta_t$). If the sequence $\{m_t\}$ satisfies $\tau_t \lesssim \sqrt{T}$ and $N_T^M \lesssim T/k$, where $k \geq 1$ is an algorithm-independent constant, then we can take $T \asymp kn$ and $\eta \asymp T^{-\frac{1}{2}}$ to derive*

$$\mathbb{E}[F(A(S))] - F(\mathbf{w}^*) \lesssim \frac{1}{\sqrt{kn}}\big(\|\mathbf{w}_1 - \mathbf{w}^*\|^2 + F(\mathbf{w}^*)\big).$$

• *(Adaptive $\eta_t$). For any $\{m_t\}$, taking $T \asymp mn$ and*

$$\eta_t = \begin{cases} \min\{\frac{1}{\sqrt{T}}, \frac{1}{12mL}\}, & \text{if } \tau_t \leq m, \\ \min\{\frac{1}{\sqrt{T}}, \frac{1}{4L((1+\frac{m}{n})\tau_t+m)}\}, & \text{if } \tau_t > m, \end{cases}$$

*yields an excess risk bound of $O\big(\frac{1}{\sqrt{mn}}(\|\mathbf{w}_1 - \mathbf{w}^*\|^2 + \frac{m}{k}F(\mathbf{w}^*))\big)$, where $k := T/N_T^M$.*

• *(Low noise). If $F(\mathbf{w}^*) \lesssim \frac{1}{mn}$ and $T/N_T^M \asymp m$, we can take $T \asymp m^2n$ and $\eta_t = \min\{\frac{1}{12mL}, \frac{1}{4L((1+\frac{m}{n})\tau_t+m)}\}$ to derive an excess risk bound of $O\big(\frac{1}{mn}(\|\mathbf{w}_1 - \mathbf{w}^*\|^2 + 1)\big)$.*

*Remark* 4.9. Consider the cyclic delayed architecture (Agarwal & Duchi, 2011) or an architecture where each worker contributes evenly, i.e. $N_T^M \asymp T/m$. In this case, Corollary 4.8 achieves the minimax-optimal bounds of $O(1/\sqrt{mn})$, showing that asynchrony has a provably limited impact on the generalization of ASGD. In the imbalanced case where updates are dominated by a fast worker (i.e. $k \approx 1$), our analysis still implies a non-vacuous excess risk bound of $O(1/\sqrt{n})$, recovering the classical rate of SGD and demonstrating ASGD's robustness to delays in generalization. Utilizing delay-adaptive step size, we can remove the constraints on delays and develop excess risk bounds that depend on the worker participation and interpolate between the typical rate $O((m/k)/\sqrt{mn})$ and the fast rate $O(1/mn)$. In the homogeneous data setting where each worker can access the whole dataset $S$ with size $n$, Deng et al. (2024) additionally assume Lipschitz objectives, $F(\mathbf{w}^*) = 0$, and $\bar{\tau} \leq T^{1/4}$ to derive an excess risk bound of order $O(1/\bar{\tau} + 1/n)$. In contrast, our analysis yields an excess risk of order $O(1/\sqrt{n})$ in general and achieves the fast rate $O(1/n)$ under a low-noise condition, without these additional restrictions, and is therefore sharper. Table 2 summarizes our conclusions regarding the excess risk and provides a comparison with prior work.

# 5. Generalization for Non-smooth Problems

We now consider non-smooth loss functions with $(\alpha, L)$-Hölder continuous gradients. The generalization-via-stability lemma for non-smooth losses and other auxiliary lemmas and proofs are deferred to Appendix C.

## 5.1. Stability Analysis

The following theorem indicates that the cost of relaxing the smoothness assumption to $(\alpha, L)$-Hölder continuity can be controlled by choosing an appropriate step size.

**Theorem 5.1** (Stability of ASGD: non-smooth case). *Assume that for any $\mathbf{z}$, $\mathbf{w} \mapsto f(\mathbf{w}; \mathbf{z})$ is non-negative and convex, and that $\mathbf{w} \mapsto \nabla f(\mathbf{w}; \mathbf{z})$ is $(\alpha, L)$-Hölder continuous with $\alpha \in [0, 1)$. Let $\{\mathbf{w}_t\}$ be produced by Algorithm 1 and $A(S) = \mathbf{w}_{T+1}$. Then, $A$ is on-average $\epsilon_{stab,j}$-model stable w.r.t. subset $S_j$, where*

$$\epsilon_{stab,j}^2 \leq \frac{16c_{\alpha,L}^2}{n}\Big(1+\frac{N_T^M}{n}\Big)\sum_{t=1}^{T}\eta_t^2\mathbb{I}_{[m_t=j]}\mathbb{E}_{S,I}\big[F_{S_{m_t}}(\mathbf{w}_{t-\tau_t})\big]^{\frac{2\alpha}{1+\alpha}}$$

$$+\frac{2(1-\alpha)}{1+\alpha}\sum_{t=1}^{T}\Big((1+\frac{m}{n})\tau_t+m\Big)^{\frac{1+\alpha}{1-\alpha}}(2^{-\alpha}L\eta_t)^{\frac{2}{1-\alpha}},$$

(5.1)

*with*

$$c_{\alpha,L} = \begin{cases} (1+\frac{1}{\alpha})^{\frac{\alpha}{1+\alpha}}L^{\frac{1}{1+\alpha}}, & \text{if } \alpha > 0, \\ \sup_{\mathbf{z}}\|\nabla f(\mathbf{0}; \mathbf{z})\|_2 + L, & \text{if } \alpha = 0. \end{cases}$$

(5.2)

*Remark* 5.2. Under homogeneous data where workers access the entire $S$, Eq. (5.1) can be rewritten as

$$\epsilon_{stab}^2 \leq \frac{16c_{\alpha,L}^2}{n}\Big(1+\frac{T}{n}\Big)\sum_{t=1}^{T}\eta_t^2\mathbb{E}_{S,A}\big[F_S(\mathbf{w}_{t-\tau_t})\big]^{\frac{2\alpha}{1+\alpha}}+$$

$$\frac{2(1-\alpha)}{1+\alpha}\sum_{t=1}^{T}\Big((1+\frac{m}{n})\tau_t+m\Big)^{\frac{1+\alpha}{1-\alpha}}(2^{-\alpha}L\eta_t)^{\frac{2}{1-\alpha}}. \quad (5.3)$$

Under an extra assumption on the bounded parameter space, Deng et al. (2024) get the following stability bounds

$$\epsilon_{stab}^2 \lesssim \frac{1+T/n}{n}\sum_{t=1}^{T}\eta_t^2\mathbb{E}_{S,A}\big[F_S(\mathbf{w}_{t-\tau_t})^{\frac{2\alpha}{1+\alpha}}\big]$$

$$+\sum_{t=1}^{T}\sum_{l=1}^{\tau_t}\eta_t\eta_{t-l} + \sum_{t=1}^{T}\eta_t^{\frac{2}{1-\alpha}}. \quad (5.4)$$

For brevity, let $\eta_t = \eta$, $\tau_t \asymp m$, and $m \leq n$. The last term in Eq. (5.3), which captures the cost of relaxing smoothness, scales as $O\big(T\eta^{\frac{2}{1-\alpha}}m^{\frac{1+\alpha}{1-\alpha}}\big)$, whereas the last two terms in (5.4) scale as $O(T\eta^2m)$. Under the standard step-size condition $\eta \lesssim 1/m$ (Deng et al., 2024; Stich & Karimireddy, 2020; Islamov et al., 2024), our bound is strictly tighter. Moreover, choosing $\eta_t \asymp 1/((\tau_t + m)\sqrt{T})$ yields a contribution of order $O(m^{-1}T^{-\frac{\alpha}{1-\alpha}})$ for the last term in Eq. (5.3), ensuring a non-vacuous stability guarantee.

## 5.2. Excess Risk Analysis

We next derive excess risk bounds by combining stability and optimization guarantees. The following lemma controls the optimization error of ASGD for non-smooth objectives.

**Lemma 5.3.** *Let assumptions of Theorem 5.1 hold, and let $\{\mathbf{w}_t\}$ be produced by Algorithm 1. If the step size satisfies $(\max\{\tau_t + m - 1, 1\})\eta_t \leq 1/(4L^{\frac{1}{\alpha}})$ for any $t \in [T]$ [1], then we have*

$$\sum_{t=1}^{T} \eta_t \mathbb{E}_{S,I}\Big[ F_{S_{m_t}}(\mathbf{w}_{t-\tau_t}) - F(\mathbf{w}^*) \Big] \lesssim \|\mathbf{w}_1 - \mathbf{w}^*\|^2$$
$$+\sum_{t=1}^{T}(\tau_t + m)\eta_t^2 + \Big(\frac{1}{n}\sum_{t=1}^{T}(\tau_t+m)\eta_t^2 + \sum_{t=1}^{T}\eta_t^2\Big) F^{\frac{2\alpha}{1+\alpha}}(\mathbf{w}^*).$$

Combining Theorem 5.1 and Lemma 5.3, we obtain excess risk guarantees of ASGD for non-smooth objectives.

**Theorem 5.4.** *Let assumptions of Theorem 5.1 hold. Let $\{\mathbf{w}_t\}$ be produced by Algorithm 1 with a constant step size, i.e. $\eta_t = c\hat{\tau}_M^{-\zeta}T^{-\theta}$, where $\hat{\tau}_M := \max_{t \in [T]}\{\tau_t + m\}$ and $c, \zeta, \theta > 0$ are constants. Assume $F(\mathbf{w}^*) \leq 1$ and $m \leq n$. Consider the balanced worker participation where $N_T^M \asymp T/m$ and $\hat{\tau}_M \asymp m$.*

• *If $\alpha \geq 1/2$, taking $T \asymp m^2 n$ and $\zeta = \theta = \frac{1}{2}$ yields*

$$\mathbb{E}[F(A(S))] - F(\mathbf{w}^*) \lesssim (mn)^{-\frac{1}{2}}\big(1 + F^{\frac{2\alpha}{1+\alpha}}(\mathbf{w}^*)\big), \quad (5.5)$$

*where $A(S) = \sum_{t=1}^{T}\mathbf{w}_{t-\tau_t}/T$.*

• *If $\alpha < 1/2$, we can take $T \asymp m^{\frac{3}{1+\alpha}}n^{\frac{2-\alpha}{1+\alpha}}$, $\zeta = \frac{1+\alpha}{4-2\alpha}$ and $\theta = \frac{3-3\alpha}{4-2\alpha}$ to get Eq. (5.5).*

*Remark* 5.5. Consider ASGD with a cyclic delayed architecture for the Lipschitz problem (i.e. $\alpha = 0$). Taking $T \asymp m^3 n^2$ and $\eta \asymp \hat{\tau}_M^{-\frac{1}{4}}T^{-\frac{3}{4}}$, Eq. (5.5) scale as $O(\frac{1}{\sqrt{mn}})$. Assuming convexity, Lipschitzness, Hölder continuity, low-noise condition and bounded parameter space, Deng et al. (2024) take $T \asymp n$ and $\eta \asymp 1/(\bar{\tau}\sqrt{T})$ to obtain the risk bound of $O(\bar{\tau}^{-\frac{1}{2}} + n^{\frac{1+\alpha}{2}})$, which is worse than $O(n^{-\frac{1}{6}})$ since they require $\bar{\tau} \lesssim T^{\min\{\frac{1}{3}, \frac{\alpha}{3-2\alpha}\}}$. In comparison, under a more general data setting, we remove these assumptions and achieve sharper rates.

# 6. Extension to Heterogeneous Data

We now extend the analysis to the heterogeneous data setting. Assume that worker $j \in [m]$ has access to a local dataset $S_j = \{z_{1,j}, \ldots, z_{n,j}\}$, where samples are drawn i.i.d. from a worker-specific distribution $\mathcal{D}_j$. Define the local and global population risks as

$$F_j(\mathbf{w}) = \mathbb{E}_{\mathbf{z} \sim \mathcal{D}_j}[f(\mathbf{w}; \mathbf{z})], \quad F(\mathbf{w}) = \frac{1}{m}\sum_{j=1}^{m}F_j(\mathbf{w}).$$

Convergence guarantees for standard ASGD with arbitrary delays on heterogeneous data are compromised by a bias towards fast workers. Instead, we focus on the *random* ASGD

---

[1]This assumption is only needed for $\alpha > 0$, that is, when $\alpha = 0$, Lemma 5.3 holds for any non-negative sequence $\{\eta_t\}$.

proposed by Koloskova et al. (2022), whose pseudocode can be found in Appendix D. Unlike Algorithm 1, after receiving an update at iteration $t$, the server assigns the next job to a worker $a_{t+1}$ sampled uniformly from $[m]$, independently of the past. We collect the sequence $\{a_t\}$ as $A$.

For clarity, we retain some notations from Islamov et al. (2024). A gradient computation is indexed by $(j, t) \in [m] \times \mathbb{N}^+$, meaning that worker $j$ computes a stochastic gradient on model $\mathbf{w}_t$. Let $\mathcal{R}_t$ and $\mathcal{A}_t$ denote the sets of received and assigned jobs prior to time step $t$. For any $t \in \{1, \ldots, T\}$, they evolve as

$$\mathcal{R}_{t+1} = \mathcal{R}_t \cup \{(m_t, t - \tau_t)\}, \quad \mathcal{A}_{t+1} = \mathcal{A}_t \cup \{(a_{t+1}, t+1)\}.$$

By the settings of *random ASGD*, we naturally define $\mathcal{R}_1 = \emptyset$, $\mathcal{A}_1 = \{(j, 1) | j \in \{1, \ldots, m\}\}$.

## 6.1. Optimization and Excess Risk Analysis

Lemma 6.1 extends the convergence results in Koloskova et al. (2022) to convex and smooth case, removing the Lipschitz, bounded noise, and bounded data heterogeneity assumption. The proofs can be found in Appendix D.

**Lemma 6.1.** *Assume for any $\mathbf{z}$, $\mathbf{w} \mapsto f(\mathbf{w}; \mathbf{z})$ is $L$-smooth, convex. Let $\{\mathbf{w}_t\}$ be produced by random SGD and $\mathbf{w}^*$ denote the population risk minimizer.*

• *(Constant $\eta_t$). If we assume that the delays are bounded by a constant $\tau_M$, then taking $(\tau_M + m)\eta \leq \frac{1}{4L}$ yields*

$$\sum_{t=2}^{T}\eta \mathbb{E}_{S,I,A}\Big[F_S(\mathbf{w}_t) - F(\mathbf{w}^*)\Big] \lesssim \|\mathbf{w}_1 - \mathbf{w}^*\|^2$$
$$+ F(\mathbf{w}^*)\Big(\big(\frac{\tau_M + m - 1}{mn} + 1\big)T\eta^2 + 1\Big). \quad (6.1)$$

• *(Adaptive $\eta_t$). If we assume that the delays and the workers are independent, then taking $(\tau_t + m)\eta_t \leq \frac{1}{4L}$ yields*

$$\sum_{t=2}^{T}\eta_{t+\hat{\tau}_t}\mathbb{E}_{S,I,A}\Big[F_S(\mathbf{w}_t) - F(\mathbf{w}^*)\Big] \lesssim \|\mathbf{w}_1 - \mathbf{w}^*\|^2$$
$$+ F(\mathbf{w}^*)\Big(\sum_{t=2}^{T}(\frac{\hat{\tau}_t + m - 1}{mn} + 1)\eta_{t+\hat{\tau}_t}^2 + 1\Big), \quad (6.2)$$

*where for $t \in \{2, 3, \cdots\}$, $t + \hat{\tau}_t$ denotes the time when the gradients calculated on $\mathbf{w}_t$ is applied.*

Combining Lemma 6.1 with the stability analysis in Section 4.1 yields the following excess risk bound.

**Theorem 6.2** (Excess risk: heterogeneous data). *Let the conditions of Theorem 4.1 and Lemma 6.1 hold. Assume $m \leq n$ and ignore all the algorithm-independent constants.*

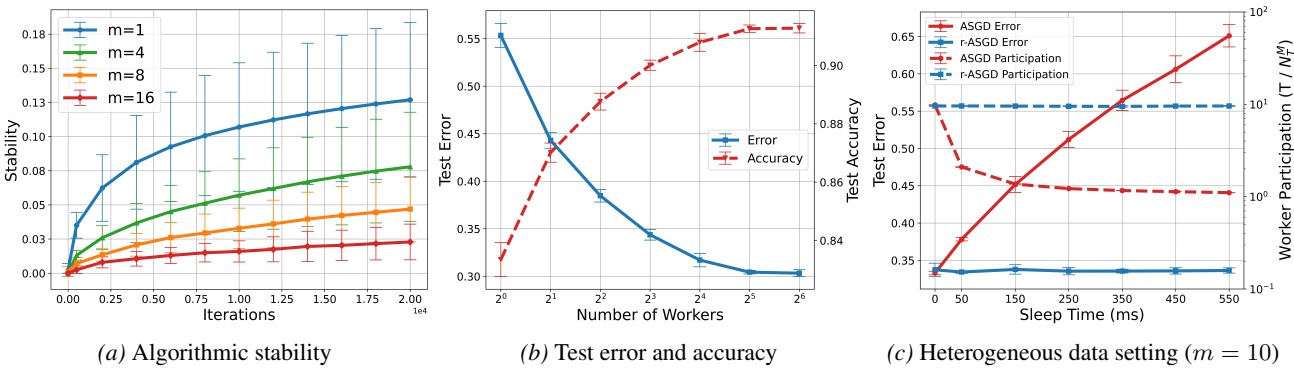

*(a)* Algorithmic stability        *(b)* Test error and accuracy        *(c)* Heterogeneous data setting ($m = 10$)

*Figure 1.* Experiment on MNIST. Left panel: evolution of algorithmic stability of ASGD with varying numbers of workers. Middle panel: impact of worker numbers on testing error and accuracy. Right panel: effects of straggling behavior on worker participation and testing errors. $T/N_T^M$ reflects the constant $k$ defined in Corollary 4.8.

- *(Constant $\eta_t$).* If we assume $\tau_M \lesssim \sqrt{T}$, then taking $T \asymp mn$, $\eta \asymp 1/\sqrt{T}$ and $A(S) = \frac{1}{T-1}\sum_{t=2}^{T}\mathbf{w}_t$ yields

$$\mathbb{E}[F(A(S))] - F(\mathbf{w}^*) \lesssim \frac{1}{\sqrt{mn}}\Big(1 + \frac{\mathbb{E}_A[N_T^M]}{n}\Big). \tag{6.3}$$

- *(Adaptive $\eta_t$).* If the delays are independent of workers, then taking $T, m, n$ satisfying $T \asymp mn$, applying the delay-adaptive step size chosen in Corollary 4.8, and letting $A(S) = \frac{\sum_{t=2}^{T}\eta_{t+\hat{\tau}_t}\mathbf{w}_t}{\sum_{t=2}^{T}\eta_{t+\hat{\tau}_t}}$ , Eq. (6.3) holds for arbitrary delays.

*Remark* 6.3 (Illustration). The condition $\tau_t \leq \tau_M$ can be satisfied by tracking the accumulated delay of each worker and forcing the server to wait for those whose delays approach $\tau_M$. The assumption that delays are independent of the workers has been briefly discussed in Koloskova et al. (2022). As noted therein, without this assumption, an adaptive $\eta_t$ introduces a bias towards workers that compute more quickly, which leads to convergence to an incorrect objective. Let $N_T^A := \max_{j\in[m]}\sum_{t=1}^{T}\mathbb{I}_{[a_t=j]}$. It holds $N_T^M \leq N_T^A + 1$. The classical balls-into-bins result (Raab & Steger, 1998) gives $\mathbb{E}_A[N_T^A] = \frac{T}{m} + O(\sqrt{T\log m/m})$. Ignoring polylogarithmic factors in $m$, we obtain $\mathbb{E}_A[N_T^M] = \tilde{O}(T/m)$, and hence an excess risk of order $\tilde{O}(1/\sqrt{mn})$, demonstrating that *random* ASGD generalizes well on heterogeneous data. To our knowledge, this is the first excess risk guarantee for ASGD under heterogeneous data.

## 7. Experimental Results

In this section, we conduct experiments on smooth, non-smooth and non-convex problems to verify the algorithmic stability and generalization performance of ASGD and its variants. We present the results for convex and smooth problems on the MNIST dataset. We defer the experimental

details, supplementary experiments on non-smooth and non-convex problems, and comparisons across different ASGD variants to Appendix E. Since $F(\mathbf{w}^*)$ is fixed for a certain task, test error can directly reflect the excess risk. The left and middle panel in Figure 1 shows that increasing the number of workers makes ASGD more stable during the training and also reduces the test error, which is consistent with Theorem 4.1 and Corollary 4.8. We simulate straggling behavior in training by forcing workers to sleep for a short time, and replicate data heterogeneity by restricting each worker to access only one category of labeling data. The right panel of Figure 1 shows the results under imbalance training and heterogeneous data setting and illustrates that while $N_T^M$ increases, the test error of ASGD also increases due to the imbalanced training. In contrast, random ASGD keeps the worker balance and therefore obtains a good generalization performance. These results further validate our theoretical findings in Sections 4 and 6.

## 8. Conclusion

We provided a stability-based generalization analysis of Asynchronous SGD. For convex and smooth losses, we derived sharp stability and excess risk bounds under minimal assumptions, showing that the benefit of increased data from more workers outweighs the adverse effect of delays. In balanced regimes, the resulting bounds interpolate between the minimax rate $O(1/\sqrt{mn})$ and the fast rate $O(1/mn)$. We further extended the analysis to non-smooth objectives via Hölder-continuous gradients and to heterogeneous data via *random* ASGD. These results indicate that asynchrony and heterogeneity have a provably benign effect on stability and generalization of ASGD, thereby deepening our theoretical understanding of the algorithm under realistic assumptions. Important directions for future work include extending stability-based generalization guarantees to non-convex problems and eliminating the dependency on maximum delay in the heterogeneous data setting.

## Acknowledgement

We are grateful to the area chair and reviewers for their constructive comments and suggestions. The work of Yunwen Lei is partially supported by the Research Grants Council of Hong Kong [Project Nos. 17302624, 17305425].

## Impact Statement

This paper presents work whose goal is to advance the field of Machine Learning. There are many potential societal consequences of our work, none which we feel must be specifically highlighted here.

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

## Appendix Overview

The appendix is organized as follows.

- **Appendix A** collects commonly used notations and auxiliary lemmas employed throughout the analysis.

- **Appendix B** presents omitted proofs for the stability and generalization analysis under convex and smooth objectives.

- **Appendix C** contains the generalization-via-stability lemma for non-smooth losses with $(\alpha, L)$-Hölder continuous gradients, together with all supporting proofs for the generalization analysis of non-smooth problems.

- **Appendix D** includes proofs for the heterogeneous data setting, including optimization and excess risk bounds for random ASGD.

- **Appendix E** reports supplementary experimental details and results for non-smooth and non-convex problems.

## A. Notations and Basic Inequalities

In this section, we first summarize the most commonly used notations in this paper in Table 3 and Table 4.

| Notation | Description |
|---|---|
| $m$ | Number of workers participating in asynchronous training. |
| $n$ | Size of local dataset at each worker. |
| $T$ | Total number of iterations for parameter updates at the server. |
| $\eta_t$ | Step size used at iteration $t$. |
| $\eta_M$ | Maximum of step size used in first $T$ iterations. |
| $m_t$ | Index of the worker whose gradient is applied at time $t$. |
| $a_t$ | Index of the worker that is assigned a job by server at time $t - 1$. |
| $\tau_t$ | Delay of the gradient applied at time $t$. |
| $\tau_M$ | The maximum delay: unless otherwise specified $\tau_M := \max_{t \in \mathbb{N}+}\{\tau_t\}$ |
| $\hat{\tau}_M$ | A constant related to the maximum delay: $\hat{\tau}_M := \max_{t \in [T]}\{\tau_t + m\}$ |
| $i_t$ | Index of the data sample used by worker $m_t$ at time $t - \tau_t$. |
| $\mathbf{w}_t$ | the model outputted by ASGD at the $t$-th iteration. |
| $\mathcal{R}_t$ | The set of all received jobs before the time step $t$ |
| $\mathcal{A}_t$ | The set of all assigned jobs before the time step $t$ |
| $M$ | Sequence of worker arrivals $\{m_t\}_{t=1}^T$. |
| $I$ | Sequence of sampled indices $\{i_t\}_{t=1}^T$. |
| $A$ | Sequence of worker assignments $\{a_t\}_{t=1}^T$. |
| $N_T^M$ | Maximum updates done by a single worker, i.e. $\max_{j \in [m]} \sum_{t=1}^T \mathbb{I}[m_t = j]$ |

*Table 3.* Asynchronous SGD and delay-related notation.

To facilitate the study of perturbed dataset $S^{(ij)}$ defined in Definition 3.1 in stability analysis, let $\mathbf{w}_t$ and $\mathbf{w}_t^{(ij)}$ denote the model produced by Algorithm 1 at $t$-th iteration running on dataset $S$ and $S^{(ij)}$, respectively. Further, let $S_{i_t,m_t}$ and $S_{i_t,m_t}^{(ij)}$ be the sample point chosen from $S$ and $S^{(ij)}$ for the update of model $\mathbf{w}_t$ and $\mathbf{w}_t^{(ij)}$, i.e.,

$$S_{i_t,m_t} = \mathbf{z}_{i_t,m_t} \quad \text{and} \quad S_{i_t,m_t}^{(ij)} = \begin{cases} \tilde{\mathbf{z}}_{i,j}, & \text{if } i_t = i \text{ and } m_t = j, \\ \mathbf{z}_{i_t,m_t}, & \text{else.} \end{cases} \tag{A.1}$$

Throughout the paper, expectations are taken with respect to (w.r.t.) different sources of randomness. We use $\mathbb{E}_I[\cdot]$ and $\mathbb{E}_A[\cdot]$ to denote the expectation w.r.t. $I = \{i_t\}$ and $A = \{a_t\}$, respectively. Moreover, $\mathbb{E}_{S,\tilde{S},I}[\cdot]$ ($\mathbb{E}_{S,\tilde{S},I,A}[\cdot]$) or simply $\mathbb{E}[\cdot]$ denotes expectation w.r.t. both $S$ and $\widetilde{S}$ and all algorithmic randomness of ASGD (*random ASGD*).

The following are some of the basic inequalities widely used in our analyses.

| Notation | Description | Expression |
|---|---|---|
| $\mathfrak{D}_{t,i,j}$ | Gradient difference under the same sample. | $\nabla f(\mathbf{w}_{t-\tau_t}; S_{i_t,m_t}) - \nabla f(\mathbf{w}_{t-\tau_t}^{(ij)}; S_{i_t,m_t})$. |
| $\mathfrak{C}_{t,i,j}$ | Gradient difference induced by data replacement. | $\nabla f(\mathbf{w}_{t-\tau_t}; \mathbf{z}_{i,j}) - \nabla f(\mathbf{w}_{t-\tau_t}^{(ij)}; \tilde{\mathbf{z}}_{i,j})$. |
| $e_t$ | Error term caused by delays. | $-2\eta_t \mathbb{E}_I\left[ \mathbb{I}_{[i_t \neq i \text{ or } m_t \neq j]} \langle \mathbf{w}_t - \mathbf{w}_{t-\tau_t} - (\mathbf{w}_t^{(ij)} - \mathbf{w}_{t-\tau_t}^{(ij)}), \mathfrak{D}_{t,i,j} \rangle \right]$. |
| $\epsilon_{stab,j}^2$ | On-average stability parameter w.r.t. worker $j$. | $\mathbb{E}_{S,\tilde{S},I}\left[ \frac{1}{n} \sum_{i=1}^n \|A(S) - A(S^{(ij)})\|^2 \right]$. |
| $\tilde{\Delta}_{t,i,j}$ | Maximum expected distance between coupled iterates up to time $t$. | $\max_{l \in [t]} \left( \mathbb{E}_{S,\tilde{S},I} \|\mathbf{w}_l - \mathbf{w}_l^{(ij)}\|^2 \right)^{1/2}$. |

*Table 4.* Stability and generalization analysis notation.

**Lemma A.1.** *Let $a, b \geq 0$. If $x^2 \leq ax + b$, then $x^2 \leq a^2 + 2b$.*

**Lemma A.2** (Young's Inequality). *If $p, q \in \mathbb{R}^+$ such that $\frac{1}{p} + \frac{1}{q} = 1$, then for any $a, b \in \mathbb{R}$,*

$$ab \leq \frac{1}{p}|a|^p + \frac{1}{q}|b|^q.$$

**Lemma A.3** (Cauchy-Schwarz Inequality). *For any random variables $X, Y \in L^2$, where $L^2 = \{X : \mathbb{E}[|X|^2] < \infty\}$, we have*

$$|\mathbb{E}[XY]| \leq \left( \mathbb{E}[X^2] \right)^{\frac{1}{2}} \left( \mathbb{E}[Y^2] \right)^{\frac{1}{2}}.$$

**Lemma A.4** (Lemma A.1 in Lei & Ying (2020)). *Assume for all $\mathbf{z} \in \mathcal{Z}$, the map $\mathbf{w} \mapsto f(\mathbf{w}; \mathbf{z})$ is nonnegative, and $\mathbf{w} \mapsto \nabla f(\mathbf{w}; \mathbf{z})$ is $(\alpha, L)$-Hölder continuous where $\alpha \in [0, 1]$. Then we have*

$$\|\nabla f(\mathbf{w}; \mathbf{z})\|_2 \leq c_{\alpha,L} f^{\frac{\alpha}{1+\alpha}}(\mathbf{w}; \mathbf{z}), \quad \forall \mathbf{w} \in \mathcal{W}, \mathbf{z} \in \mathcal{Z},$$

*where*

$$c_{\alpha,L} = \begin{cases} (1 + \frac{1}{\alpha})^{\frac{\alpha}{1+\alpha}} L^{\frac{1}{1+\alpha}}, & \text{if } \alpha > 0, \\ \sup_{\mathbf{z}} \|\nabla f(\mathbf{0}; \mathbf{z})\|_2 + L, & \text{if } \alpha = 0. \end{cases}$$

The following two lemmas establish the co-coercivity of convex functions with Hölder continuous gradients ($\alpha \in [0, 1]$).

**Lemma A.5** (Lemma D.2 in Lei & Ying (2020)). *Assume for all $\mathbf{z} \in \mathcal{Z}$, the map $\mathbf{w} \mapsto f(\mathbf{w}; \mathbf{z})$ is convex, and $\mathbf{w} \mapsto \nabla f(\mathbf{w}; \mathbf{z})$ is $(\alpha, L)$-Hölder continuous where $\alpha \in [0, 1]$. Then we have*

$$\langle \mathbf{w} - \mathbf{w}', \nabla f(\mathbf{w}; \mathbf{z}) - \nabla f(\mathbf{w}'; \mathbf{z}) \rangle \geq \frac{2\alpha L^{-\frac{1}{\alpha}}}{1+\alpha} \|\nabla f(\mathbf{w}; \mathbf{z}) - \nabla f(\mathbf{w}'; \mathbf{z})\|_2^{\frac{1+\alpha}{\alpha}}. \tag{A.2}$$

**Lemma A.6** (Proposition 1(b) in Ying & Zhou (2017)). *Let conditions in Lemma A.5 holds. Then we have*

$$\langle \mathbf{w} - \mathbf{w}', \nabla f(\mathbf{w}; \mathbf{z}) \rangle - \left( f(\mathbf{w}; \mathbf{z}) - f(\mathbf{w}'; \mathbf{z}) \right) \geq \frac{\alpha L^{-\frac{1}{\alpha}}}{1+\alpha} \|\nabla f(\mathbf{w}; \mathbf{z}) - \nabla f(\mathbf{w}'; \mathbf{z})\|_2^{\frac{1+\alpha}{\alpha}}. \tag{A.3}$$

*Remark* A.7. When $\alpha = 0$, Eq. (A.2) and Eq. (A.3) holds by the convexity of $f$. When $\alpha = 1$, Lemma A.4 indicates the self-bounding property of smooth function, that is, $\|\nabla f(\mathbf{w}; \mathbf{z})\|_2 \leq \sqrt{2L} f^{\frac{1}{2}}(\mathbf{w}; \mathbf{z})$, and the above two lemmas show the co-coercivity of convex and $L$-smooth functions that is widely used in optimization analysis (Nesterov, 2013). In this case, Eq. (A.2) and Eq. (A.3) have the following forms:

$$\langle \mathbf{w} - \mathbf{w}', \nabla f(\mathbf{w}; \mathbf{z}) - \nabla f(\mathbf{w}'; \mathbf{z}) \rangle \geq \frac{1}{L} \|\nabla f(\mathbf{w}; \mathbf{z}) - \nabla f(\mathbf{w}'; \mathbf{z})\|_2^2 \tag{A.4}$$

and

$$\langle \mathbf{w} - \mathbf{w}', \nabla f(\mathbf{w}; \mathbf{z}) \rangle - \big( f(\mathbf{w}; \mathbf{z}) - f(\mathbf{w}'; \mathbf{z}) \big) \geq \frac{1}{2L} \|\nabla f(\mathbf{w}; \mathbf{z}) - \nabla f(\mathbf{w}'; \mathbf{z})\|_2^2. \tag{A.5}$$

In the following subsection, for the sake of completeness, we give the proof to Lemma 3.3 which is a direct extension of Theorem 2 in Lei & Ying (2020).

### A.1. Proof for Lemma 3.3

*Proof.* For a certain index $j \in [m]$, since $\tilde{S}$ is a independent copy of $S$, we know from the symmetry that

$$\mathbb{E}_{S,I}\big[F(A(S)) - F_{S_j}(A(S))\big] = \mathbb{E}_{S,\tilde{S},I}\Big[\frac{1}{n}\sum_{i=1}^n f(A(S); \tilde{\mathbf{z}}_{i,j}) - \frac{1}{n}\sum_{i=1}^n f(A(S); \mathbf{z}_{i,j})\Big]$$

$$= \frac{1}{n}\sum_{i=1}^n \mathbb{E}_{S,\tilde{S},I}\big[f(A(S^{(ij)}); \mathbf{z}_{i,j}) - f(A(S); \mathbf{z}_{i,j})\big]. \tag{A.6}$$

Note that the function $\mathbf{w} \mapsto f(\mathbf{w}; \mathbf{z})$ is smooth. The following inequality holds by the property of smoothness.

$$f(A(S^{(ij)}); \mathbf{z}_{i,j}) - f(A(S); \mathbf{z}_{i,j}) \leq \big\langle A(S^{(ij)}) - A(S), \nabla f(A(S); \mathbf{z}_{i,j}) \big\rangle + \frac{L}{2}\big\|A(S^{(ij)}) - A(S)\big\|_2^2,$$

which further implies that for any $\gamma > 0$,

$$f(A(S^{(ij)}); \mathbf{z}_{i,j}) - f(A(S); \mathbf{z}_{i,j}) \leq \frac{1}{2\gamma}\big\|\nabla f(A(S); \mathbf{z}_{i,j})\big\|_2^2 + \frac{L+\gamma}{2}\big\|A(S^{(ij)}) - A(S)\big\|_2^2, \tag{A.7}$$

where we have used the inequality $\langle a, b \rangle \leq \frac{1}{2\gamma}\|a\|^2 + \frac{\gamma}{2}\|b\|^2$, $\forall a, b \in \mathbb{R}^d$. According to the self-bounding property of smooth function (Lemma A.4 with $\alpha = 1$), we have

$$\big\|\nabla f(A(S); \mathbf{z}_{i,j})\big\|_2^2 \leq 2L f(A(S); \mathbf{z}_{i,j}). \tag{A.8}$$

Combining Eq. (A.6), Eq. (A.7) and Eq. (A.8), we derive

$$\mathbb{E}_{S,I}\big[F(A(S)) - F_{S_j}(A(S))\big] \leq \frac{L}{\gamma n}\sum_{i=1}^n \mathbb{E}_{S,I}[f(A(S); \mathbf{z}_{i,j})] + \frac{L+\gamma}{2n}\sum_{t=1}^n \mathbb{E}_{S,\tilde{S},I}\big[\big\|A(S^{(ij)}) - A(S)\big\|_2^2\big]$$

$$= \frac{L}{\gamma}\mathbb{E}_{S,I}[F_{S_j}(A(S))] + \frac{L+\gamma}{2n}\sum_{i=1}^n \mathbb{E}_{S,\tilde{S},I}[\|A(S^{(ij)}) - A(S)\|_2^2].$$

The proof is complete. $\qquad\square$

## B. Generalization in Convex and Smooth Case

In this section, we give the proofs for the results in Section 4. We first establish a lemma to control the accumulation of error terms due to delays.

**Lemma B.1** (Lemma B.3 in Islamov et al. (2024)). *Let a stochastic sequence $\{\tau_t\}$ be produced by Algorithm 1. Then, for any non-negative sequence $\{q_t\}$, we have*

$$\sum_{t=1}^T \sum_{l=1}^{\tau_t} q_{t-l} \leq (m-1) \sum_{t=1}^T q_t.$$

*Proof.* We rewrite the summation as

$$\sum_{t=1}^T \sum_{l=1}^{\tau_t} q_{t-l} = \sum_{t=1}^T s_t q_t,$$

where $s_{t'}$ denotes the number of occurrences of $q_{t'}$ in the summation $\sum_{t=1}^{T} \sum_{l=1}^{\tau_t} q_{t-l}$ for any $t' \in [T]$. Since $\{\tau_t\}$ is produced by Algorithm 1, a term $q_{t'}$ appears in $\sum_{l=1}^{\tau_t} q_{t-l}$ if and only if the gradient computation job indexed by $t$ is assigned before time $t'$ and completed after time $t'$, i.e., when $t' \in \{t - \tau_t, \ldots, t - 1\}$. Note that the system consists of $m$ workers, and one of them just finished computation at step $t'$. Therefore, at step $t'$, at most $m - 1$ other workers are currently busy calculating gradients that started before $t'$ and will finish after $t'$. Consequently, for any $t' \in [T]$, we have $s_{t'} \leq m - 1$. The proof is complete. $\qquad\square$

### B.1. Proof of Theorem 4.1

*Proof.* **Proof sketch. The first step** is to derive a one-step recursion for the weight distance by considering two cases depending on the value of $S_{i_t, m_t}^{(ij)}$, corresponding to Eq. (B.1)–Eq. (B.5). **The second step** is to control the error accumulation caused by delayed updates using Lemma B.1, as shown in Eq. (B.6)–Eq.(B.12). **The final step** is to close the recursion and derives the stability bound in terms of the optimization error and worker participation count $N_T^M$.

**First,** we expand the one-step difference recursion. For any given time step $t \in [T]$, index $i \in [n]$ and $j \in [m]$ and sequence of worker arrivals $M = \{m_t\}$, recall the delayed gradient updates of ASGD in the form of (Algorithm 1),

$$\|\mathbf{w}_{t+1} - \mathbf{w}_{t+1}^{(ij)}\|^2 = \left\| \mathbf{w}_t - \eta_t \nabla f(\mathbf{w}_{t-\tau_t}; S_{i_t, m_t}) - \left( \mathbf{w}_t^{(ij)} - \eta_t \nabla f(\mathbf{w}_{t-\tau_t}^{(ij)}; S_{i_t, m_t}^{(ij)}) \right) \right\|^2$$
$$= \|\mathbf{w}_t - \mathbf{w}_t^{(ij)}\|^2 + \eta_t^2 \|\nabla f(\mathbf{w}_{t-\tau_t}; S_{i_t, m_t}) - \nabla f(\mathbf{w}_{t-\tau_t}^{(ij)}; S_{i_t, m_t}^{(ij)})\|^2$$
$$- 2\eta_t \langle \mathbf{w}_t - \mathbf{w}_t^{(ij)}, \nabla f(\mathbf{w}_{t-\tau_t}; S_{i_t, m_t}) - \nabla f(\mathbf{w}_{t-\tau_t}^{(ij)}; S_{i_t, m_t}^{(ij)}) \rangle, \qquad (B.1)$$

where $S_{i_t, m_t}^{(ij)}$ is defined in Eq. (A.1). For the cross term in Eq. (B.1), we can decompose it as follows:

$$- 2\eta_t \langle \mathbf{w}_t - \mathbf{w}_t^{(ij)}, \nabla f(\mathbf{w}_{t-\tau_t}; S_{i_t, m_t}) - \nabla f(\mathbf{w}_{t-\tau_t}^{(ij)}; S_{i_t, m_t}^{(ij)}) \rangle$$
$$= -2\eta_t \langle \mathbf{w}_{t-\tau_t} - \mathbf{w}_{t-\tau_t}^{(ij)}, \nabla f(\mathbf{w}_{t-\tau_t}; S_{i_t, m_t}) - \nabla f(\mathbf{w}_{t-\tau_t}^{(ij)}; S_{i_t, m_t}^{(ij)}) \rangle$$
$$- 2\eta_t \langle \mathbf{w}_t - \mathbf{w}_{t-\tau_t} - (\mathbf{w}_t^{(ij)} - \mathbf{w}_{t-\tau_t}^{(ij)}), \nabla f(\mathbf{w}_{t-\tau_t}; S_{i_t, m_t}) - \nabla f(\mathbf{w}_{t-\tau_t}^{(ij)}; S_{i_t, m_t}^{(ij)}) \rangle. \qquad (B.2)$$

Since $S$ differs from $S^{(ij)}$ by one sample, different examples may be encountered when updating $\mathbf{w}_t$ and $\mathbf{w}_t^{(ij)}$, which leads to a binary discussion w.r.t. $S_{i_t, m_t}^{(ij)}$:

*Case 1:* We assume that either $i_t \neq i$ or $m_t \neq j$, leading to $S_{i_t, m_t} = S_{i_t, m_t}^{(ij)}$. In this case, noticing that $\mathbf{w} \mapsto f(\mathbf{w}; \mathbf{z})$ is convex and smooth, we can know from the coercivity of $f$ (Lemma A.5 with $\alpha = 1$) that

$$\langle \mathbf{w}_{t-\tau_t} - \mathbf{w}_{t-\tau_t}^{(ij)}, \nabla f(\mathbf{w}_{t-\tau_t}; S_{i_t, m_t}) - \nabla f(\mathbf{w}_{t-\tau_t}^{(ij)}; S_{i_t, m_t}) \rangle$$
$$\geq \frac{1}{L} \|\nabla f(\mathbf{w}_{t-\tau_t}; S_{i_t, m_t}) - \nabla f(\mathbf{w}_{t-\tau_t}^{(ij)}; S_{i_t, m_t})\|_2^2 := \frac{1}{L} \|\mathfrak{D}_{t,i,j}\|^2,$$

where the definition of $\mathfrak{D}_{t,i,j}$ can be founded in Table 4. Combining the above inequality with Eq. (B.1) and Eq. (B.2) yields

$$\|\mathbf{w}_{t+1} - \mathbf{w}_{t+1}^{(ij)}\|^2 \leq \|\mathbf{w}_t - \mathbf{w}_t^{(ij)}\|^2 + (\eta_t^2 - \frac{2\eta_t}{L})\|\mathfrak{D}_{t,i,j}\|^2 - 2\eta_t \langle \mathbf{w}_t - \mathbf{w}_{t-\tau_t} - (\mathbf{w}_t^{(ij)} - \mathbf{w}_{t-\tau_t}^{(ij)}), \mathfrak{D}_{t,i,j} \rangle. \qquad (B.3)$$

*Case 2:* We consider the case where $i_t = i$ and $m_t = j$. In this case, we have

$$\|\mathbf{w}_{t+1} - \mathbf{w}_{t+1}^{(ij)}\|^2 = \left\| \mathbf{w}_t - \eta_t \nabla f(\mathbf{w}_{t-\tau_t}; \mathbf{z}_{i,j}) - \left( \mathbf{w}_t^{(ij)} - \eta_t \nabla f(\mathbf{w}_{t-\tau_t}^{(ij)}; \tilde{\mathbf{z}}_{i,j}) \right) \right\|^2$$
$$\leq \|\mathbf{w}_t - \mathbf{w}_t^{(ij)}\|^2 + \eta_t^2 \|\mathfrak{C}_{t,i,j}\|^2 + 2\eta_t \|\mathbf{w}_t - \mathbf{w}_t^{(ij)}\| \|\mathfrak{C}_{t,i,j}\|, \qquad (B.4)$$

where we have used the Cauchy-Schwarz inequality, i.e. $|\langle a, b \rangle| \leq \|a\|_2 \|b\|_2$. The definition of $\mathfrak{C}_{t,i,j}$ is shown in Table 4. Combining Eq. (B.3) and Eq. (B.4), we obtain

$$\|\mathbf{w}_{t+1} - \mathbf{w}_{t+1}^{(ij)}\|^2 \leq \|\mathbf{w}_t - \mathbf{w}_t^{(ij)}\|^2 + \eta_t^2 \mathbb{I}_{[i_t=i, m_t=j]} \|\mathfrak{C}_{t,i,j}\|^2 + 2\eta_t \mathbb{I}_{[i_t=i, m_t=j]} \|\mathbf{w}_t - \mathbf{w}_t^{(ij)}\| \|\mathfrak{C}_{t,i,j}\|$$
$$+ (\eta_t^2 - \frac{2\eta_t}{L}) \mathbb{I}_{[i_t \neq i \text{ or } m_t \neq j]} \|\mathfrak{D}_{t,i,j}\|^2 - 2\eta_t \mathbb{I}_{[i_t \neq i \text{ or } m_t \neq j]} \langle \mathbf{w}_t - \mathbf{w}_{t-\tau_t} - (\mathbf{w}_t^{(ij)} - \mathbf{w}_{t-\tau_t}^{(ij)}), \mathfrak{D}_{t,i,j} \rangle,$$

where $\mathbb{I}_{[\cdot]}$ denotes the indicator function, i.e. returning $1$ if the argument holds and $0$ otherwise. Note that given $S, S^{(ij)}$ and $\{m_t\}$, $\mathbf{w}_t$ and $\mathbf{w}_t^{(ij)}$ are determined by $\{i_{t'}\}_{t'=1}^{t-1}$. Then, since the sequence $\{i_t\}$ is independent of each other, $\mathbf{w}_t, \mathbf{w}_t^{(ij)}, \mathbf{w}_{t-\tau_t}$ and $\mathbf{w}_{t-\tau_t}^{(ij)}$ are independent of $i_t$. Also, $i_t$ follows the uniform distribution over $[n]$, which implies that $\mathbb{E}_I[i_t = i] = \frac{1}{n}$. Taking expectation to both sides of the above inequality w.r.t. $I$ yields

$$
\mathbb{E}_I\big[\|\mathbf{w}_{t+1} - \mathbf{w}_{t+1}^{(ij)}\|^2\big] \leq \mathbb{E}_I\big[\|\mathbf{w}_t - \mathbf{w}_t^{(ij)}\|^2\big] + \frac{1}{n}\eta_t^2\mathbb{I}_{[m_t=j]}\mathbb{E}_I\big[\|\mathfrak{C}_{t,i,j}\|^2\big]
$$
$$
+ \frac{2}{n}\eta_t\mathbb{I}_{[m_t=j]}\mathbb{E}_I\big[\|\mathbf{w}_t - \mathbf{w}_t^{(ij)}\|\|\mathfrak{C}_{t,i,j}\|\big] + (\eta_t^2 - \frac{2\eta_t}{L})\mathbb{E}_I\big[\mathbb{I}_{[i_t\neq i \text{ or } m_t\neq j]}\|\mathfrak{D}_{t,i,j}\|^2\big]
$$
$$
\underbrace{-2\eta_t\mathbb{E}_I\Big[\mathbb{I}_{[i_t\neq i \text{ or } m_t\neq j]}\big\langle \mathbf{w}_t - \mathbf{w}_{t-\tau_t} - (\mathbf{w}_t^{(ij)} - \mathbf{w}_{t-\tau_t}^{(ij)}), \mathfrak{D}_{t,i,j}\big\rangle\Big]}_{:=e_t}.
$$
(B.5)

**The second step** is to control the accumulated delay effect contributed by the term $e_t$ in the right-hand side (RHS) of Eq. (B.5). According to the update formula of Algorithm 1, sum of $e_t$ w.r.t. $t$ can written as

$$
\sum_{t=1}^{T} e_t = -2\sum_{t=1}^{T}\sum_{l=1}^{\tau_t}\eta_t\eta_{t-l}\mathbb{E}_I\Big[\mathbb{I}_{[i_t\neq i \text{ or } m_t\neq j]}\big\langle \nabla f\big(\mathbf{w}_{t-l-\tau_{t-l}}; S_{i_{t-l},m_{t-l}}\big) - \nabla f\big(\mathbf{w}_{t-l-\tau_{t-l}}^{(ij)}; S_{i_{t-l},m_{t-l}}^{(ij)}\big), \mathfrak{D}_{t,i,j}\big\rangle\Big].
$$
(B.6)

Analogous to the first step, a binary discussion for $S_{i_{t-l},m_{t-l}}^{(ij)}$ is needed, where $l \in \{1,\cdots,\tau_t\}$. First, consider the case where either $i_t \neq i$ or $m_t \neq j$. In this case, we have $S_{i_{t-l},m_{t-l}}^{(ij)} = S_{i_{t-l},m_{t-l}}$, leading to

$$
\big\langle \nabla f\big(\mathbf{w}_{t-l-\tau_{t-l}}; S_{i_{t-l},m_{t-l}}\big) - \nabla f\big(\mathbf{w}_{t-l-\tau_{t-l}}^{(ij)}; S_{i_{t-l},m_{t-l}}^{(ij)}\big), \mathfrak{D}_{t,i,j}\big\rangle = \big\langle \mathfrak{D}_{t-l,i,j}, \mathfrak{D}_{t,i,j}\big\rangle.
$$

Second, if $i_t = i$ and $m_t = j$, we have $S_{i_{t-l},m_{t-l}} = \mathbf{z}_{i,j}$ and $S_{i_{t-l},m_{t-l}}^{(ij)} = \tilde{\mathbf{z}}_{i,j}$. Then, it follows that

$$
\big\langle \nabla f\big(\mathbf{w}_{t-l-\tau_{t-l}}; S_{i_{t-l},m_{t-l}}\big) - \nabla f\big(\mathbf{w}_{t-l-\tau_{t-l}}^{(ij)}; S_{i_{t-l},m_{t-l}}^{(ij)}\big), \mathfrak{D}_{t,i,j}\big\rangle = \big\langle \mathfrak{C}_{t-l,i,j}, \mathfrak{D}_{t,i,j}\big\rangle.
$$

Combining the above two cases, the term $\sum_{t=1}^{T} e_t$ can be further decompose as follows:

$$
\sum_{t=1}^{T} e_t = -2\sum_{t=1}^{T}\sum_{l=1}^{\tau_t}\eta_t\eta_{t-l}\mathbb{E}_I\Big[\mathbb{I}_{[i_t\neq i \text{ or } m_t\neq j]}\mathbb{I}_{[i_{t-l}\neq i \text{ or } m_{t-l}\neq j]}\big\langle \mathfrak{D}_{t-l,i,j}, \mathfrak{D}_{t,i,j}\big\rangle\Big]
$$
$$
-2\sum_{t=1}^{T}\sum_{l=1}^{\tau_t}\eta_t\eta_{t-l}\mathbb{E}_I\Big[\mathbb{I}_{[i_t\neq i \text{ or } m_t\neq j]}\mathbb{I}_{[i_{t-l}=i \text{ and } m_{t-l}=j]}\big\langle \mathfrak{C}_{t-l,i,j}, \mathfrak{D}_{t,i,j}\big\rangle\Big].
$$
(B.7)

For the first term in the RHS of Eq. (B.7), using Young's inequality (Lemma A.2) with $p = q = 2$ yields

$$
-2\sum_{t=1}^{T}\sum_{l=1}^{\tau_t}\eta_t\eta_{t-l}\mathbb{E}_I\Big[\mathbb{I}_{[i_t\neq i \text{ or } m_t\neq j]}\mathbb{I}_{[i_{t-l}\neq i \text{ or } m_{t-l}\neq j]}\big\langle \mathfrak{D}_{t-l,i,j}, \mathfrak{D}_{t,i,j}\big\rangle\Big]
$$
$$
\leq \sum_{t=1}^{T}\sum_{l=1}^{\tau_t}\eta_t^2\mathbb{E}_I\big[\mathbb{I}_{[i_t\neq i \text{ or } m_t\neq j]}\|\mathfrak{D}_{t,i,j}\|^2\big] + \sum_{t=1}^{T}\sum_{l=1}^{\tau_t}\eta_{t-l}^2\mathbb{E}_I\big[\mathbb{I}_{[i_{t-l}\neq i \text{ or } m_{t-l}\neq j]}\|\mathfrak{D}_{t-l,i,j}\|^2\big].
$$

According to Lemma B.1, the second term in the RHS of the above inequality can be bounded as

$$
\sum_{t=1}^{T}\sum_{l=1}^{\tau_t}\eta_{t-l}^2\mathbb{E}_I\big[\mathbb{I}_{[i_{t-l}\neq i \text{ or } m_{t-l}\neq j]}\|\mathfrak{D}_{t-l,i,j}\|^2\big] \leq (m-1)\sum_{t=1}^{T}\eta_t^2\mathbb{E}_I\big[\mathbb{I}_{[i_t\neq i \text{ or } m_t\neq j]}\|\mathfrak{D}_{t,i,j}\|^2\big].
$$

It then follows that

$$-2\sum_{t=1}^{T}\sum_{l=1}^{\tau_t}\eta_t\eta_{t-l}\mathbb{E}_I\Big[\mathbb{I}_{[i_t\neq i\text{ or }m_t\neq j]}\mathbb{I}_{[i_{t-l}\neq i\text{ or }m_{t-l}\neq j]}\big\langle\mathfrak{D}_{t-l,i,j},\mathfrak{D}_{t,i,j}\big\rangle\Big]$$

$$\leq\sum_{t=1}^{T}\big(\tau_t+m-1\big)\eta_t^2\mathbb{E}_I\Big[\mathbb{I}_{[i_t\neq i\text{ or }m_t\neq j]}\|\mathfrak{D}_{t,i,j}\|^2\Big]. \quad\text{(B.8)}$$

For the second term in the RHS of Eq. (B.7), we have decomposition

$$-2\sum_{t=1}^{T}\sum_{l=1}^{\tau_t}\eta_t\eta_{t-l}\mathbb{E}_I\Big[\mathbb{I}_{[i_t\neq i\text{ or }m_t\neq j]}\mathbb{I}_{[i_{t-l}=i,m_{t-l}=j]}\big\langle\mathfrak{C}_{t-l,i,j},\mathfrak{D}_{t,i,j}\big\rangle\Big]$$

$$\leq\sum_{t=1}^{T}\sum_{l=1}^{\tau_t}\mathbb{E}_I\Big[\mathbb{I}_{[i_t\neq i\text{ or }m_t\neq j]}\mathbb{I}_{[i_{t-l}=i,m_{t-l}=j]}\big(\frac{\eta_{t-l}^2}{m}\|\mathfrak{C}_{t-l,i,j}\|^2+m\eta_t^2\|\mathfrak{D}_{t,i,j}\|^2\big)\Big], \quad\text{(B.9)}$$

where we have used the inequality $2ab\leq\frac{1}{m}a^2+mb^2,\forall m>0$. Note that, for $l\in\{1,\cdots,\tau_t\}$, $i_{t-l}$ is independent of both $\mathbf{w}_{t-l-\tau_{t-l}}$ and $\mathbf{w}_{t-\tau_t}$, and that $\mathbb{E}_I[i_{t-l}]=\frac{1}{n}$. Then, the left-hand side of Eq. (B.9) can be further bounded by

$$\frac{1}{nm}\sum_{t=1}^{T}\sum_{l=1}^{\tau_t}\eta_{t-l}^2\mathbb{I}_{[m_{t-l}=j]}\mathbb{E}_I\big[\|\mathfrak{C}_{t-l,i,j}\|^2\big]+\frac{m}{n}\sum_{t=1}^{T}\sum_{l=1}^{\tau_t}\eta_t^2\mathbb{I}_{[m_{t-l}=j]}\mathbb{E}_I\big[\mathbb{I}_{[i_t\neq i\text{ or }m_t\neq j]}\|\mathfrak{D}_{t,i,j}\|^2\big]. \quad\text{(B.10)}$$

Note that the first term in Eq. (B.10) can be bounded using Lemma B.1 and that $\sum_{l=1}^{\tau_t}\mathbb{I}_{[m_{t-l}=j]}\leq\tau_t$. Then, combining Eq. (B.9) and Eq. (B.10) yields

$$-2\sum_{t=1}^{T}\sum_{l=1}^{\tau_t}\eta_t\eta_{t-l}\mathbb{E}_I\Big[\mathbb{I}_{[i_t\neq i\text{ or }m_t\neq j]}\mathbb{I}_{[i_{t-l}=i,m_{t-l}=j]}\big\langle\mathfrak{C}_{t-l,i,j},\mathfrak{D}_{t,i,j}\big\rangle\Big]$$

$$\leq\frac{1}{n}\sum_{t=1}^{T}\eta_t^2\mathbb{I}_{[m_t=j]}\mathbb{E}_I\big[\|\mathfrak{C}_{t,i,j}\|^2\big]+\frac{m}{n}\sum_{t=1}^{T}\tau_t\eta_t^2\mathbb{E}_I\big[\mathbb{I}_{[i_t\neq i\text{ or }m_t\neq j]}\|\mathfrak{D}_{t,i,j}\|^2\big]. \quad\text{(B.11)}$$

Plugging Eq. (B.8) and Eq. (B.11) into Eq. (B.7), we derive

$$\sum_{t=1}^{T}e_t:=-2\sum_{t=1}^{T}\eta_t\mathbb{E}_I\Big[\mathbb{I}_{[i_t\neq i\text{ or }m_t\neq j]}\big\langle\mathbf{w}_t-\mathbf{w}_{t-\tau_t}-(\mathbf{w}_t^{(ij)}-\mathbf{w}_{t-\tau_t}^{(ij)}),\mathfrak{D}_{t,i,j}\big\rangle\Big]$$

$$\leq\sum_{t=1}^{T}\big((1+\frac{m}{n})\tau_t+m-1\big)\eta_t^2\mathbb{E}_I\Big[\mathbb{I}_{[i_t\neq i\text{ or }m_t\neq j]}\|\mathfrak{D}_{t,i,j}\|^2\Big]+\frac{1}{n}\sum_{t=1}^{T}\eta_t^2\mathbb{I}_{[m_t=j]}\mathbb{E}_I\big[\|\mathfrak{C}_{t,i,j}\|^2\big]. \quad\text{(B.12)}$$

**The third step is** to close the recursion and derive the stability bound. Taking a summation over $t$ to the both sides of Eq. (B.5) yields (note $\mathbf{w}_1=\mathbf{w}_1^{(i)}$)

$$\mathbb{E}_I\big[\|\mathbf{w}_{T+1}-\mathbf{w}_{T+1}^{(ij)}\|^2\big]\leq\frac{1}{n}\sum_{t=1}^{T}\eta_t^2\mathbb{I}_{[m_t=j]}\mathbb{E}_I\big[\|\mathfrak{C}_{t,i,j}\|^2\big]+\frac{2}{n}\sum_{t=1}^{T}\eta_t\mathbb{I}_{[m_t=j]}\mathbb{E}_I\big[\|\mathbf{w}_t-\mathbf{w}_t^{(ij)}\|\|\mathfrak{C}_{t,i,j}\|\big]$$

$$+\sum_{t=1}^{T}(\eta_t^2-\frac{2\eta_t}{L})\mathbb{E}_I\Big[\mathbb{I}_{[i_t\neq i\text{ or }m_t\neq j]}\|\mathfrak{D}_{t,i,j}\|^2\Big]-2\sum_{t=1}^{T}\eta_t\mathbb{E}_I\Big[\mathbb{I}_{[i_t\neq i\text{ or }m_t\neq j]}\big\langle\mathbf{w}_t-\mathbf{w}_{t-\tau_t}-(\mathbf{w}_t^{(ij)}-\mathbf{w}_{t-\tau_t}^{(ij)}),\mathfrak{D}_{t,i,j}\big\rangle\Big]. \quad\text{(B.13)}$$

Plugging Eq. (B.12) into Eq. (B.13), we obtain

$$\mathbb{E}_I[\|\mathbf{w}_{T+1}-\mathbf{w}_{T+1}^{(ij)}\|^2]\leq\frac{2}{n}\sum_{t=1}^{T}\eta_t^2\mathbb{I}_{[m_t=j]}\mathbb{E}_I\big[\|\mathfrak{C}_{t,i,j}\|^2\big]+\frac{2}{n}\sum_{t=1}^{T}\eta_t\mathbb{I}_{[m_t=j]}\mathbb{E}_I\big[\|\mathbf{w}_t-\mathbf{w}_t^{(ij)}\|\|\mathfrak{C}_{t,i,j}\|\big]$$

$$+\sum_{t=1}^{T}\Big(\big((1+\frac{m}{n})\tau_t+m\big)\eta_t-2L^{-1}\Big)\eta_t\mathbb{E}_I\Big[\mathbb{I}_{[i_t\neq i\text{ or }m_t\neq j]}\|\mathfrak{D}_{t,i,j}\|^2\Big].$$

Under the assumption that $\left((1 + \frac{m}{n})\tau_t + m\right)\eta_t \leq 2/L$, the above inequality further implies that

$$\mathbb{E}_I[\|\mathbf{w}_{T+1} - \mathbf{w}_{T+1}^{(ij)}\|^2] \leq \frac{2}{n}\sum_{t=1}^{T}\eta_t^2\mathbb{I}_{[m_t=j]}\mathbb{E}_I[\|\mathfrak{C}_{t,i,j}\|^2] + \frac{2}{n}\sum_{t=1}^{T}\eta_t\mathbb{I}_{[m_t=j]}\mathbb{E}_I[\|\mathbf{w}_t - \mathbf{w}_t^{(ij)}\|\|\mathfrak{C}_{t,i,j}\|].$$

Taking expectation w.r.t. $S, \tilde{S}$ to the both sides of the above inequality yields

$$\mathbb{E}_{S,\tilde{S},I}[\|\mathbf{w}_{T+1} - \mathbf{w}_{T+1}^{(ij)}\|^2] \leq \frac{2}{n}\sum_{t=1}^{T}\eta_t^2\mathbb{I}_{[m_t=j]}\mathbb{E}_{S,\tilde{S},I}[\|\mathfrak{C}_{t,i,j}\|^2]$$

$$+ \frac{2}{n}\sum_{t=1}^{T}\eta_t\mathbb{I}_{[m_t=j]}\left(\mathbb{E}_{S,\tilde{S},I}[\|\mathbf{w}_t - \mathbf{w}_t^{(ij)}\|^2]\right)^{\frac{1}{2}}\left(\mathbb{E}_{S,\tilde{S},I}[\|\mathfrak{C}_{t,i,j}\|^2]\right)^{\frac{1}{2}}, \quad \text{(B.14)}$$

where we have used the Cauchy-Schwarz inequality (Lemma A.3). Note that Eq. (B.14) holds for all $T \in \mathbb{N}^+$ and the RHS of Eq. (B.14) is a non-decreasing function of $T$. Therefore, we introduce $\widetilde{\Delta}_{t,i,j} = \max_{l\in[t]}\left(\mathbb{E}_{S,\tilde{S},I}\|\mathbf{w}_l - \mathbf{w}_l^{(ij)}\|^2\right)^{1/2}$ as shown in Table 4 and get

$$\widetilde{\Delta}_{T+1,i,j}^2 \leq \frac{2}{n}\sum_{t=1}^{T}\eta_t^2\mathbb{I}_{[m_t=j]}\mathbb{E}_{S,\tilde{S},I}[\|\mathfrak{C}_{t,i,j}\|^2] + \frac{2}{n}\sum_{t=1}^{T}\eta_t\mathbb{I}_{[m_t=j]}\widetilde{\Delta}_{T+1,i,j}\left(\mathbb{E}_{S,\tilde{S},I}[\|\mathfrak{C}_{t,i,j}\|^2]\right)^{\frac{1}{2}}. \quad \text{(B.15)}$$

Denote $N_T^M = \max_{j\in[m]}\sum_{t=1}^{T}\mathbb{I}[m_t = j]$ as given in Table 3. Using the Lemma A.1 with $x = \widetilde{\Delta}_{T+1,i,j}$, we know from Eq. (B.15) that

$$\widetilde{\Delta}_{T+1,i,j}^2 \leq \frac{4}{n}\sum_{t=1}^{T}\eta_t^2\mathbb{I}_{[m_t=j]}\mathbb{E}_{S,\tilde{S},I}[\|\mathfrak{C}_{t,i,j}\|^2] + \frac{4}{n^2}\left(\sum_{t=1}^{T}\eta_t\mathbb{I}_{[m_t=j]}\left(\mathbb{E}_{S,\tilde{S},I}[\|\mathfrak{C}_{t,i,j}\|^2]\right)^{\frac{1}{2}}\right)^2$$

$$\leq \frac{4}{n}\left(1 + \frac{N_T^M}{n}\right)\sum_{t=1}^{T}\eta_t^2\mathbb{I}_{[m_t=j]}\mathbb{E}_{S,\tilde{S},I}[\|\mathfrak{C}_{t,i,j}\|^2],$$

where the second inequality holds by the inequality $(\sum_{k=1}^{K}a_i)^2 \leq K\sum_{k=1}^{K}a_i^2, \forall K \in \mathbb{N}^+$ and the fact that the maximum number of non-zero elements in the summation term is $N_T^M$. Taking a summation for index $i$ from $1$ to $n$ to both sides of the above inequality, we derive

$$\frac{1}{n}\sum_{i=1}^{n}\widetilde{\Delta}_{T+1,i,j}^2 \leq \frac{4}{n}\left(1 + \frac{N_T^M}{n}\right)\sum_{t=1}^{T}\eta_t^2\mathbb{I}_{[m_t=j]}\left(\frac{1}{n}\sum_{i=1}^{n}\mathbb{E}_{S,\tilde{S},I}[\|\mathfrak{C}_{t,i,j}\|^2]\right). \quad \text{(B.16)}$$

Finally, the term in Eq. (B.16) in terms of $\mathfrak{C}_{t,i,j}$ can be bounded as follows:

$$\frac{1}{n}\sum_{i=1}^{n}\eta_t^2\mathbb{I}_{[m_t=j]}\mathbb{E}_{S,\tilde{S},I}[\|\mathfrak{C}_{t,i,j}\|^2]$$

$$\leq \frac{2}{n}\sum_{i=1}^{n}\eta_t^2\mathbb{I}_{[m_t=j]}\left(\mathbb{E}_{S,\tilde{S},I}[\|\nabla f(\mathbf{w}_{t-\tau_t};\mathbf{z}_{i,j})\|^2] + \mathbb{E}_{S,\tilde{S},I}[\|\nabla f(\mathbf{w}_{t-\tau_t}^{(ij)};\tilde{\mathbf{z}}_{i,j})\|^2]\right)$$

$$\leq \frac{4L}{n}\sum_{i=1}^{n}\eta_t^2\mathbb{I}_{[m_t=j]}\left(\mathbb{E}_{S,\tilde{S},I}[f(\mathbf{w}_{t-\tau_t};\mathbf{z}_{i,j})] + \mathbb{E}_{S,\tilde{S},I}[f(\mathbf{w}_{t-\tau_t}^{(ij)};\tilde{\mathbf{z}}_{i,j})]\right)$$

$$= 8L\eta_t^2\mathbb{I}_{[m_t=j]}\mathbb{E}_{S,I}[F_{S_j}(\mathbf{w}_{t-\tau_t})] = 8L\eta_t^2\mathbb{I}_{[m_t=j]}\mathbb{E}_{S,I}[F_{S_{m_t}}(\mathbf{w}_{t-\tau_t})], \quad \text{(B.17)}$$

where the first inequality holds by inequality $\|a + b\|^2 \leq 2\|a\|^2 + 2\|b\|^2$, the second inequality holds due to the self-bounding property of function $\mathbf{w} \mapsto f(\mathbf{w};\mathbf{z}_{i,j})$ (Lemma A.4 with $\alpha = 1$), and the third equality holds by the identity $\mathbb{E}_S[f(\mathbf{w}_{t-\tau_t};\mathbf{z}_{i,j})] = \mathbb{E}_{\tilde{S}}[f(\mathbf{w}_{t-\tau_t}^{(ij)};\tilde{\mathbf{z}}_{i,j})]$. Plugging Eq. (B.17) into Eq. (B.16), we derive the stability bound:

$$\epsilon_{stab,j}^2 \leq \frac{32L}{n}\left(1 + \frac{N_T^M}{n}\right)\sum_{t=1}^{T}\eta_t^2\mathbb{I}_{[m_t=j]}\mathbb{E}_{S,I}[F_{S_{m_t}}(\mathbf{w}_{t-\tau_t})].$$

The proof is complete. $\qquad\square$

**B.2. Proof of Lemma 4.4**

*Proof.* According to the update formula of Algorithm 1, we have

$$
\begin{aligned}
\|\mathbf{w}_{t+1} - \mathbf{w}^*\|^2 &= \|\mathbf{w}_t - \eta_t \nabla f(\mathbf{w}_{t-\tau_t}; S_{i_t, m_t}) - \mathbf{w}^*\|^2 \\
&= \|\mathbf{w}_t - \mathbf{w}^*\|^2 + \eta_t^2 \|\nabla f(\mathbf{w}_{t-\tau_t}; S_{i_t, m_t})\|^2 - 2\eta_t \langle \mathbf{w}_t - \mathbf{w}^*, \nabla f(\mathbf{w}_{t-\tau_t}; S_{i_t, m_t}) \rangle,
\end{aligned} \tag{B.18}
$$

where $\mathbf{w}^* = \arg\min_{\mathbf{w} \in \mathcal{W}} F(\mathbf{w})$. We can decompose the last term in the RHS of Eq. (B.18) as follows:

$$
\langle \mathbf{w}_t - \mathbf{w}^*, \nabla f(\mathbf{w}_{t-\tau_t}; S_{i_t, m_t}) \rangle = \langle \mathbf{w}_{t-\tau_t} - \mathbf{w}^*, \nabla f(\mathbf{w}_{t-\tau_t}; S_{i_t, m_t}) \rangle + \langle \mathbf{w}_t - \mathbf{w}_{t-\tau_t}, \nabla f(\mathbf{w}_{t-\tau_t}; S_{i_t, m_t}) \rangle. \tag{B.19}
$$

**For the first term** in the RHS of Eq. (B.19), taking expectation w.r.t. $S, I$, we derive

$$
\mathbb{E}_{S,I}\Big[ \langle \mathbf{w}_{t-\tau_t} - \mathbf{w}^*, \nabla f(\mathbf{w}_{t-\tau_t}; S_{i_t, m_t}) \rangle \Big] = \mathbb{E}_{S,I}\Big[ \langle \mathbf{w}_{t-\tau_t} - \mathbf{w}^*, \nabla F_{S_{m_t}}(\mathbf{w}_{t-\tau_t}) \rangle \Big],
$$

where the equality holds due to $i_t$ being independent of $\mathbf{w}_{t-\tau_t}$, and

$$
\mathbb{E}_{i_t}[\nabla f(\mathbf{w}_{t-\tau_t}; S_{i_t, m_t}) | \mathbf{w}_{t-\tau_t}] = \frac{1}{n}\sum_{i=1}^{n} \nabla f(\mathbf{w}_{t-\tau_t}; S_{i, m_t}) = \nabla F_{S_{m_t}}(\mathbf{w}_{t-\tau_t}).
$$

Since $\mathbf{w} \mapsto f(\mathbf{w}; \mathbf{z})$ is convex and $L$-smooth for any $\mathbf{z} \in \mathcal{Z}$, $\mathbf{w} \mapsto F_{S_j}(\mathbf{w})$ is also convex and $L$-smooth. Using Lemma A.6 with $\alpha = 1$, we obtain

$$
\mathbb{E}_{S,I}\Big[ \langle \mathbf{w}_{t-\tau_t} - \mathbf{w}^*, \nabla F_{S_{m_t}}(\mathbf{w}_{t-\tau_t}) \rangle \Big] \geq \mathbb{E}_{S,I}\Big[ F_{S_{m_t}}(\mathbf{w}_{t-\tau_t}) - F(\mathbf{w}^*) + \frac{1}{2L}\big\| \nabla F_{S_{m_t}}(\mathbf{w}_{t-\tau_t}) - \nabla F_{S_{m_t}}(\mathbf{w}^*) \big\|^2 \Big]. \tag{B.20}
$$

Similarly, applying Lemma A.6 followed by an expectation w.r.t $S, I$, we have

$$
\begin{aligned}
\mathbb{E}_{S,I}\Big[ \langle \mathbf{w}_{t-\tau_t} - \mathbf{w}^*, \nabla f(\mathbf{w}_{t-\tau_t}; S_{i_t, m_t}) \rangle \Big] & \\
&\geq \mathbb{E}_{S,I}\Big[ F_{S_{m_t}}(\mathbf{w}_{t-\tau_t}) - F(\mathbf{w}^*) + \frac{1}{2L}\big\| \nabla f(\mathbf{w}_{t-\tau_t}; S_{i_t, m_t}) - \nabla f(\mathbf{w}^*; S_{i_t, m_t}) \big\|^2 \Big], \quad \text{(B.21)}
\end{aligned}
$$

which will be lately used for the control of term $\eta_t^2 \|\nabla f(\mathbf{w}_{t-\tau_t}; S_{i_t, m_t})\|^2$ in Eq. (B.18).

**For the second term** in the RHS of Eq. (B.19), taking expectation w.r.t. $S, I$ and multiplying both sides by $\eta_t$, we get

$$
\eta_t \mathbb{E}_{S,I}\Big[ \langle \mathbf{w}_t - \mathbf{w}_{t-\tau_t}, \nabla f(\mathbf{w}_{t-\tau_t}; S_{i_t, m_t}) \rangle \Big] = \eta_t \mathbb{E}_{S,I}\Big[ \langle \mathbf{w}_t - \mathbf{w}_{t-\tau_t}, \nabla F_{S_{m_t}}(\mathbf{w}_{t-\tau_t}) \rangle \Big],
$$

where we have used the independence between $i_t$ and both $\mathbf{w}_t$ and $\mathbf{w}_{t-\tau_t}$. It then follows from the update formula that

$$
\begin{aligned}
\eta_t \mathbb{E}_{S,I}\Big[ &\langle \mathbf{w}_t - \mathbf{w}_{t-\tau_t}, \nabla F_{S_{m_t}}(\mathbf{w}_{t-\tau_t}) \rangle \Big] \\
&= \sum_{l=1}^{\tau_t} \eta_t \eta_{t-l} \mathbb{E}_{S,I}\Big[ \langle \nabla f(\mathbf{w}_{t-l-\tau_{t-l}}; S_{i_{t-l}, m_{t-l}}), \nabla F_{S_{m_t}}(\mathbf{w}_{t-\tau_t}) \rangle \Big] \\
&= \sum_{l=1}^{\tau_t} \eta_t \eta_{t-l} \mathbb{E}_{S,I}\Big[ \langle \nabla F_{S_{m_{t-l}}}(\mathbf{w}_{t-l-\tau_{t-l}}), \nabla F_{S_{m_t}}(\mathbf{w}_{t-\tau_t}) \rangle \Big] \\
&\geq -\frac{1}{2}\sum_{l=1}^{\tau_t} \mathbb{E}_{S,I}\Big[ \eta_{t-l}^2 \big\| \nabla F_{S_{m_{t-l}}}(\mathbf{w}_{t-l-\tau_{t-l}}) \big\|^2 + \eta_t^2 \big\| \nabla F_{S_{m_t}}(\mathbf{w}_{t-\tau_t}) \big\|^2 \Big], \quad \text{(B.22)}
\end{aligned}
$$

where the second equality holds since $i_{t-l}$ is independent to $\{i_{t'}\}_{t'=1}^{t-\tau_t - 1}$ for $\tau_t > 0$ and therefore independent to $\mathbf{w}_{t-\tau_t}$ and $\mathbf{w}_{t-l-\tau_{t-l}}$ for any $l \in \{1, \cdots, \tau_t\}$. The last inequality holds by Young's inequality. Note that Eq. (B.22) naturally holds

when $\tau_t = 0$ since in this case, the equality $\mathbf{w}_t = \mathbf{w}_{t-\tau_t}$ holds. Taking a summation over $t$ to both sides of Eq. (B.22) yields

$$\sum_{t=1}^{T} \eta_t \mathbb{E}_{S,I}\Big[\big\langle \mathbf{w}_t - \mathbf{w}_{t-\tau_t}, \nabla F_{S_{m_t}}(\mathbf{w}_{t-\tau_t})\big\rangle\Big]$$

$$\geq -\frac{1}{2}\sum_{t=1}^{T}\sum_{l=1}^{\tau_t} \eta_{t-l}^2 \mathbb{E}_{S,I}\Big[\big\|\nabla F_{S_{m_{t-l}}}(\mathbf{w}_{t-l-\tau_{t-l}})\big\|^2\Big] - \frac{1}{2}\sum_{t=1}^{T}\sum_{l=1}^{\tau_t} \eta_t^2 \mathbb{E}_{S,I}\Big[\big\|\nabla F_{S_{m_t}}(\mathbf{w}_{t-\tau_t})\big\|^2\Big]$$

$$\geq -\frac{1}{2}\sum_{t=1}^{T}(\tau_t + m - 1)\eta_t^2 \mathbb{E}_{S,I}\Big[\big\|\nabla F_{S_{m_t}}(\mathbf{w}_{t-\tau_t})\big\|^2\Big]$$

$$\geq -\sum_{t=1}^{T}(\tau_t + m - 1)\eta_t^2 \mathbb{E}_{S,I}\Big[\big\|\nabla F_{S_{m_t}}(\mathbf{w}_{t-\tau_t}) - \nabla F_{S_{m_t}}(\mathbf{w}^*)\big\|^2 + \big\|\nabla F_{S_{m_t}}(\mathbf{w}^*)\big\|^2\Big], \tag{B.23}$$

where the second and third inequality holds by Lemma B.1 and inequality $\|a + b\|^2 \leq 2\|a\|^2 + 2\|b\|^2$, $\forall a, b \in \mathbb{R}^d$, respectively. Recalling the definition of $\mathbf{w}^*$, the following inequality holds

$$\mathbb{E}_S\Big[\big\|\nabla F_{S_{m_t}}(\mathbf{w}^*)\big\|^2\Big] = \mathbb{E}_S\Big[\big\langle \frac{1}{n}\sum_{i=1}^{n}\nabla f(\mathbf{w}^*; S_{i,m_t}), \frac{1}{n}\sum_{i'=1}^{n}\nabla f(\mathbf{w}^*; S_{i',m_t})\big\rangle\Big]$$

$$= \frac{1}{n^2}\sum_{i=1}^{n}\mathbb{E}_S\big[\|\nabla f(\mathbf{w}^*; S_{i,m_t})\|^2\big] \leq \frac{2LF(\mathbf{w}^*)}{n}, \tag{B.24}$$

where the second equality holds since each sample in $S$ is drawn independently from $\mathcal{D}$ and $\mathbb{E}_S[\nabla f(\mathbf{w}^*; S_{i,m_t})] = 0$, and the last inequality holds by Lemma A.4. Recall Eq. (B.19). Multiplying $\eta_t$ to both sides followed by a summation over $t$, we derive

$$2\sum_{t=1}^{T}\eta_t \mathbb{E}_{S,I}\Big[\big\langle \mathbf{w}_t - \mathbf{w}^*, \nabla f(\mathbf{w}_{t-\tau_t}; S_{i_t,m_t})\big\rangle\Big] = 2\sum_{t=1}^{T}\eta_t \mathbb{E}_{S,I}\Big[\big\langle \mathbf{w}_t - \mathbf{w}_{t-\tau_t}, \nabla F_{S_{m_t}}(\mathbf{w}_{t-\tau_t})\big\rangle\Big]$$

$$+ \sum_{t=1}^{T}\eta_t \mathbb{E}_{S,I}\Big[\big\langle \mathbf{w}_{t-\tau_t} - \mathbf{w}^*, \nabla F_{S_{m_t}}(\mathbf{w}_{t-\tau_t})\big\rangle\Big] + \sum_{t=1}^{T}\eta_t \mathbb{E}_{S,I}\Big[\big\langle \mathbf{w}_{t-\tau_t} - \mathbf{w}^*, \nabla f(\mathbf{w}_{t-\tau_t}; S_{i_t,m_t})\big\rangle\Big],$$

where we use

$$2\eta_t \mathbb{E}\big[\big\langle \mathbf{w}_{t-\tau_t} - \mathbf{w}^*, \nabla f(\mathbf{w}_{t-\tau_t}; S_{i_t,m_t})\big\rangle\big]$$
$$= \eta_t \mathbb{E}\big[\big\langle \mathbf{w}_{t-\tau_t} - \mathbf{w}^*, \nabla F_{S_{m_t}}(\mathbf{w}_{t-\tau_t})\big\rangle\big] + \eta_t \mathbb{E}\big[\big\langle \mathbf{w}_{t-\tau_t} - \mathbf{w}^*, \nabla f(\mathbf{w}_{t-\tau_t}; S_{i_t,m_t})\big\rangle\big].$$

Combining Eq. (B.20), Eq. (B.21), Eq. (B.23) and Eq. (B.24) with above equality, we obtain

$$2\sum_{t=1}^{T}\eta_t \mathbb{E}_{S,I}\Big[\big\langle \mathbf{w}_t - \mathbf{w}^*, \nabla f(\mathbf{w}_{t-\tau_t}; S_{i_t,m_t})\big\rangle\Big] \geq 2\sum_{t=1}^{T}\eta_t \mathbb{E}_{S,I}\Big[F_{S_{m_t}}(\mathbf{w}_{t-\tau_t}) - F(\mathbf{w}^*)\Big]$$

$$+ \sum_{t=1}^{T}\big(\frac{1}{2L} - 2(\tau_t + m - 1)\eta_t\big)\eta_t \mathbb{E}_{S,I}\Big[\big\|\nabla F_{S_{m_t}}(\mathbf{w}_{t-\tau_t}) - \nabla F_{S_{m_t}}(\mathbf{w}^*)\big\|^2\Big]$$

$$+ \sum_{t=1}^{T}\frac{1}{2L}\eta_t \mathbb{E}_{S,I}\Big[\big\|\nabla f(\mathbf{w}_{t-\tau_t}; S_{i_t,m_t}) - \nabla f(\mathbf{w}^*; S_{i_t,m_t})\big\|^2\Big] - \frac{4LF(\mathbf{w}^*)}{n}\sum_{t=1}^{T}(\tau_t + m - 1)\eta_t^2. \tag{B.25}$$

Under the assumption that $(\tau_t + m - 1)\eta_t \leq 1/(4L)$, the second term in the RHS of the above inequality is larger than or equal to zero and following inequality holds:

$$\frac{4LF(\mathbf{w}^*)}{n}\sum_{t=1}^{T}(\tau_t + m - 1)\eta_t^2 \leq \frac{F(\mathbf{w}^*)}{n}\sum_{t=1}^{T}\eta_t.$$

**We now return to Eq.** (B.18)**.** Taking expectation to the both sides followed by a summation over $t$ yields

$$2\sum_{t=1}^{T}\eta_t\mathbb{E}_{S,I}\Big[\big\langle \mathbf{w}_t - \mathbf{w}^*, \nabla f(\mathbf{w}_{t-\tau_t}; S_{i_t,m_t})\big\rangle\Big] \le \|\mathbf{w}_1 - \mathbf{w}^*\|^2 - \mathbb{E}_{S,I}\big[\|\mathbf{w}_{T+1} - \mathbf{w}^*\|^2\big]$$

$$+ \sum_{t=1}^{T}\eta_t^2\mathbb{E}_{S,I}\big[\|\nabla f(\mathbf{w}_{t-\tau_t}; S_{i_t,m_t})\|^2\big]. \quad \text{(B.26)}$$

For the last term in the RHS of above inequality, using the inequality $\|a+b\|^2 \le 2\|a\|^2 + 2\|b\|^2$, $\forall a, b \in \mathbb{R}^d$, we have

$$\sum_{t=1}^{T}\eta_t^2\mathbb{E}_{S,I}\big[\|\nabla f(\mathbf{w}_{t-\tau_t}; S_{i_t,m_t})\|^2\big] = \sum_{t=1}^{T}\eta_t^2\mathbb{E}_{S,I}\big[\|\nabla f(\mathbf{w}_{t-\tau_t}; S_{i_t,m_t}) - \nabla f(\mathbf{w}^*; S_{i_t,m_t}) + \nabla f(\mathbf{w}^*; S_{i_t,m_t})\|^2\big]$$

$$\le 2\sum_{t=1}^{T}\eta_t^2\mathbb{E}_{S,I}\big[\|\nabla f(\mathbf{w}_{t-\tau_t}; S_{i_t,m_t}) - \nabla f(\mathbf{w}^*; S_{i_t,m_t})\|^2\big] + 2\sum_{t=1}^{T}\eta_t^2\mathbb{E}_{S,I}\big[\|\nabla f(\mathbf{w}^*; S_{i_t,m_t})\|^2\big],$$

which further implies that

$$\sum_{t=1}^{T}\eta_t^2\mathbb{E}_{S,I}\big[\|\nabla f(\mathbf{w}_{t-\tau_t}; S_{i_t,m_t})\|^2\big] \le 2\sum_{t=1}^{T}\eta_t^2\mathbb{E}_{S,I}\big[\|\nabla f(\mathbf{w}_{t-\tau_t}; S_{i_t,m_t}) - \nabla f(\mathbf{w}^*; S_{i_t,m_t})\|^2\big] + 4L\sum_{t=1}^{T}\eta_t^2 F(\mathbf{w}^*),$$

$$\text{(B.27)}$$

where we have used the Lemma A.4, that is, $\mathbb{E}_{S,I}\big[\|\nabla f(\mathbf{w}^*; S_{i_t,m_t})\|^2\big] \le 2L\mathbb{E}_S\big[F_{S_{m_t}}(\mathbf{w}^*)\big] = 2LF(\mathbf{w}^*)$. Plugging Eq. (B.27) into Eq. (B.26) yields

$$2\sum_{t=1}^{T}\eta_t\mathbb{E}_{S,I}\Big[\big\langle \mathbf{w}_t - \mathbf{w}^*, \nabla f(\mathbf{w}_{t-\tau_t}; S_{i_t,m_t})\big\rangle\Big] \le \|\mathbf{w}_1 - \mathbf{w}^*\|^2 - \mathbb{E}_{S,I}\big[\|\mathbf{w}_{T+1} - \mathbf{w}^*\|^2\big]$$

$$+ 2\sum_{t=1}^{T}\eta_t^2\mathbb{E}_{S,I}\big[\|\nabla f(\mathbf{w}_{t-\tau_t}; S_{i_t,m_t}) - \nabla f(\mathbf{w}^*; S_{i_t,m_t})\|^2\big] + 4L\sum_{t=1}^{T}\eta_t^2 F(\mathbf{w}^*),$$

which, together with Eq. (B.25), implies that

$$2\sum_{t=1}^{T}\eta_t\mathbb{E}_{S,I}\Big[F_{S_{m_t}}(\mathbf{w}_{t-\tau_t}) - F(\mathbf{w}^*)\Big] \le \|\mathbf{w}_1 - \mathbf{w}^*\|^2 - \mathbb{E}_{S,I}\big[\|\mathbf{w}_{T+1} - \mathbf{w}^*\|^2\big] + \frac{F(\mathbf{w}^*)}{n}\sum_{t=1}^{T}\eta_t + 4L\sum_{t=1}^{T}\eta_t^2 F(\mathbf{w}^*)$$

$$+ \sum_{t=1}^{T}(2\eta_t - \frac{1}{2L})\eta_t\mathbb{E}_{S,I}\Big[\big\|\nabla f(\mathbf{w}_{t-\tau_t}; S_{i_t,m_t}) - \nabla f(\mathbf{w}^*; S_{i_t,m_t})\big\|^2\Big]. \quad \text{(B.28)}$$

Under the assumption that $\eta_t \le 1/(4L)$, above inequality implies Eq. (4.5). The proof is complete. $\qquad \square$

## B.3. Proof of Theorem 4.6

*Proof.* For any $t \in \{1, \cdots, T\}$, let $A(S) = \mathbf{w}_{t-\tau_t}$ and $j = m_t$. We know from Lemma 3.3 that for any $\gamma > 0$,

$$\mathbb{E}_{S,I}[F(\mathbf{w}_{t-\tau_t}) - F_{S_{m_t}}(\mathbf{w}_{t-\tau_t})] \le \frac{L+\gamma}{2n}\sum_{i=1}^{n}\mathbb{E}_{S,\tilde{S},I}[\|\mathbf{w}_{t-\tau_t}^{(i,m_t)} - \mathbf{w}_{t-\tau_t}\|_2^2] + \frac{L}{\gamma}\mathbb{E}_{S,I}[F_{S_{m_t}}(\mathbf{w}_{t-\tau_t})].$$

Multiplying both sides of the above inequality by $\eta_t$ and taking a summation over $t$ from 1 to $T$, we derive

$$\sum_{t=1}^{T}\eta_t\mathbb{E}_{S,I}[F(\mathbf{w}_{t-\tau_t}) - F_{S_{m_t}}(\mathbf{w}_{t-\tau_t})] \le \frac{L}{\gamma}\sum_{t=1}^{T}\eta_t\mathbb{E}_{S,I}[F_{S_{m_t}}(\mathbf{w}_{t-\tau_t})]$$

$$+ \sum_{t=1}^{T}\eta_t\Big(\frac{L+\gamma}{2n}\sum_{i=1}^{n}\mathbb{E}_{S,\tilde{S},I}[\|\mathbf{w}_{t-\tau_t}^{(i,m_t)} - \mathbf{w}_{t-\tau_t}\|_2^2]\Big). \quad \text{(B.29)}$$

Using Theorem 4.1, we know that $\mathbf{w}_{t-\tau_t}$ is on-average $\epsilon_{stab,m_t}$-model stable w.r.t. subset $S_{m_t}$ where

$$\epsilon_{stab,m_t}^2 := \frac{1}{n}\sum_{i=1}^{n}\mathbb{E}_{S,\tilde{S},I}[\|\mathbf{w}_{t-\tau_t}^{(i,m_t)} - \mathbf{w}_{t-\tau_t}\|_2^2] \leq \frac{32L}{n}\Big(1 + \frac{N_{t-\tau_t-1}^M}{n}\Big)\sum_{k=1}^{t-\tau_t-1}\eta_k^2\mathbb{I}_{[m_k=m_t]}\mathbb{E}_{S,I}\big[F_{S_{m_k}}(\mathbf{w}_{k-\tau_k})\big]. \quad \text{(B.30)}$$

Multiplying $\eta_t$ to the both sides of Eq. (B.30) followed by a summation over $t$ from 1 to $T$, we obtain

$$\sum_{t=1}^{T}\eta_t\Big(\frac{1}{n}\sum_{i=1}^{n}\mathbb{E}_{S,\tilde{S},I}[\|\mathbf{w}_{t-\tau_t}^{(i,m_t)} - \mathbf{w}_{t-\tau_t}\|_2^2]\Big)$$

$$\leq \sum_{t=1}^{T}\eta_t\Big(\frac{32L}{n}\Big(1 + \frac{N_{t-\tau_t-1}^M}{n}\Big)\sum_{k=1}^{t-\tau_t-1}\eta_k^2\mathbb{I}_{[m_k=m_t]}\mathbb{E}_{S,I}\big[F_{S_{m_k}}(\mathbf{w}_{k-\tau_k})\big]\Big)$$

$$\leq \frac{32L\eta_M^2}{n}\Big(1 + \frac{N_T^M}{n}\Big)\sum_{t=1}^{T}\sum_{k=1}^{t-\tau_t-1}\eta_k\mathbb{I}_{[m_k=m_t]}\mathbb{E}_{S,I}\big[F_{S_{m_k}}(\mathbf{w}_{k-\tau_k})\big]$$

$$\leq \frac{32L\eta_M^2}{n}\Big(1 + \frac{N_T^M}{n}\Big)\sum_{t=1}^{T}\sum_{k=1}^{T}\eta_k\mathbb{I}_{[m_k=m_t]}\mathbb{E}_{S,I}\big[F_{S_{m_k}}(\mathbf{w}_{k-\tau_k})\big]$$

$$= \frac{32L\eta_M^2}{n}\Big(1 + \frac{N_T^M}{n}\Big)\sum_{k=1}^{T}\Big(\sum_{t=1}^{T}\mathbb{I}_{[m_k=m_t]}\Big)\eta_k\mathbb{E}_{S,I}\big[F_{S_{m_k}}(\mathbf{w}_{k-\tau_k})\big]$$

$$\leq 32L\eta_M^2\frac{N_T^M}{n}\Big(1 + \frac{N_T^M}{n}\Big)\sum_{k=1}^{T}\eta_k\mathbb{E}_{S,I}\big[F_{S_{m_k}}(\mathbf{w}_{k-\tau_k})\big], \quad \text{(B.31)}$$

where the second inequality holds since $N_t^M$ is a non-decreasing function of $t$, the third inequality holds since $F_{S_j}(\cdot)$ is non-negative for any $j \in [m]$, and the last inequality holds by the definition of $N_T^M$. According to Eq. (4.5), the following inequalities holds:

$$\sum_{t=1}^{T}\eta_t\mathbb{E}_{S,I}\big[F_{S_{m_t}}(\mathbf{w}_{t-\tau_t})\big] \leq \frac{1}{2}\|\mathbf{w}_1 - \mathbf{w}^*\|^2 + 2\sum_{t=1}^{T}\eta_t F(\mathbf{w}^*), \quad \text{(B.32)}$$

where we have used $\frac{1}{2n} + 2L\eta_t \leq 1$. Plugging Eq. (B.31) and Eq. (B.32) into Eq. (B.29) yields

$$\sum_{t=1}^{T}\eta_t\mathbb{E}_{S,I}[F(\mathbf{w}_{t-\tau_t}) - F_{S_{m_t}}(\mathbf{w}_{t-\tau_t})] \leq \Big(\frac{L}{\gamma} + 16L(L+\gamma)\eta_M^2\frac{N_T^M}{n}(1 + \frac{N_T^M}{n})\Big)\Big(\frac{1}{2}\|\mathbf{w}_1 - \mathbf{w}^*\|^2 + 2\sum_{t=1}^{T}\eta_t F(\mathbf{w}^*)\Big). \quad \text{(B.33)}$$

Combining Eq. (B.33) with Eq. (4.5) obtained from Lemma 4.4, we derive

$$\sum_{t=1}^{T}\eta_t\mathbb{E}_{S,I}[F(\mathbf{w}_{t-\tau_t}) - F(\mathbf{w}^*)] \leq \frac{1}{2}\|\mathbf{w}_1 - \mathbf{w}^*\|^2 + \sum_{t=1}^{T}\big(\frac{1}{2n} + 2L\eta_t\big)\eta_t F(\mathbf{w}^*)$$

$$+ \Big(\frac{L}{\gamma} + 16L(L+\gamma)\eta_M^2\frac{N_T^M}{n}(1 + \frac{N_T^M}{n})\Big)\Big(\frac{1}{2}\|\mathbf{w}_1 - \mathbf{w}^*\|^2 + 2\sum_{t=1}^{T}\eta_t F(\mathbf{w}^*)\Big).$$

Rearranging the terms of the above inequality, we get

$$\sum_{t=1}^{T}\eta_t\mathbb{E}_{S,I}[F(\mathbf{w}_{t-\tau_t}) - F(\mathbf{w}^*)] \leq \Big(\frac{L+\gamma}{2\gamma} + 8L(L+\gamma)\frac{N_T^M}{n}(1 + \frac{N_T^M}{n})\eta_M^2\Big)\|\mathbf{w}_1 - \mathbf{w}^*\|^2$$

$$+ \sum_{t=1}^{T}\eta_t\Big(\frac{2L}{\gamma} + 32L(L+\gamma)\frac{N_T^M}{n}(1 + \frac{N_T^M}{n})\eta_M^2 + \frac{1}{2n} + 2L\eta_t\Big)F(\mathbf{w}^*). \quad \text{(B.34)}$$

By the convexity of $\mathbf{w} \mapsto F(\mathbf{w})$, we have

$$\mathbb{E}_{S,I}[F(A(S))] := \mathbb{E}_{S,I}\Big[F\Big(\frac{\sum_{t=1}^{T}\eta_t\mathbf{w}_{t-\tau_t}}{\sum_{t=1}^{T}\eta_t}\Big)\Big] \leq \Big(\frac{1}{\sum_{t=1}^{T}\eta_t}\Big)\sum_{t=1}^{T}\eta_t\mathbb{E}_{S,I}[F(\mathbf{w}_{t-\tau_t})]. \quad \text{(B.35)}$$

Combining Eq. (B.34) and Eq.(B.35), we obtain

$$\mathbb{E}_{S,I}\big[F(A(S))\big] - F(\mathbf{w}^*) \leq \frac{1}{\sum_{t=1}^{T} \eta_t}\Big(\frac{L+\gamma}{2\gamma} + 8L(L+\gamma)\frac{N_T^M}{n}\Big(1+\frac{N_T^M}{n}\Big)\eta_M^2\Big)\|\mathbf{w}_1 - \mathbf{w}^*\|^2$$

$$+ \frac{1}{\sum_{t=1}^{T} \eta_t}\sum_{t=1}^{T}\eta_t\Big(\frac{2L}{\gamma} + 32L(L+\gamma)\frac{N_T^M}{n}\Big(1+\frac{N_T^M}{n}\Big)\eta_M^2 + \frac{1}{2n} + 2L\eta_t\Big)F(\mathbf{w}^*).$$

The proof is complete. $\qquad\square$

### B.4. Proof of Corollary 4.8

*Proof.* **For the constant step size where** $\eta_t = \eta$. Assume that for any $t \in [T]$, $\tau_t \leq c\sqrt{T}$ where $c_1$ is an algorithm-independent constant. Note that we only consider the workers that have contributed to the model updates. Then, the slowest worker must participate in the update for the first time after at least $m-1$ iterations, that is, there exists $t' \in [T]$, s.t. $m-1 \leq \tau_{t'}$. Let $T = a_1 kn$, $\eta = a_2 T^{-\frac{1}{2}}$, where $a_1, a_2 > 0$. The following inequality holds by $m \leq n$:

$$\big((1+\frac{m}{n})\tau_t + m\big)\eta \leq (2c\sqrt{a_1 kn} + c\sqrt{a_1 kn} + 1)/(a_2\sqrt{kn}) \leq (3c\sqrt{a_1} + 1)/a_2,$$

which implies that suitable choice of $a_1, a_2$ satisfies $\big((1+\frac{m}{n})\tau_t + m\big)\eta_t \leq 1/(4L)$ and therefore satisfies the conditions in Theorem 4.6. Recall Eq. (4.6):

$$\mathbb{E}_{S,I}\big[F(A(S))\big] - F(\mathbf{w}^*) \leq \frac{1}{\sum_{t=1}^{T} \eta_t}\Big(\frac{L+\gamma}{2\gamma} + 8L(L+\gamma)\frac{N_T^M}{n}(1+\frac{N_T^M}{n})\eta_M^2\Big)\|\mathbf{w}_1 - \mathbf{w}^*\|^2$$

$$+ \frac{1}{\sum_{t=1}^{T} \eta_t}\sum_{t=1}^{T}\eta_t\Big(\frac{2L}{\gamma} + 32L(L+\gamma)\frac{N_T^M}{n}(1+\frac{N_T^M}{n})\eta_M^2 + \frac{1}{2n} + 2L\eta_t\Big)F(\mathbf{w}^*). \quad \text{(B.36)}$$

Letting $\gamma = \sqrt{kn}$, Eq. (B.36) implies that

$$\mathbb{E}_{S,I}\big[F(A(S))\big] - F(\mathbf{w}^*)$$
$$\lesssim \frac{1}{\sqrt{T}}\Big(1 + \sqrt{kn}\frac{T}{kn}(1+\frac{T}{kn})\frac{1}{T}\Big)\|\mathbf{w}_1 - \mathbf{w}^*\|^2 + \Big(\frac{1}{\sqrt{kn}} + \sqrt{kn}\frac{T}{kn}(1+\frac{T}{kn})\frac{1}{T} + \frac{1}{n} + \frac{1}{\sqrt{T}}\Big)F(\mathbf{w}^*)$$
$$\lesssim \frac{1}{\sqrt{kn}}\big(\|\mathbf{w}_1 - \mathbf{w}^*\|^2 + F(\mathbf{w}^*)\big).$$

**Consider the adaptive step size**. We first establish a relationship between the number of workers $m$ and the average delay $\bar{\tau} = \sum_{t=1}^{T}\tau_t/T$ analogous to the Remark 5 in Koloskova et al. (2022), that is, $\bar{\tau} \leq m$. The reason is shown as follows. For the sake of comprehension, we denote $\tau_t^j$, $\forall j \in [m]$, as the delays accumulated in worker $j$ at step $t$ and $\tau_1^j = 0$, $\forall j \in [m]$. Then, it naturally holds that $\tau_t^{m_t} = \tau_t$, $\tau_{t+1}^{m_t} = 0$ and $\tau_{t+1}^j = \tau_t^j + 1$, $\forall j \neq m_t$. Using these notations, we have

$$\sum_{j=1}^{m}\tau_{t+1}^j = \sum_{j=1}^{m}\tau_t^j + m - 1 - \tau_t,$$

which further implies that

$$\sum_{j=1}^{m}\tau_{T+1}^j = (m-1)T - \sum_{t=1}^{T}\tau_t.$$

Since $\sum_{j=1}^{m}\tau_T^j \geq 0$, we then get $m \geq \bar{\tau}$. Since for any $t \in [T]$, $\tau_t \geq 0$, then the number of gradients that have delay larger than the average delay $\bar{\tau}$ is smaller than half of all the gradients, i.e. $\sum_{t=1}^{T}\mathbb{I}_{[\tau_t \geq \bar{\tau}]} \leq T/2$, which further indicates that $\sum_{t=1}^{T}\mathbb{I}_{[\tau_t \geq m]} \leq T/2$ and $\sum_{t=1}^{T}\mathbb{I}_{[\tau_t \leq m]} > T/2$. We choose $T = a_1 mn$ where $a_1 > 0$ and $\eta_t$ as

$$\eta_t = \begin{cases} \min\{\frac{1}{\sqrt{T}}, \frac{1}{12mL}\}, & \text{if } \tau_t \leq m, \\ \min\{\frac{1}{\sqrt{T}}, \frac{1}{4L((1+\frac{m}{n})\tau_t + m)}\}, & \text{if } \tau_t > m. \end{cases}$$

In this case, we have $\sum_{t=1}^{T} \eta_t \geq \frac{T}{2} \min\{\frac{1}{\sqrt{T}}, \frac{1}{12mL}\}$. Under the assumption that $m \leq n$, choosing $a_1 \leq 1/(12L)^2$, we have $\frac{\sqrt{T}}{2} \leq \frac{T}{24mL}$, which further implies that $\sum_{t=1}^{T} \eta_t \geq \frac{\sqrt{T}}{2}$. Letting $\gamma = \sqrt{mn}(\frac{k}{m})$, we know from Eq. (B.36) that

$$\mathbb{E}_{S,I}\big[F(A(S))\big] - F(\mathbf{w}^*) \lesssim \frac{1}{\sqrt{mn}}\Big(1 + \sqrt{mn}(\frac{k}{m})\frac{mn}{kn}(1 + \frac{mn}{kn})\frac{1}{mn}\Big)\|\mathbf{w}_1 - \mathbf{w}^*\|^2$$
$$+ \Big(\frac{1}{\sqrt{mn}}(\frac{m}{k}) + \sqrt{mn}(\frac{k}{m})\frac{mn}{kn}(1 + \frac{mn}{kn})\frac{1}{mn} + \frac{1}{n} + \frac{1}{\sqrt{mn}}\Big)F(\mathbf{w}^*), \quad \text{(B.37)}$$

which further implies that

$$\mathbb{E}_{S,I}\big[F(A(S))\big] - F(\mathbf{w}^*) \lesssim \frac{1}{\sqrt{mn}}\|\mathbf{w}_1 - \mathbf{w}^*\|^2 + \frac{1}{\sqrt{mn}}\frac{m}{k}F(\mathbf{w}^*).$$

**We now assume that** $F(\mathbf{w}^*) \lesssim 1/(mn)$. Analogous to the discussion in the second part, taking $T \asymp m^2 n$ and $\eta_t = \min\{\frac{1}{12mL}, \frac{1}{4L((1+\frac{m}{n})\tau_t + m)}\}$, we have $\sum_{t=1}^{T} \eta_t \geq \frac{T}{24mL}$. Under the assumption $N_T^M \asymp T/m$, letting $\gamma = 1$, Eq. (B.36) further indicates that

$$\mathbb{E}_{S,I}\big[F(A(S))\big] - F(\mathbf{w}^*) \lesssim \frac{1}{mn}\Big(1 + \frac{T}{mn}\Big(1 + \frac{T}{mn}\Big)\frac{1}{m^2}\Big)\|\mathbf{w}_1 - \mathbf{w}^*\|^2 + \Big(1 + \frac{T}{mn}\Big(1 + \frac{T}{mn}\Big)\frac{1}{m^2} + \frac{1}{n} + \frac{1}{m}\Big)\frac{1}{mn},$$

which implies that

$$\mathbb{E}_{S,I}\big[F(A(S))\big] - F(\mathbf{w}^*) \lesssim \frac{1}{mn}\big(\|\mathbf{w}_1 - \mathbf{w}^*\|^2 + 1\big).$$

The proof is complete. $\qquad\square$

## C. Generalization in Convex and Non-smooth Case

The following lemma (Lei & Ying, 2020) builds the relationship between stability and generalization error in the non-smooth case. Analogous to Lemma 3.3, we extend it to the distributed setting.

**Lemma C.1** (Generalization via Stability: Non-smooth Case). *Let $S, \tilde{S}$ and $S^{(ij)}$ be constructed as Definition 3.1. Let $A(S)$ be a randomized algorithm run on $S$. If for any $z$, the function $\mathbf{w} \mapsto f(\mathbf{w}; \mathbf{z})$ is nonnegative, convex, and $\mathbf{w} \mapsto \nabla f(\mathbf{w}; \mathbf{z})$ is $(\alpha, L)$-Hölder continuous, then for any $\gamma > 0$ and $j \in \{1, \ldots, m\}$, the following inequality holds*

$$\mathbb{E}_{S,I}[F(A(S)) - F_{S_j}(A(S))] \leq \frac{c_{\alpha,L}^2}{2\gamma}\mathbb{E}_{S,I}[F^{\frac{2\alpha}{1+\alpha}}(A(S))] + \frac{\gamma}{2n}\sum_{i=1}^{n}\mathbb{E}_{S,S',I}[\|A(S^{(ij)}) - A(S)\|_2^2], \quad \text{(C.1)}$$

*where $c_{\alpha,L}$ is defined in Eq. (5.2).*

### C.1. Proof of Theorem 5.1

*Proof.* We follow the same proof sketch of Theorem 4.1. **First,** we expand the one-step difference recursion. For any given index $i \in [n]$, we know from Eq. (B.1) that

$$\|\mathbf{w}_{t+1} - \mathbf{w}_{t+1}^{(ij)}\|^2 = \|\mathbf{w}_t - \mathbf{w}_t^{(ij)}\|^2 + \eta_t^2\|\nabla f(\mathbf{w}_{t-\tau_t}; S_{i_t,m_t}) - \nabla f(\mathbf{w}_{t-\tau_t}^{(ij)}; S_{i_t,m_t}^{(ij)})\|^2$$
$$- 2\eta_t\big\langle \mathbf{w}_t - \mathbf{w}_t^{(ij)}, \nabla f(\mathbf{w}_{t-\tau_t}; S_{i_t,m_t}) - \nabla f(\mathbf{w}_{t-\tau_t}^{(ij)}; S_{i_t,m_t}^{(ij)})\big\rangle. \quad \text{(C.2)}$$

Next, we have a binary discussion w.r.t. $S_{i_t,m_t}^{(ij)}$.

*Case 1:* We first consider the case where either $i_t = i$ and $m_t = j$, which means that $S_{i_t,m_t} = \mathbf{z}_{i,j}$ and $S_{i_t,m_t}^{(ij)} = \tilde{\mathbf{z}}_{i,j}$. Recall the definition of $\mathfrak{C}_{t,i,j}$ shown in Table 4. We have

$$- 2\eta_t\big\langle \mathbf{w}_t - \mathbf{w}_t^{(ij)}, \nabla f(\mathbf{w}_{t-\tau_t}; S_{i_t,m_t}) - \nabla f(\mathbf{w}_{t-\tau_t}^{(ij)}; S_{i_t,m_t}^{(ij)})\big\rangle$$
$$= -2\eta_t\big\langle \mathbf{w}_t - \mathbf{w}_t^{(ij)}, \nabla f(\mathbf{w}_{t-\tau_t}; \tilde{\mathbf{z}}_{i,j}) - \nabla f(\mathbf{w}_{t-\tau_t}^{(ij)}; \tilde{\mathbf{z}}_{i,j})\big\rangle \leq 2\eta_t\|\mathbf{w}_t - \mathbf{w}_t^{(ij)}\|\|\mathfrak{C}_{t,i,j}\|. \quad \text{(C.3)}$$

*Case 2:* We then consider the case where either $i_t \neq i$ or $m_t \neq j$, which means that $S_{i_t, m_t} = S_{i_t, m_t}^{(ij)}$. Analogous to Eq. (B.2), the cross term in Eq. (C.2) can be decomposed as:

$$
\begin{aligned}
&- 2\eta_t \big\langle \mathbf{w}_t - \mathbf{w}_t^{(ij)}, \nabla f(\mathbf{w}_{t-\tau_t}; S_{i_t, m_t}) - \nabla f(\mathbf{w}_{t-\tau_t}^{(ij)}; S_{i_t, m_t}^{(ij)}) \big\rangle \\
&= -2\eta_t \underbrace{\big\langle \mathbf{w}_{t-\tau_t} - \mathbf{w}_{t-\tau_t}^{(ij)}, \mathfrak{D}_{t,i,j} \big\rangle}_{:=\mathrm{I}} - 2\eta_t \underbrace{\big\langle \mathbf{w}_t - \mathbf{w}_{t-\tau_t} - (\mathbf{w}_t^{(ij)} - \mathbf{w}_{t-\tau_t}^{(ij)}), \mathfrak{D}_{t,i,j} \big\rangle}_{:=\mathrm{II}}.
\end{aligned} \tag{C.4}
$$

**Second**, we bounded the term I and II separately. For term I in Eq. (C.4), noticing that $\mathbf{w} \mapsto f(\mathbf{w}; \mathbf{z})$ is convex and its gradient is Hölder continuous, we can know from Lemma A.5 that

$$
\begin{aligned}
\mathrm{I} := \big\langle \mathbf{w}_{t-\tau_t} - \mathbf{w}_{t-\tau_t}^{(ij)}, \nabla f(\mathbf{w}_{t-\tau_t}; S_{i_t, m_t}) - \nabla f(\mathbf{w}_{t-\tau_t}^{(ij)}; S_{i_t, m_t}) \big\rangle \\
\geq \frac{2\alpha L^{-\frac{1}{\alpha}}}{1+\alpha} \big\| \nabla f(\mathbf{w}_{t-\tau_t}; S_{i_t, m_t}) - \nabla f(\mathbf{w}_{t-\tau_t}^{(ij)}; S_{i_t, m_t}) \big\|_2^{\frac{1+\alpha}{\alpha}} = \frac{2\alpha L^{-\frac{1}{\alpha}}}{1+\alpha} \big\| \mathfrak{D}_{t,i,j} \big\|_2^{\frac{1+\alpha}{\alpha}},
\end{aligned}
$$

which further implies that for any $\beta_t > 0$,

$$
\begin{aligned}
\big\| \mathfrak{D}_{t,i,j} \big\|_2^2 &\leq \Big( \frac{1+\alpha}{\alpha} \big\langle \mathbf{w}_{t-\tau_t} - \mathbf{w}_{t-\tau_t}^{(ij)}, \mathfrak{D}_{t,i,j} \big\rangle \Big)^{\frac{2\alpha}{1+\alpha}} \Big( \frac{L^{\frac{1}{\alpha}}}{2} \Big)^{\frac{2\alpha}{1+\alpha}} \\
&= \Big( \frac{1+\alpha}{\alpha \beta_t \eta_t} \big\langle \mathbf{w}_{t-\tau_t} - \mathbf{w}_{t-\tau_t}^{(ij)}, \mathfrak{D}_{t,i,j} \big\rangle \Big)^{\frac{2\alpha}{1+\alpha}} \Big( \frac{L^{\frac{1}{\alpha}} \beta_t \eta_t}{2} \Big)^{\frac{2\alpha}{1+\alpha}}.
\end{aligned}
$$

Note that the term $\beta_t$ is introduced to finetune the coefficient and obtain an optimal bound later. Using Lemma A.2 with $p = \frac{1+\alpha}{2\alpha}$ and $q = \frac{1+\alpha}{1-\alpha}$, above inequality indicates that

$$
\big\| \mathfrak{D}_{t,i,j} \big\|_2^2 \leq \frac{2\alpha}{1+\alpha} \Big( \frac{1+\alpha}{\alpha \beta_t \eta_t} \big\langle \mathbf{w}_{t-\tau_t} - \mathbf{w}_{t-\tau_t}^{(ij)}, \mathfrak{D}_{t,i,j} \big\rangle \Big) + \frac{1-\alpha}{1+\alpha} \Big( \frac{L^{\frac{1}{\alpha}} \beta_t \eta_t}{2} \Big)^{\frac{2\alpha}{1-\alpha}}.
$$

Rearranging the terms in above inequality, we obtain

$$
\mathrm{I} \geq \frac{\beta_t \eta_t}{2} \Big( \big\| \mathfrak{D}_{t,i,j} \big\|_2^2 - \frac{1-\alpha}{1+\alpha} \Big( \frac{L^{\frac{1}{\alpha}} \beta_t \eta_t}{2} \Big)^{\frac{2\alpha}{1-\alpha}} \Big) = \frac{\beta_t \eta_t}{2} \big\| \mathfrak{D}_{t,i,j} \big\|_2^2 - \frac{1-\alpha}{1+\alpha} L^{\frac{2}{1-\alpha}} \Big( \frac{\beta_t \eta_t}{2} \Big)^{\frac{1+\alpha}{1-\alpha}}. \tag{C.5}
$$

For the term II in Eq. (C.4), we can further decompose it using the update formula of Algorithm 1 as follows:

$$
\begin{aligned}
\mathrm{II} &:= \big\langle \mathbf{w}_t - \mathbf{w}_{t-\tau_t} - (\mathbf{w}_t^{(ij)} - \mathbf{w}_{t-\tau_t}^{(ij)}), \mathfrak{D}_{t,i,j} \big\rangle \\
&= \sum_{l=1}^{\tau_t} \big\langle \nabla f(\mathbf{w}_{t-l-\tau_{t-l}}; S_{i_{t-l}, m_{t-l}}) - \nabla f(\mathbf{w}_{t-l-\tau_{t-l}}^{(ij)}; S_{i_{t-l}, m_{t-l}}^{(ij)}), \mathfrak{D}_{t,i,j} \big\rangle.
\end{aligned}
$$

Note that term II is part of $e_t$, which is defined in Table 4. Recalling Eq. (B.12), we have

$$
\begin{aligned}
\sum_{t=1}^{T} e_t &:= -2 \sum_{t=1}^{T} \eta_t \mathbb{E}_I \Big[ \mathbb{I}_{[i_t \neq i \text{ or } m_t \neq j]} \big\langle \mathbf{w}_t - \mathbf{w}_{t-\tau_t} - (\mathbf{w}_t^{(ij)} - \mathbf{w}_{t-\tau_t}^{(ij)}), \mathfrak{D}_{t,i,j} \big\rangle \Big] \\
&\leq \sum_{t=1}^{T} \Big( (1 + \frac{m}{n}) \tau_t + m - 1 \Big) \eta_t^2 \mathbb{E}_I \Big[ \mathbb{I}_{[i_t \neq i \text{ or } m_t \neq j]} \big\| \mathfrak{D}_{t,i,j} \big\|^2 \Big] + \frac{1}{n} \sum_{t=1}^{T} \eta_t^2 \mathbb{I}_{[m_t=j]} \mathbb{E}_I \Big[ \big\| \mathfrak{C}_{t,i,j} \big\|^2 \Big]. \tag{C.6}
\end{aligned}
$$

**The third step** is to close the recursion. We now return to Eq. (C.2). Combining the above two cases, we obtain

$$
\begin{aligned}
\big\| \mathbf{w}_{t+1} - \mathbf{w}_{t+1}^{(ij)} \big\|^2 &\leq \big\| \mathbf{w}_t - \mathbf{w}_t^{(ij)} \big\|^2 + \eta_t^2 \mathbb{I}_{[i_t=i, m_t=j]} \big\| \mathfrak{C}_{t,i,j} \big\|^2 + \eta_t^2 \mathbb{I}_{[i_t \neq i \text{ or } m_t \neq j]} \big\| \mathfrak{D}_{t,i,j} \big\|^2 \\
&\quad + 2\eta_t \mathbb{I}_{[i_t=i, m_t=j]} \big\| \mathbf{w}_t - \mathbf{w}_t^{(ij)} \big\| \big\| \mathfrak{C}_{t,i,j} \big\| - 2\eta_t \mathbb{I}_{[i_t \neq i \text{ or } m_t \neq j]} \big\langle \mathbf{w}_{t-\tau_t} - \mathbf{w}_{t-\tau_t}^{(ij)}, \mathfrak{D}_{t,i,j} \big\rangle \\
&\quad - 2\eta_t \mathbb{I}_{[i_t \neq i \text{ or } m_t \neq j]} \big\langle \mathbf{w}_t - \mathbf{w}_{t-\tau_t} - (\mathbf{w}_t^{(ij)} - \mathbf{w}_{t-\tau_t}^{(ij)}), \mathfrak{D}_{t,i,j} \big\rangle.
\end{aligned}
$$

Note that $i_t$ is independent to $\mathbf{w}_t$ and $\mathbf{w}_{t-\tau_t}$, and $\mathbb{E}_I[\mathbb{I}_{[i_t=i]}] = \frac{1}{n}$. Taking expectation w.r.t. $I$ to the both sides and plugging Eq. (C.5) into above inequality yields

$$
\begin{aligned}
&\mathbb{E}_I\big[\|\mathbf{w}_{t+1} - \mathbf{w}_{t+1}^{(ij)}\|^2\big] \\
&\leq \mathbb{E}_I\big[\|\mathbf{w}_t - \mathbf{w}_t^{(ij)}\|^2\big] + \frac{2}{n}\eta_t\mathbb{I}_{[m_t=j]}\mathbb{E}_I\big[\|\mathbf{w}_t - \mathbf{w}_t^{(ij)}\|\|\mathfrak{C}_{t,i,j}\|\big] + \frac{1}{n}\eta_t^2\mathbb{I}_{[m_t=j]}\mathbb{E}_I\big[\|\mathfrak{C}_{t,i,j}\|^2\big] \\
&\quad + (1-\beta_t)\eta_t^2\mathbb{E}_I\big[\mathbb{I}_{[i_t\neq i \text{ or } m_t\neq j]}\|\mathfrak{D}_{t,i,j}\|^2\big] + \frac{1-\alpha}{1+\alpha}\beta_t^{\frac{1+\alpha}{1-\alpha}}(2^{-\alpha}L\eta_t)^{\frac{2}{1-\alpha}} \\
&\quad - 2\eta_t\mathbb{E}_I\Big[\mathbb{I}_{[i_t\neq i \text{ or } m_t\neq j]}\big\langle\mathbf{w}_t - \mathbf{w}_{t-\tau_t} - (\mathbf{w}_t^{(ij)} - \mathbf{w}_{t-\tau_t}^{(ij)}), \mathfrak{D}_{t,i,j}\big\rangle\Big].
\end{aligned}
$$

Taking a summation over $t$ from $1$ to $T$ of the above inequality, we obtain

$$
\begin{aligned}
&\mathbb{E}_I\big[\|\mathbf{w}_{T+1} - \mathbf{w}_{T+1}^{(ij)}\|^2\big] \\
&\leq \frac{2}{n}\sum_{t=1}^{T}\eta_t\mathbb{I}_{[m_t=j]}\mathbb{E}_I\big[\|\mathbf{w}_t - \mathbf{w}_t^{(ij)}\|\|\mathfrak{C}_{t,i,j}\|\big] + \frac{1}{n}\sum_{t=1}^{T}\eta_t^2\mathbb{I}_{[m_t=j]}\mathbb{E}_I\big[\|\mathfrak{C}_{t,i,j}\|^2\big] \\
&\quad + \sum_{t=1}^{T}(1-\beta_t)\eta_t^2\mathbb{E}_I\big[\mathbb{I}_{[i_t\neq i \text{ or } m_t\neq j]}\|\mathfrak{D}_{t,i,j}\|^2\big] + \frac{1-\alpha}{1+\alpha}\sum_{t=1}^{T}\beta_t^{\frac{1+\alpha}{1-\alpha}}(2^{-\alpha}L\eta_t)^{\frac{2}{1-\alpha}} \\
&\quad - 2\sum_{t=1}^{T}\eta_t\mathbb{E}_I\Big[\mathbb{I}_{[i_t\neq i \text{ or } m_t\neq j]}\big\langle\mathbf{w}_t - \mathbf{w}_{t-\tau_t} - (\mathbf{w}_t^{(ij)} - \mathbf{w}_{t-\tau_t}^{(ij)}), \mathfrak{D}_{t,i,j}\big\rangle\Big].
\end{aligned}
$$

Plugging Eq. (C.6) into the above inequality, we derive

$$
\begin{aligned}
&\mathbb{E}_I\big[\|\mathbf{w}_{T+1} - \mathbf{w}_{T+1}^{(ij)}\|^2\big] \\
&\leq \frac{2}{n}\sum_{t=1}^{T}\eta_t\mathbb{I}_{[m_t=j]}\mathbb{E}_I\big[\|\mathbf{w}_t - \mathbf{w}_t^{(ij)}\|\|\mathfrak{C}_{t,i,j}\|\big] + \frac{2}{n}\sum_{t=1}^{T}\eta_t^2\mathbb{I}_{[m_t=j]}\mathbb{E}_I\big[\|\mathfrak{C}_{t,i,j}\|^2\big] \\
&\quad + \sum_{t=1}^{T}\big((1+\frac{m}{n})\tau_t + m - \beta_t\big)\eta_t^2\mathbb{E}_I\big[\mathbb{I}_{[i_t\neq i \text{ or } m_t\neq j]}\|\mathfrak{D}_{t,i,j}\|^2\big] + \frac{1-\alpha}{1+\alpha}\sum_{t=1}^{T}\beta_t^{\frac{1+\alpha}{1-\alpha}}(2^{-\alpha}L\eta_t)^{\frac{2}{1-\alpha}}. \quad\text{(C.7)}
\end{aligned}
$$

Letting $\beta_t = (1+\frac{m}{n})\tau_t + m$, Eq. (C.7) further implies that

$$
\begin{aligned}
\mathbb{E}_I\big[\|\mathbf{w}_{T+1} - \mathbf{w}_{T+1}^{(ij)}\|^2\big] &\leq \frac{2}{n}\sum_{t=1}^{T}\eta_t\mathbb{I}_{[m_t=j]}\mathbb{E}_I\big[\|\mathbf{w}_t - \mathbf{w}_t^{(ij)}\|\|\mathfrak{C}_{t,i,j}\|\big] + \frac{2}{n}\sum_{t=1}^{T}\eta_t^2\mathbb{I}_{[m_t=j]}\mathbb{E}_I\big[\|\mathfrak{C}_{t,i,j}\|^2\big] \\
&\quad + \frac{1-\alpha}{1+\alpha}\sum_{t=1}^{T}\big((1+\frac{m}{n})\tau_t + m\big)^{\frac{1+\alpha}{1-\alpha}}(2^{-\alpha}L\eta_t)^{\frac{2}{1-\alpha}}.
\end{aligned}
$$

Taking expectation w.r.t. $S, \tilde{S}$ to both sides of the above inequality yields

$$
\begin{aligned}
\mathbb{E}_{S,\tilde{S},I}[\|\mathbf{w}_{T+1} - \mathbf{w}_{T+1}^{(ij)}\|^2] &\leq \frac{2}{n}\sum_{t=1}^{T}\eta_t^2\mathbb{I}_{[m_t=j]}\mathbb{E}_{S,\tilde{S},I}\big[\|\mathfrak{C}_{t,i,j}\|^2\big] \\
&\quad + \frac{2}{n}\sum_{t=1}^{T}\eta_t\mathbb{I}_{[m_t=j]}\Big(\mathbb{E}_{S,\tilde{S},I}\big[\|\mathbf{w}_t - \mathbf{w}_t^{(ij)}\|^2\big]\Big)^{\frac{1}{2}}\Big(\mathbb{E}_{S,\tilde{S},I}\big[\|\mathfrak{C}_{t,i,j}\|^2\big]\Big)^{\frac{1}{2}} \\
&\quad + \frac{1-\alpha}{1+\alpha}\sum_{t=1}^{T}\big((1+\frac{m}{n})\tau_t + m\big)^{\frac{1+\alpha}{1-\alpha}}(2^{-\alpha}L\eta_t)^{\frac{2}{1-\alpha}}, \quad\text{(C.8)}
\end{aligned}
$$

where we have used the Cauchy-Schwarz inequality. Note that Eq. (C.8) holds for all $T \in \mathbb{N}^+$ and the RHS of Eq. (C.8) is a

non-decreasing function of $T$. Introducing the notion $\widetilde{\Delta}_{t,i,j}$ defined in Table 4, the above inequality further indicates that

$$\widetilde{\Delta}^2_{T+1,i,j} \leq \frac{2}{n} \sum_{t=1}^T \eta_t^2 \mathbb{I}_{[m_t=j]} \mathbb{E}_{S,\tilde{S},I}\big[\|\mathfrak{C}_{t,i,j}\|^2\big] + \frac{2}{n} \sum_{t=1}^T \eta_t \mathbb{I}_{[m_t=j]} \widetilde{\Delta}_{T+1,i,j} \Big(\mathbb{E}_{S,\tilde{S},I}\big[\|\mathfrak{C}_{t,i,j}\|^2\big]\Big)^{\frac{1}{2}}$$

$$+ \frac{1-\alpha}{1+\alpha} \sum_{t=1}^T \Big((1+\frac{m}{n})\tau_t + m\Big)^{\frac{1+\alpha}{1-\alpha}} (2^{-\alpha} L\eta_t)^{\frac{2}{1-\alpha}}. \quad \text{(C.9)}$$

Using the Lemma A.1 with $x = \widetilde{\Delta}_{T+1,i,j}$, we have

$$\widetilde{\Delta}^2_{T+1,i,j} \leq \frac{4}{n} \sum_{t=1}^T \eta_t^2 \mathbb{I}_{[m_t=j]} \mathbb{E}_{S,\tilde{S},I}\big[\|\mathfrak{C}_{t,i,j}\|^2\big] + \frac{4}{n^2} \Big(\sum_{t=1}^T \eta_t \mathbb{I}_{[m_t=j]} \big(\mathbb{E}_{S,\tilde{S},I}\big[\|\mathfrak{C}_{t,i,j}\|^2\big]\big)^{\frac{1}{2}}\Big)^2$$

$$+ \frac{2(1-\alpha)}{1+\alpha} \sum_{t=1}^T \Big((1+\frac{m}{n})\tau_t + m\Big)^{\frac{1+\alpha}{1-\alpha}} (2^{-\alpha} L\eta_t)^{\frac{2}{1-\alpha}}$$

$$\leq \frac{4}{n}\Big(1+\frac{N_T^M}{n}\Big) \sum_{t=1}^T \eta_t^2 \mathbb{I}_{[m_t=j]} \mathbb{E}_{S,\tilde{S},I}\big[\|\mathfrak{C}_{t,i,j}\|^2\big] + \frac{2(1-\alpha)}{1+\alpha} \sum_{t=1}^T \Big((1+\frac{m}{n})\tau_t + m\Big)^{\frac{1+\alpha}{1-\alpha}} (2^{-\alpha} L\eta_t)^{\frac{2}{1-\alpha}},$$

where the second inequality holds by Cauchy-Schwarz inequality and the fact that the maximum number of non-zero elements in the summation term is $N_T^M$. Taking a summation over index $i$ to both sides of the above inequality, we derive

$$\frac{1}{n} \sum_{i=1}^n \widetilde{\Delta}^2_{T+1,i,j} \leq \frac{4}{n}\Big(1+\frac{N_T^M}{n}\Big) \sum_{t=1}^T \eta_t^2 \mathbb{I}_{[m_t=j]} \Big(\frac{1}{n}\sum_{i=1}^n \mathbb{E}_{S,\tilde{S},I}\big[\|\mathfrak{C}_{t,i,j}\|^2\big]\Big)$$

$$+ \frac{2(1-\alpha)}{1+\alpha} \sum_{t=1}^T \Big((1+\frac{m}{n})\tau_t + m\Big)^{\frac{1+\alpha}{1-\alpha}} (2^{-\alpha} L\eta_t)^{\frac{2}{1-\alpha}}. \quad \text{(C.10)}$$

Analogous to Eq. (B.17), we use Lemma A.4 to control the term involving with $\mathfrak{C}_{t,i,j}$ as follows:

$$\frac{1}{n} \sum_{i=1}^n \mathbb{E}_{S,\tilde{S},I}\big[\|\mathfrak{C}_{t,i,j}\|^2\big] = \frac{1}{n} \sum_{i=1}^n \mathbb{E}_{S,\tilde{S},I}\Big[\big\|\nabla f(\mathbf{w}_{t-\tau_t}; \mathbf{z}_{i,j}) - \nabla f(\mathbf{w}^{(ij)}_{t-\tau_t}; \tilde{\mathbf{z}}_{i,j})\big\|^2\Big]$$

$$\leq \frac{1}{n} \sum_{i=1}^n \mathbb{E}_{S,\tilde{S},I}\Big[2\|\nabla f(\mathbf{w}_{t-\tau_t}; \mathbf{z}_{i,j})\|^2 + 2\|\nabla f(\mathbf{w}^{(ij)}_{t-\tau_t}; \tilde{\mathbf{z}}_{i,j})\|^2\Big]$$

$$\leq \frac{2c_{\alpha,L}^2}{n} \sum_{i=1}^n \mathbb{E}_{S,\tilde{S},I}\Big[f^{\frac{2\alpha}{1+\alpha}}(\mathbf{w}_{t-\tau_t}; \mathbf{z}_{i,j}) + f^{\frac{2\alpha}{1+\alpha}}(\mathbf{w}^{(ij)}_{t-\tau_t}; \tilde{\mathbf{z}}_{i,j})\Big]$$

$$\leq \frac{4c_{\alpha,L}^2}{n} \sum_{i=1}^n \mathbb{E}_{S,I}\big[f^{\frac{2\alpha}{1+\alpha}}(\mathbf{w}_{t-\tau_t}; \mathbf{z}_{i,j})\big] \leq 4c_{\alpha,L}^2 \mathbb{E}_{S,I}\big[F_{S_j}(\mathbf{w}_{t-\tau_t})\big]^{\frac{2\alpha}{1+\alpha}}, \quad \text{(C.11)}$$

where the second inequality holds by Cauchy-Schwarz inequality, the third inequality holds with Lemma A.4, the fourth inequality holds by the symmetry of $\mathbf{z}_{i,j}$ and $\tilde{\mathbf{z}}_{i,j}$, and the last inequality holds since the function $x \mapsto f^{\frac{2\alpha}{1+\alpha}}$ is concave and therefore it holds that

$$\frac{1}{n} \sum_{i=1}^n \mathbb{E}_{S,I}\big[f^{\frac{2\alpha}{1+\alpha}}(\mathbf{w}_{t-\tau_t}; \mathbf{z}_{i,j})\big] \leq \frac{1}{n} \sum_{i=1}^n \mathbb{E}_{S,I}\big[f(\mathbf{w}_{t-\tau_t}; \mathbf{z}_{i,j})\big]^{\frac{2\alpha}{1+\alpha}} \leq \Big(\frac{1}{n} \sum_{i=1}^n \mathbb{E}_{S,I}\big[f(\mathbf{w}_{t-\tau_t}; \mathbf{z}_{i,j})\big]\Big)^{\frac{2\alpha}{1+\alpha}}.$$

Plugging Eq. (C.11) into Eq. (C.10), we obtain the stability bound:

$$\epsilon_{stab,j}^2 \leq \frac{16c_{\alpha,L}^2}{n}\Big(1+\frac{N_T^M}{n}\Big) \sum_{t=1}^T \eta_t^2 \mathbb{I}_{[m_t=j]} \mathbb{E}_{S,I}\big[F_{S_{m_t}}(\mathbf{w}_{t-\tau_t})\big]^{\frac{2\alpha}{1+\alpha}} + \frac{2(1-\alpha)}{1+\alpha} \sum_{t=1}^T \Big((1+\frac{m}{n})\tau_t + m\Big)^{\frac{1+\alpha}{1-\alpha}} (2^{-\alpha} L\eta_t)^{\frac{2}{1-\alpha}}.$$

The proof is complete. $\qquad\square$

## C.2. Proof of Lemma 5.3

In this subsection, we first give the complete version of Lemma 5.3.

**Lemma C.2** (Formal version of Lemma 5.3). *Assume for any* $\mathbf{z}$, $\mathbf{w} \mapsto f(\mathbf{w}; \mathbf{z})$ *is non-negative convex and* $\mathbf{w} \mapsto \nabla f(\mathbf{w}; \mathbf{z})$ *is* $(\alpha, L)$-*Hölder continuous. Let* $\{\mathbf{w}_t\}$ *be produced by Algorithm 1. If the step size* $\{\eta_t\}$ *satisfies* $(\max\{\tau_t + m - 1, 2\})\eta_t \leq 1/(2L^{\frac{1}{\alpha}})$ *for any* $t \in [T]$ [2], *then we have*

$$2\sum_{t=1}^{T} \eta_t \mathbb{E}_{S,I}\Big[F_{S_{m_t}}(\mathbf{w}_{t-\tau_t}) - F(\mathbf{w}^*)\Big] \leq \|\mathbf{w}_1 - \mathbf{w}^*\|^2$$

$$+ 2c_{\alpha,1}\sum_{t=1}^{T}(\tau_t + m)\eta_t^2 + 2c_{\alpha,2}F^{\frac{2\alpha}{1+\alpha}}(\mathbf{w}^*)\Big(\frac{1}{n}\sum_{t=1}^{T}(\tau_t + m - 1)\eta_t^2 + \sum_{t=1}^{T}\eta_t^2\Big), \quad \text{(C.12)}$$

*where*

$$c_{\alpha,1} = \begin{cases} \frac{1-\alpha}{1+\alpha}, & \text{if } \alpha \in (0,1), \\ (L + \sup_{\mathbf{z} \in \mathcal{Z}} \|\nabla f(0; \mathbf{z})\|)^2, & \text{if } \alpha = 0, \end{cases} \quad \text{and} \quad c_{\alpha,2} = \begin{cases} c_{\alpha,L}^2, & \text{if } \alpha \in (0,1), \\ 0, & \text{if } \alpha = 0. \end{cases}$$

*Proof.* We first consider the case where $\alpha \in (0, 1]$. Note the decomposition given in Eq. (B.19):

$$\eta_t\langle\mathbf{w}_t - \mathbf{w}^*, \nabla f(\mathbf{w}_{t-\tau_t}; S_{i_t, m_t})\rangle = \eta_t\langle\mathbf{w}_{t-\tau_t} - \mathbf{w}^*, \nabla f(\mathbf{w}_{t-\tau_t}; S_{i_t, m_t})\rangle + \eta_t\langle\mathbf{w}_t - \mathbf{w}_{t-\tau_t}, \nabla f(\mathbf{w}_{t-\tau_t}; S_{i_t, m_t})\rangle.$$
$$\text{(C.13)}$$

Noting that function $\mathbf{w} \mapsto \nabla f(\mathbf{w}; \mathbf{z})$ is $(\alpha, L)$-Hölder continuous, the expectation w.r.t. $S, I$ of the first term in the RHS of the above inequality can be bounded as follows:

$$\eta_t\mathbb{E}_{S,I}\Big[\langle\mathbf{w}_{t-\tau_t} - \mathbf{w}^*, \nabla f(\mathbf{w}_{t-\tau_t}; S_{i_t, m_t})\rangle\Big] = \eta_t\mathbb{E}_{S,I}\Big[\langle\mathbf{w}_{t-\tau_t} - \mathbf{w}^*, \nabla F_{S_{m_t}}(\mathbf{w}_{t-\tau_t})\rangle\Big]$$

$$\geq \eta_t\mathbb{E}_{S,I}\Big[F_{S_{m_t}}(\mathbf{w}_{t-\tau_t}) - F(\mathbf{w}^*)\Big] + \frac{\alpha L^{-\frac{1}{\alpha}}}{1+\alpha}\eta_t\mathbb{E}_{S,I}\Big[\big\|\nabla F_{S_{m_t}}(\mathbf{w}_{t-\tau_t}) - \nabla F_{S_{m_t}}(\mathbf{w}^*)\big\|^{\frac{1+\alpha}{\alpha}}\Big], \quad \text{(C.14)}$$

where the first equality holds since $i_t$ is independent to $\mathbf{w}_{t-\tau_t}$ and follows the uniform distribution on $[n]$, and the second inequality holds by Lemma A.6. Similarly, we also have

$$\eta_t\mathbb{E}_{S,I}\Big[\langle\mathbf{w}_{t-\tau_t} - \mathbf{w}^*, \nabla f(\mathbf{w}_{t-\tau_t}; S_{i_t, m_t})\rangle\Big]$$

$$\geq \eta_t\mathbb{E}_{S,I}\Big[F_{S_{m_t}}(\mathbf{w}_{t-\tau_t}) - F(\mathbf{w}^*)\Big] + \frac{\alpha L^{-\frac{1}{\alpha}}}{1+\alpha}\eta_t\mathbb{E}_{S,I}\Big[\big\|\nabla f(\mathbf{w}_{t-\tau_t}; S_{i_t, m_t}) - \nabla f(\mathbf{w}^*; S_{i_t, m_t})\big\|^{\frac{1+\alpha}{\alpha}}\Big]. \quad \text{(C.15)}$$

Recalling Eq. (B.22) the second term in Eq. (C.13) can be controlled by the following inequality

$$\eta_t\mathbb{E}_{S,I}\Big[\langle\mathbf{w}_t - \mathbf{w}_{t-\tau_t}, \nabla f(\mathbf{w}_{t-\tau_t}; S_{i_t, m_t})\rangle\Big] \geq -\frac{1}{2}\sum_{l=1}^{\tau_t}\mathbb{E}_{S,I}\Big[\eta_{t-l}^2\big\|\nabla F_{S_{m_{t-l}}}(\mathbf{w}_{t-l-\tau_{t-l}})\big\|^2 + \eta_t^2\big\|\nabla F_{S_{m_t}}(\mathbf{w}_{t-\tau_t})\big\|^2\Big].$$
$$\text{(C.16)}$$

Analogous to Eq. (B.23), taking a summation over $t$ to the both sides of Eq. (C.16) yields

$$\sum_{t=1}^{T}\eta_t\mathbb{E}_{S,I}\Big[\langle\mathbf{w}_t - \mathbf{w}_{t-\tau_t}, \nabla f(\mathbf{w}_{t-\tau_t}; S_{i_t, m_t})\rangle\Big]$$

$$\geq -\sum_{t=1}^{T}(\tau_t + m - 1)\eta_t^2\mathbb{E}_{S,I}\Big[\big\|\nabla F_{S_{m_t}}(\mathbf{w}_{t-\tau_t}) - \nabla F_{S_{m_t}}(\mathbf{w}^*)\big\|^2 + \big\|\nabla F_{S_{m_t}}(\mathbf{w}^*)\big\|^2\Big]$$

$$\geq -\sum_{t=1}^{T}(\tau_t + m - 1)\eta_t^2\mathbb{E}_{S,I}\Big[\big\|\nabla F_{S_{m_t}}(\mathbf{w}_{t-\tau_t}) - \nabla F_{S_{m_t}}(\mathbf{w}^*)\big\|^2\Big] - \frac{c_{\alpha,L}^2F^{\frac{2\alpha}{1+\alpha}}(\mathbf{w}^*)}{n}\sum_{t=1}^{T}(\tau_t + m - 1)\eta_t^2, \quad \text{(C.17)}$$

---

[2]This assumption only holds for $\alpha > 0$, that is, when $\alpha = 0$, Lemma 5.3 holds for any non-negative sequence $\{\eta_t\}$.

where the second inequality holds since

$$
\mathbb{E}_S\left[\left\|\nabla F_{S_{m_t}}(\mathbf{w}^*)\right\|^2\right] = \mathbb{E}_S\left[\left\langle\frac{1}{n}\sum_{i=1}^{n}\nabla f(\mathbf{w}^*;S_{i,m_t}),\frac{1}{n}\sum_{i'=1}^{n}\nabla f(\mathbf{w}^*;S_{i',m_t})\right\rangle\right]
$$

$$
= \frac{1}{n^2}\sum_{i=1}^{n}\mathbb{E}_S\left[\|\nabla f(\mathbf{w}^*;S_{i,m_t})\|^2\right] \leq \frac{c_{\alpha,L}^2\mathbb{E}_S[f^{\frac{2\alpha}{1+\alpha}}(\mathbf{w}^*;S_{i,m_t})]}{n} \leq \frac{c_{\alpha,L}^2 F^{\frac{2\alpha}{1+\alpha}}(\mathbf{w}^*)}{n}.
$$

The last two inequalities hold by Lemma A.4 and the concavity of function $x \mapsto x^{\frac{2\alpha}{1+\alpha}}$, respectively. Further, according to Lemma A.2, we have

$$
(\tau_t + m - 1)\eta_t^2\left\|\nabla F_{S_{m_t}}(\mathbf{w}_{t-\tau_t}) - \nabla F_{S_{m_t}}(\mathbf{w}^*)\right\|^2
$$

$$
= \left(((\tau_t + m - 1)\eta_t^2)^{\frac{1-\alpha}{1+\alpha}}\right) \cdot \left(((\tau_t + m - 1)\eta_t^2\left\|\nabla F_{S_{m_t}}(\mathbf{w}_{t-\tau_t}) - \nabla F_{S_{m_t}}(\mathbf{w}^*)\right\|^{\frac{1+\alpha}{\alpha}}\right)^{\frac{2\alpha}{1+\alpha}}
$$

$$
\leq \frac{1-\alpha}{1+\alpha}(\tau_t + m - 1)\eta_t^2 + \frac{2\alpha}{1+\alpha}(\tau_t + m - 1)\eta_t^2\left\|\nabla F_{S_{m_t}}(\mathbf{w}_{t-\tau_t}) - \nabla F_{S_{m_t}}(\mathbf{w}^*)\right\|^{\frac{1+\alpha}{\alpha}}.
$$

Plugging the above inequality into Eq. (C.17), we obtain

$$
\sum_{t=1}^{T}\eta_t\mathbb{E}_{S,I}\left[\left\langle\mathbf{w}_t - \mathbf{w}_{t-\tau_t}, \nabla f(\mathbf{w}_{t-\tau_t};S_{i_t,m_t})\right\rangle\right] \geq -\left(\frac{1-\alpha}{1+\alpha} + \frac{c_{\alpha,L}^2 F^{\frac{2\alpha}{1+\alpha}}(\mathbf{w}^*)}{n}\right)\sum_{t=1}^{T}(\tau_t + m - 1)\eta_t^2
$$

$$
- \frac{2\alpha}{1+\alpha}\sum_{t=1}^{T}(\tau_t + m - 1)\eta_t^2\mathbb{E}_{S,I}\left[\left\|\nabla F_{S_{m_t}}(\mathbf{w}_{t-\tau_t}) - \nabla F_{S_{m_t}}(\mathbf{w}^*)\right\|^{\frac{1+\alpha}{\alpha}}\right]. \quad \text{(C.18)}
$$

Combining Eq. (C.14), Eq, (C.15) and Eq. (C.18), we get

$$
2\sum_{t=1}^{T}\eta_t\mathbb{E}_{S,I}\left[\left\langle\mathbf{w}_t - \mathbf{w}^*, \nabla f(\mathbf{w}_{t-\tau_t};S_{i_t,m_t})\right\rangle\right]
$$

$$
\geq 2\sum_{t=1}^{T}\eta_t\mathbb{E}_{S,I}\left[F_{S_{m_t}}(\mathbf{w}_{t-\tau_t}) - F(\mathbf{w}^*)\right] - 2\left(\frac{1-\alpha}{1+\alpha} + \frac{c_{\alpha,L}^2 F^{\frac{2\alpha}{1+\alpha}}(\mathbf{w}^*)}{n}\right)\sum_{t=1}^{T}(\tau_t + m - 1)\eta_t^2
$$

$$
- \frac{\alpha}{1+\alpha}\sum_{t=1}^{T}\eta_t\left(4(\tau_t + m - 1)\eta_t - L^{-\frac{1}{\alpha}}\right)\mathbb{E}_{S,I}\left[\left\|\nabla F_{S_{m_t}}(\mathbf{w}_{t-\tau_t}) - \nabla F_{S_{m_t}}(\mathbf{w}^*)\right\|^{\frac{1+\alpha}{\alpha}}\right]
$$

$$
+ \frac{\alpha L^{-\frac{1}{\alpha}}}{1+\alpha}\sum_{t=1}^{T}\eta_t\mathbb{E}_{S,I}\left[\left\|\nabla f(\mathbf{w}_{t-\tau_t};S_{i_t,m_t}) - \nabla f(\mathbf{w}^*;S_{i_t,m_t})\right\|^{\frac{1+\alpha}{\alpha}}\right]. \quad \text{(C.19)}
$$

Recall the one-step recursion given in Eq. (B.18). Taking expectation w.r.t. $S, I$ to both sides followed by a summation over $t$, we have

$$
\sum_{t=1}^{T}\mathbb{E}_{S,I}\left[\|\mathbf{w}_{t+1} - \mathbf{w}^*\|^2\right] \leq \sum_{t=1}^{T}\mathbb{E}_{S,I}\left[\|\mathbf{w}_t - \mathbf{w}^*\|^2\right]
$$

$$
- 2\sum_{t=1}^{T}\eta_t\mathbb{E}_{S,I}\left[\left\langle\mathbf{w}_t - \mathbf{w}^*, \nabla f(\mathbf{w}_{t-\tau_t};S_{i_t,m_t})\right\rangle\right] + \sum_{t=1}^{T}\eta_t^2\mathbb{E}_{S,I}\left[\left\|\nabla f(\mathbf{w}_{t-\tau_t};S_{i_t,m_t})\right\|^2\right], \quad \text{(C.20)}
$$

which further implies that

$$
\mathbb{E}_{S,I}\big[\|\mathbf{w}_{T+1} - \mathbf{w}^*\|^2\big] \leq \|\mathbf{w}_1 - \mathbf{w}^*\|^2 - 2\sum_{t=1}^{T}\eta_t\mathbb{E}_{S,I}\Big[\big\langle \mathbf{w}_t - \mathbf{w}^*, \nabla f(\mathbf{w}_{t-\tau_t}; S_{i_t,m_t})\big\rangle\Big]
$$

$$
+ 2\sum_{t=1}^{T}\eta_t^2\mathbb{E}_{S,I}\Big[\big\|\nabla f(\mathbf{w}_{t-\tau_t}; S_{i_t,m_t}) - \nabla f(\mathbf{w}^*; S_{i_t,m_t})\big\|^2\Big] + 2\sum_{t=1}^{T}\eta_t^2\mathbb{E}_{S,I}\Big[\big\|\nabla f(\mathbf{w}^*; S_{i_t,m_t})\big\|^2\Big]
$$

$$
\leq \|\mathbf{w}_1 - \mathbf{w}^*\|^2 - 2\sum_{t=1}^{T}\eta_t\mathbb{E}_{S,I}\Big[\big\langle \mathbf{w}_t - \mathbf{w}^*, \nabla f(\mathbf{w}_{t-\tau_t}; S_{i_t,m_t})\big\rangle\Big]
$$

$$
+ \frac{2(1-\alpha)}{1+\alpha}\sum_{t=1}^{T}\eta_t^2 + \frac{4\alpha}{1+\alpha}\sum_{t=1}^{T}\eta_t^2\mathbb{E}_{S,I}\Big[\big\|\nabla f(\mathbf{w}_{t-\tau_t}; S_{i_t,m_t}) - \nabla f(\mathbf{w}^*; S_{i_t,m_t})\big\|^{\frac{1+\alpha}{\alpha}}\Big]
$$

$$
+ 2\sum_{t=1}^{T}\eta_t^2\mathbb{E}_{S,I}\Big[\big\|\nabla f(\mathbf{w}^*; S_{i_t,m_t})\big\|^2\Big],
$$

where the first and second inequality holds by inequality $\|a+b\|^2 \leq 2\|a\|^2 + 2\|b\|^2$, $\forall a, b \in \mathbb{R}^d$ and Young's inequality, respectively. Plugging Eq. (C.19) into the above inequality, we obtain

$$
2\sum_{t=1}^{T}\eta_t\mathbb{E}_{S,I}\Big[F_{S_{m_t}}(\mathbf{w}_{t-\tau_t}) - F(\mathbf{w}^*)\Big] \leq \|\mathbf{w}_1 - \mathbf{w}^*\|^2 + 2\sum_{t=1}^{T}\eta_t^2\mathbb{E}_{S,I}\Big[\big\|\nabla f(\mathbf{w}^*; S_{i_t,m_t})\big\|^2\Big]
$$

$$
+ 2\Big(\frac{1-\alpha}{1+\alpha} + \frac{c_{\alpha,L}^2 F^{\frac{2\alpha}{1+\alpha}}(\mathbf{w}^*)}{n}\Big)\sum_{t=1}^{T}(\tau_t + m - 1)\eta_t^2 + \frac{2(1-\alpha)}{1+\alpha}\sum_{t=1}^{T}\eta_t^2
$$

$$
+ \frac{\alpha}{1+\alpha}\sum_{t=1}^{T}\eta_t\Big(4(\tau_t + m - 1)\eta_t - L^{-\frac{1}{\alpha}}\Big)\mathbb{E}_{S,I}\Big[\big\|\nabla F_{S_{m_t}}(\mathbf{w}_{t-\tau_t}) - \nabla F_{S_{m_t}}(\mathbf{w}^*)\big\|^{\frac{1+\alpha}{\alpha}}\Big]
$$

$$
+ \frac{\alpha}{1+\alpha}\sum_{t=1}^{T}\eta_t\Big(4\eta_t - L^{-\frac{1}{\alpha}}\Big)\mathbb{E}_{S,I}\Big[\big\|\nabla f(\mathbf{w}_{t-\tau_t}; S_{i_t,m_t}) - \nabla f(\mathbf{w}^*; S_{i_t,m_t})\big\|^{\frac{1+\alpha}{\alpha}}\Big].
$$

Using the assumption that $(\max\{\tau_t + m - 1, 1\})\eta_t \leq 1/(4L^{\frac{1}{\alpha}})$ and Lemma A.4, we know from the above inequality that

$$
2\sum_{t=1}^{T}\eta_t\mathbb{E}_{S,I}\Big[F_{S_{m_t}}(\mathbf{w}_{t-\tau_t}) - F(\mathbf{w}^*)\Big] \leq \|\mathbf{w}_1 - \mathbf{w}^*\|^2
$$

$$
+ 2\Big(\frac{1-\alpha}{1+\alpha} + \frac{c_{\alpha,L}^2 F^{\frac{2\alpha}{1+\alpha}}(\mathbf{w}^*)}{n}\Big)\sum_{t=1}^{T}(\tau_t + m - 1)\eta_t^2 + \frac{2(1-\alpha)}{1+\alpha}\sum_{t=1}^{T}\eta_t^2 + 2c_{\alpha,L}^2 F^{\frac{2\alpha}{1+\alpha}}(\mathbf{w}^*)\sum_{t=1}^{T}\eta_t^2.
$$

This proves Eq. (C.12). Next, we consider the case where $\alpha = 0$. In this case, the following relationship holds for any $\mathbf{w} \in \mathcal{W}, \mathbf{z} \in \mathcal{Z}$:

$$
\|\nabla f(\mathbf{w}; \mathbf{z})\| \leq L + \sup_{\mathbf{z} \in \mathcal{Z}}\|\nabla f(0; \mathbf{z})\|. \tag{C.21}
$$

In this case, Eq. (C.17) can be improved as:

$$
\sum_{t=1}^{T}\eta_t\mathbb{E}_{S,I}\Big[\big\langle \mathbf{w}_t - \mathbf{w}_{t-\tau_t}, \nabla f(\mathbf{w}_{t-\tau_t}; S_{i_t,m_t})\big\rangle\Big]
$$

$$
\geq -\sum_{t=1}^{T}(\tau_t + m - 1)\eta_t^2\mathbb{E}_{S,I}\Big[\big\|\nabla F_{S_{m_t}}(\mathbf{w}_{t-\tau_t})\big\|^2\Big] \geq -\big(L + \sup_{\mathbf{z} \in \mathcal{Z}}\|\nabla f(0; \mathbf{z})\|\big)^2\sum_{t=1}^{T}(\tau_t + m - 1)\eta_t^2.
$$

Combining the above inequality with Eq. (C.14) yields

$$\sum_{t=1}^{T} \eta_t \mathbb{E}_{S,I} \Big[ \big\langle \mathbf{w}_t - \mathbf{w}^*, \nabla f(\mathbf{w}_{t-\tau_t}; S_{i_t, m_t}) \big\rangle \Big]$$

$$\geq \sum_{t=1}^{T} \eta_t \mathbb{E}_{S,I} \Big[ F_{S_{m_t}}(\mathbf{w}_{t-\tau_t}) - F(\mathbf{w}^*) \Big] - \big( L + \sup_{\mathbf{z} \in \mathcal{Z}} \|\nabla f(0; \mathbf{z})\| \big)^2 \sum_{t=1}^{T} (\tau_t + m - 1)\eta_t^2. \quad \text{(C.22)}$$

Besides, according to Eq. (C.21), we also have

$$\sum_{t=1}^{T} \eta_t^2 \mathbb{E}_{S,I} \Big[ \big\| \nabla f(\mathbf{w}_{t-\tau_t}; S_{i_t, m_t}) \big\|^2 \Big] \leq \big( L + \sup_{\mathbf{z} \in \mathcal{Z}} \|\nabla f(0; \mathbf{z})\| \big)^2 \sum_{t=1}^{T} \eta_t^2.$$

Plugging above two inequalities into Eq. (C.20), we derive

$$\sum_{t=1}^{T} \mathbb{E}_{S,I} \big[ \|\mathbf{w}_{t+1} - \mathbf{w}^*\|^2 \big] \leq \sum_{t=1}^{T} \mathbb{E}_{S,I} \big[ \|\mathbf{w}_t - \mathbf{w}^*\|^2 \big] + \big( L + \sup_{\mathbf{z} \in \mathcal{Z}} \|\nabla f(0; \mathbf{z})\| \big)^2 \sum_{t=1}^{T} \eta_t^2$$

$$- 2 \Big( \sum_{t=1}^{T} \eta_t \mathbb{E}_{S,I} \Big[ F_{S_{m_t}}(\mathbf{w}_{t-\tau_t}) - F(\mathbf{w}^*) \Big] - \big( L + \sup_{\mathbf{z} \in \mathcal{Z}} \|\nabla f(0; \mathbf{z})\| \big)^2 \sum_{t=1}^{T} (\tau_t + m - 1)\eta_t^2 \Big),$$

which further implies that

$$2 \sum_{t=1}^{T} \eta_t \mathbb{E}_{S,I} \Big[ F_{S_{m_t}}(\mathbf{w}_{t-\tau_t}) - F(\mathbf{w}^*) \Big] \leq \mathbb{E}_{S,I} \big[ \|\mathbf{w}_1 - \mathbf{w}^*\|^2 \big] + 2 \big( L + \sup_{\mathbf{z} \in \mathcal{Z}} \|\nabla f(0; \mathbf{z})\| \big)^2 \sum_{t=1}^{T} (\tau_t + m)\eta_t^2.$$

The proof is complete. □

### C.3. Proof of Theorem 5.4

Theorem 5.4 is a direct corollary of the following theorem by carefully tuning the hyper-parameters $\zeta$, $\theta$ and $\gamma$. Therefore, we first give the proof of the following theorem.

**Theorem C.3** (Excess Risk: Non-smooth Case). *Assume that the function $\mathbf{w} \mapsto f(\mathbf{w}; \mathbf{z})$ is nonnegative, convex, and $F(\mathbf{w}^*) \leq 1$. Assume $\mathbf{w} \mapsto \nabla f(\mathbf{w}; \mathbf{z})$ is $(\alpha, L)$-Hölder continuous with $\alpha \in [0, 1)$ for any $\mathbf{z} \in \mathcal{Z}$. Let $\{\mathbf{w}_t\}$ be produced by Algorithm 1 and constant step size, i.e. $\eta = c \hat{\tau}_M^{-\zeta} T^{-\theta}$, where $\hat{\tau}_M = \max_{t \in [T]} \{\tau_t + m\}$ and $c, \zeta, \theta > 0$. If $\{\eta_t\}$ further satisfies $(\max\{\tau_t + m - 1, 1\})\eta_t \leq 1/(4L^{\frac{1}{\alpha}})$, then it follows for any $\gamma > 0$ and $m \lesssim n$ that*

$$\mathbb{E}_{S,I}[F(A(S)) - F(\mathbf{w}^*)] \lesssim T^{\theta-1} \hat{\tau}_M^{\zeta} + T^{-\theta} \hat{\tau}_M^{1-\zeta} + \gamma^{\frac{1+\alpha}{\alpha-1}} + \gamma T^{1-\frac{2\theta}{1-\alpha}} \hat{\tau}_M^{\frac{1+\alpha}{1-\alpha} - \frac{2\zeta}{1-\alpha}}$$

$$+ \Big( \gamma^{-1} T^{-1} + \frac{\gamma T^{-2\theta-1} \hat{\tau}_M^{-2\zeta} N_T^M}{n} \big( 1 + \frac{N_T^M}{n} \big) \Big) \Big( T F^{\frac{2\alpha}{1+\alpha}}(\mathbf{w}^*) + T^{\frac{1-\alpha+2\alpha\theta}{1+\alpha}} \hat{\tau}_M^{\frac{2\alpha\zeta}{1+\alpha}} + T^{\frac{1-\alpha+2\alpha(1-\theta)}{1+\alpha}} \hat{\tau}_M^{\frac{2\alpha(1-\zeta)}{1+\alpha}} \Big), \quad \text{(C.23)}$$

*where we denote $A(S) = \sum_{t=1}^{T} \mathbf{w}_{t-\tau_t}/T$.*

*Proof.* For any $T \in \mathbb{N}^+$, assume that $\{\mathbf{w}_t\}_{t=1}^{T}$ is generated by Algorithm 1. It has been proved in Theorem 5.1 that for any $j \in [m]$ and $t \in [T]$, the following stability bound holds:

$$\epsilon_{t+1,j}^2 := \frac{1}{n} \sum_{i=1}^{n} \mathbb{E}_{S,S',I}[\|\mathbf{w}_{t+1} - \mathbf{w}_{t+1}^{(ij)}\|^2] \leq \frac{16c_{\alpha,L}^2}{n} \Big( 1 + \frac{N_t^M}{n} \Big) \sum_{k=1}^{t} \eta_k^2 \mathbb{I}_{[m_k=j]} \mathbb{E}_{S,I} \big[ F_{S_{m_k}}(\mathbf{w}_{k-\tau_k}) \big]^{\frac{2\alpha}{1+\alpha}}$$

$$+ \frac{2(1-\alpha)}{1+\alpha} \sum_{k=1}^{t} \Big( \big( 1 + \frac{m}{n} \big) \tau_k + m \Big)^{\frac{1+\alpha}{1-\alpha}} (2^{-\alpha} L \eta_k)^{\frac{2}{1-\alpha}}. \quad \text{(C.24)}$$

According to Theorem C.1, we have

$$\mathbb{E}_{S,I}[F(\mathbf{w}_{t+1}) - F_{S_j}(\mathbf{w}_{t+1})] \leq \frac{c_{\alpha,L}^2}{2\gamma} \mathbb{E}_{S,I}[F^{\frac{2\alpha}{1+\alpha}}(\mathbf{w}_{t+1})] + \frac{\gamma}{2}\epsilon_{t+1,j}^2. \tag{C.25}$$

Denote $\delta_{t,j} := \max\left\{\mathbb{E}_{S,I}[F(\mathbf{w}_t) - F_{S_j}(\mathbf{w}_t)], 0\right\}$. It follows from the concavity of function $x \mapsto x^{\frac{2\alpha}{1+\alpha}}$ that

$$\mathbb{E}_{S,I}[F^{\frac{2\alpha}{1+\alpha}}(\mathbf{w}_{t+1})] \leq \left(\mathbb{E}_{S,I}[F(\mathbf{w}_{t+1})]\right)^{\frac{2\alpha}{1+\alpha}} = \left(\mathbb{E}_{S,I}[F(\mathbf{w}_{t+1}) - F_{S_j}(\mathbf{w}_{t+1})] + \mathbb{E}_{S,I}[F_{S_j}(\mathbf{w}_{t+1})]\right)^{\frac{2\alpha}{1+\alpha}}$$

$$\leq \left(\delta_{t+1,j} + \mathbb{E}_{S,I}[F_{S_j}(\mathbf{w}_{t+1})]\right)^{\frac{2\alpha}{1+\alpha}} \leq \delta_{t+1,j}^{\frac{2\alpha}{1+\alpha}} + \mathbb{E}_{S,I}[F_{S_j}(\mathbf{w}_{t+1})]^{\frac{2\alpha}{1+\alpha}}.$$

Plugging the above inequality into Eq. (C.25) yields

$$\delta_{t+1,j} \leq \frac{c_{\alpha,L}^2}{2\gamma}\left(\delta_{t+1,j}^{\frac{2\alpha}{1+\alpha}} + \mathbb{E}_{S,I}[F_{S_j}(\mathbf{w}_{t+1})]^{\frac{2\alpha}{1+\alpha}}\right) + \frac{\gamma}{2}\epsilon_{t+1,j}^2$$

$$\leq \frac{1}{2}\left(\frac{2\alpha}{1+\alpha}\delta_{t+1,j} + \frac{1-\alpha}{1+\alpha}\left(\frac{c_{\alpha,L}^2}{\gamma}\right)^{\frac{1+\alpha}{1-\alpha}}\right) + \frac{c_{\alpha,L}^2}{2\gamma}\mathbb{E}_{S,I}[F_{S_j}(\mathbf{w}_{t+1})]^{\frac{2\alpha}{1+\alpha}} + \frac{\gamma}{2}\epsilon_{t+1,j}^2,$$

where the second inequality holds by Lemma A.2. For any given $t - \tau_t \in [T]$, hiding the constants $\alpha, L$, and letting $j = m_t$, the above inequality implies that

$$\mathbb{E}_{S,I}[F(\mathbf{w}_{t-\tau_t}) - F_{S_{m_t}}(\mathbf{w}_{t-\tau_t})] \leq \delta_{t-\tau_t,m_t} \lesssim \gamma^{\frac{1+\alpha}{\alpha-1}} + \gamma^{-1}\mathbb{E}_{S,I}[F_{S_{m_t}}(\mathbf{w}_{t-\tau_t})]^{\frac{2\alpha}{1+\alpha}} + \gamma\epsilon_{t-\tau_t,m_t}^2. \tag{C.26}$$

Multiplying $\eta_t$ to both sides of Eq. (C.26) followed by a summation over $t$ from 1 to $T$, we get

$$\sum_{t=1}^{T} \eta_t \mathbb{E}_{S,I}[F(\mathbf{w}_{t-\tau_t}) - F_{S_{m_t}}(\mathbf{w}_{t-\tau_t})] \lesssim \gamma^{\frac{1+\alpha}{\alpha-1}} \sum_{t=1}^{T} \eta_t + \gamma^{-1} \sum_{t=1}^{T} \eta_t \mathbb{E}_{S,I}[F_{S_{m_t}}(\mathbf{w}_{t-\tau_t})]^{\frac{2\alpha}{1+\alpha}} + \gamma \sum_{t=1}^{T} \eta_t \epsilon_{t-\tau_t,m_t}^2. \tag{C.27}$$

Analogous to Eq. (B.31), assuming $\{\eta_t\}$ is non-increasing, which naturally holds for constant step size, we can bound the last term in Eq. (C.27) as follows:

$$\sum_{t=1}^{T} \eta_t \epsilon_{t-\tau_t,m_t}^2 \leq \frac{16c_{\alpha,L}^2}{n}\left(1 + \frac{N_T^M}{n}\right) \sum_{k=1}^{T} \eta_k^3 \left(\sum_{t=1}^{T} \mathbb{I}_{[m_k=m_t]}\right)\mathbb{E}_{S,I}\left[F_{S_{m_k}}(\mathbf{w}_{k-\tau_k})\right]^{\frac{2\alpha}{1+\alpha}}$$

$$+ \frac{2(1-\alpha)}{1+\alpha}\sum_{t=1}^{T}\eta_t \sum_{k=1}^{t-\tau_t-1}\left((1+\frac{m}{n})\tau_k + m\right)^{\frac{1+\alpha}{1-\alpha}}(2^{-\alpha}L\eta_k)^{\frac{2}{1-\alpha}}$$

$$\leq 16c_{\alpha,L}^2\frac{N_T^M}{n}\left(1 + \frac{N_T^M}{n}\right)\sum_{k=1}^{T}\eta_k^3\mathbb{E}_{S,I}\left[F_{S_{m_k}}(\mathbf{w}_{k-\tau_k})\right]^{\frac{2\alpha}{1+\alpha}}$$

$$+ \frac{2(1-\alpha)}{1+\alpha}\sum_{t=1}^{T}\sum_{k=1}^{T}\left((1+\frac{m}{n})\tau_k + m\right)^{\frac{1+\alpha}{1-\alpha}}(2^{-\alpha}L\eta_k)^{\frac{3-\alpha}{1-\alpha}}, \tag{C.28}$$

where the first inequality holds since $\epsilon_{t,j}^2$ is non-decreasing with $t$, i.e. $\epsilon_{t,j}^2 \leq \epsilon_{T,j}^2$, $\forall t \in [T]$ and $\eta_t$ is non-increasing, the second inequality holds by the definition of $N_T^M$. Consider the case of constant step size, i.e. $\eta_t = c\hat{\tau}_M^{-\zeta}T^{-\theta}$. Plugging Eq. (C.28) into Eq. (C.27) and ignoring the constants including $\alpha, L, c$, we derive

$$\sum_{t=1}^{T} \eta_t \mathbb{E}_{S,I}[F(\mathbf{w}_{t-\tau_t}) - F_{S_{m_t}}(\mathbf{w}_{t-\tau_t})] \lesssim \gamma^{\frac{1+\alpha}{\alpha-1}}T^{1-\theta}\hat{\tau}_M^{-\zeta} + \gamma^{-1}T^{-\theta}\hat{\tau}_M^{-\zeta}\sum_{t=1}^{T}\mathbb{E}_{S,I}[F_{S_{m_t}}(\mathbf{w}_{t-\tau_t})]^{\frac{2\alpha}{1+\alpha}}$$

$$+ \frac{\gamma N_T^M}{n}\left(1 + \frac{N_T^M}{n}\right)T^{-3\theta}\hat{\tau}_M^{-3\zeta}\sum_{t=1}^{T}\mathbb{E}_{S,I}\left[F_{S_{m_t}}(\mathbf{w}_{t-\tau_t})\right]^{\frac{2\alpha}{1+\alpha}} + \gamma T^{2-\frac{(3-\alpha)\theta}{1-\alpha}}\hat{\tau}_M^{\frac{1+\alpha}{1-\alpha} - \frac{(3-\alpha)\zeta}{1-\alpha}}, \tag{C.29}$$

where we have used

$$\sum_{t=1}^{T}\sum_{k=1}^{T}\Big(\big(1+\frac{m}{n}\big)\tau_k+m\Big)^{\frac{1+\alpha}{1-\alpha}}\eta_k^{\frac{3-\alpha}{1-\alpha}}\lesssim T^{2-\frac{(3-\alpha)\theta}{1-\alpha}}(\tau_M+m)^{\frac{1+\alpha}{1-\alpha}-\frac{(3-\alpha)\zeta}{1-\alpha}}\lesssim T^{2-\frac{(3-\alpha)\theta}{1-\alpha}}\hat{\tau}_M^{\frac{1+\alpha}{1-\alpha}-\frac{(3-\alpha)\zeta}{1-\alpha}}.$$

Note that we also assume that $\eta_t$ satisfies $\eta_t(\max\{\tau_t+m-1,2\})\le 1/(2L^{\frac{1}{\alpha}})$. Using Lemma 5.3, we obtain

$$\sum_{t=1}^{T}\eta_t\mathbb{E}_{S,I}\Big[F_{S_{m_t}}(\mathbf{w}_{t-\tau_t})-F(\mathbf{w}^*)\Big]\lesssim\|\mathbf{w}_1-\mathbf{w}^*\|^2+T^{1-2\theta}\hat{\tau}_M^{1-2\zeta}+F^{\frac{2\alpha}{1+\alpha}}(\mathbf{w}^*)\Big(\frac{T^{1-2\theta}}{n}\hat{\tau}_M^{1-2\zeta}+T^{1-2\theta}\hat{\tau}_M^{-2\zeta}\Big).$$

Ignoring $\|\mathbf{w}_1-\mathbf{w}^*\|^2$ which is independent of Algorithm 1 and assuming $F(\mathbf{w}^*)\le 1$, the above inequality implies that

$$\sum_{t=1}^{T}\eta_t\mathbb{E}_{S,I}\Big[F_{S_{m_t}}(\mathbf{w}_{t-\tau_t})-F(\mathbf{w}^*)\Big]\lesssim 1+T^{1-2\theta}\hat{\tau}_M^{1-2\zeta} \tag{C.30}$$

and

$$\sum_{t=1}^{T}\mathbb{E}_{S,I}[F_{S_{m_t}}(\mathbf{w}_{t-\tau_t})]\lesssim TF(\mathbf{w}^*)+T^{\theta}\hat{\tau}_M^{\zeta}+T^{1-\theta}\hat{\tau}_M^{1-\zeta}. \tag{C.31}$$

Note that

$$\sum_{t=1}^{T}\mathbb{E}_{S,I}[F_{S_{m_t}}(\mathbf{w}_{t-\tau_t})]^{\frac{2\alpha}{1+\alpha}}=T\big(\frac{1}{T}\sum_{t=1}^{T}\mathbb{E}_{S,I}[F_{S_{m_t}}(\mathbf{w}_{t-\tau_t})]^{\frac{2\alpha}{1+\alpha}}\big)$$

$$\le T\big(\frac{1}{T}\sum_{t=1}^{T}\mathbb{E}_{S,I}[F_{S_{m_t}}(\mathbf{w}_{t-\tau_t})]\big)^{\frac{2\alpha}{1+\alpha}}=T^{\frac{1-\alpha}{1+\alpha}}\big(\sum_{t=1}^{T}\mathbb{E}_{S,I}[F_{S_{m_t}}(\mathbf{w}_{t-\tau_t})]\big)^{\frac{2\alpha}{1+\alpha}},$$

where the inequality holds by the concavity of function $x\mapsto x^{\frac{2\alpha}{1+\alpha}}$. Plugging Eq. (C.31) into the above inequality yields

$$\sum_{t=1}^{T}\mathbb{E}_{S,I}[F_{S_{m_t}}(\mathbf{w}_{t-\tau_t})]^{\frac{2\alpha}{1+\alpha}}\lesssim TF^{\frac{2\alpha}{1+\alpha}}(\mathbf{w}^*)+T^{\frac{1-\alpha+2\alpha\theta}{1+\alpha}}\hat{\tau}_M^{\frac{2\alpha\zeta}{1+\alpha}}+T^{\frac{1-\alpha+2\alpha(1-\theta)}{1+\alpha}}\hat{\tau}_M^{\frac{2\alpha(1-\zeta)}{1+\alpha}}. \tag{C.32}$$

We plug Eq. (C.32) into Eq. (C.29) and obtain

$$\sum_{t=1}^{T}\eta_t\mathbb{E}_{S,I}[F(\mathbf{w}_{t-\tau_t})-F_{S_{m_t}}(\mathbf{w}_{t-\tau_t})]\lesssim\gamma^{\frac{1+\alpha}{\alpha-1}}T^{1-\theta}\hat{\tau}_M^{-\zeta}+\gamma T^{2-\frac{(3-\alpha)\theta}{1-\alpha}}\hat{\tau}_M^{\frac{1+\alpha}{1-\alpha}-\frac{(3-\alpha)\zeta}{1-\alpha}}$$

$$+\Big(\gamma^{-1}T^{-\theta}\hat{\tau}_M^{-\zeta}+\frac{\gamma T^{-3\theta}\hat{\tau}_M^{-3\zeta}N_T^M}{n}\big(1+\frac{N_T^M}{n}\big)\Big)\Big(TF^{\frac{2\alpha}{1+\alpha}}(\mathbf{w}^*)+T^{\frac{1-\alpha+2\alpha\theta}{1+\alpha}}\hat{\tau}_M^{\frac{2\alpha\zeta}{1+\alpha}}+T^{\frac{1-\alpha+2\alpha(1-\theta)}{1+\alpha}}\hat{\tau}_M^{\frac{2\alpha(1-\zeta)}{1+\alpha}}\Big). \tag{C.33}$$

Combining Eq. (C.30) and Eq. (C.33), we derive

$$\sum_{t=1}^{T}\eta_t\mathbb{E}_{S,I}[F(\mathbf{w}_{t-\tau_t})-F(\mathbf{w}^*)]\lesssim 1+T^{1-2\theta}\hat{\tau}_M^{1-2\zeta}+\gamma^{\frac{1+\alpha}{\alpha-1}}T^{1-\theta}\hat{\tau}_M^{-\zeta}+\gamma T^{2-\frac{(3-\alpha)\theta}{1-\alpha}}\hat{\tau}_M^{\frac{1+\alpha}{1-\alpha}-\frac{(3-\alpha)\zeta}{1-\alpha}}$$

$$+\Big(\gamma^{-1}T^{-\theta}\hat{\tau}_M^{-\zeta}+\frac{\gamma T^{-3\theta}\hat{\tau}_M^{-3\zeta}N_T^M}{n}\big(1+\frac{N_T^M}{n}\big)\Big)\Big(TF^{\frac{2\alpha}{1+\alpha}}(\mathbf{w}^*)+T^{\frac{1-\alpha+2\alpha\theta}{1+\alpha}}\hat{\tau}_M^{\frac{2\alpha\zeta}{1+\alpha}}+T^{\frac{1-\alpha+2\alpha(1-\theta)}{1+\alpha}}\hat{\tau}_M^{\frac{2\alpha(1-\zeta)}{1+\alpha}}\Big). \tag{C.34}$$

Since $\mathbf{w}\mapsto f(\mathbf{w};\mathbf{z})$ is convex, we have

$$\mathbb{E}_{S,I}[F(A(S))-F(\mathbf{w}^*)]:=\mathbb{E}_{S,I}[F\big(\frac{\sum_{t=1}^{T}\eta\mathbf{w}_{t-\tau_t}}{T\eta}\big)-F(\mathbf{w}^*)]\le\frac{1}{T\eta}\sum_{t=1}^{T}\eta\mathbb{E}_{S,I}\big[F(\mathbf{w}_{t-\tau_t})-F(\mathbf{w}^*)\big],$$

which together with Eq. (C.34) yields the excess risk bounds

$$\mathbb{E}_{S,I}[F(A(S)) - F(\mathbf{w}^*)] \lesssim T^{\theta-1}\hat{\tau}_M^{\zeta} + T^{-\theta}\hat{\tau}_M^{1-\zeta}$$

$$+ \gamma^{\frac{1+\alpha}{\alpha-1}} + \gamma T^{1-\frac{2\theta}{1-\alpha}}\hat{\tau}_M^{\frac{1+\alpha}{1-\alpha}-\frac{2\zeta}{1-\alpha}} + \left(\gamma^{-1}T^{-1} + \frac{\gamma T^{-2\theta-1}\hat{\tau}_M^{-2\zeta}N_T^M}{n}\left(1 + \frac{N_T^M}{n}\right)\right)$$

$$\times \left(TF^{\frac{2\alpha}{1+\alpha}}(\mathbf{w}^*) + T^{\frac{1-\alpha+2\alpha\theta}{1+\alpha}}\hat{\tau}_M^{\frac{2\alpha\zeta}{1+\alpha}} + T^{\frac{1-\alpha+2\alpha(1-\theta)}{1+\alpha}}\hat{\tau}_M^{\frac{2\alpha(1-\zeta)}{1+\alpha}}\right).$$

The proof is complete. □

*Proof of Theorem 5.4.* Assume that $\eta_t = c\hat{\tau}_M^{-\zeta}T^{-\theta}$, $m \leq c_1 n$ and $T \geq c_2 mn$, where $c_1, c_2 > 0$ are algorithm-independent constants. Then, for any $\zeta, \theta > 0$ satisfying $\zeta + \theta = 1$, the following inequality holds

$$(\max\{\tau_t + m - 1, 1\})\eta_t \leq \max\left\{\hat{\tau}_M, 1\right\} \cdot c\hat{\tau}_M^{-\zeta}T^{-\theta} \leq c,$$

where the last inequality holds by $\hat{\tau}_M \leq T$. Therefore, choosing $c \leq 1/(4L^{\frac{1}{\alpha}})$ makes the conditions in Theorem C.3 satisfied. Note that we assume $N_T^M \asymp T/m$.

**For the part 1,** letting $\zeta = \frac{1}{2}$ and $\theta = \frac{1}{2}$, it follows from Eq. (C.23) that

$$\mathbb{E}_{S,I}[F(A(S)) - F(\mathbf{w}^*)] \lesssim T^{-\frac{1}{2}}\hat{\tau}_M^{\frac{1}{2}} + T^{-\frac{1}{2}}\hat{\tau}_M^{\frac{1}{2}} + \gamma^{\frac{1+\alpha}{\alpha-1}} + \gamma T^{-\frac{\alpha}{1-\alpha}}\hat{\tau}_M^{\frac{\alpha}{1-\alpha}}$$

$$+ \left(\gamma^{-1}T^{-1} + \gamma T^{-2}\hat{\tau}_M^{-1}(\frac{T}{mn})^2)\right)\left(TF^{\frac{2\alpha}{1+\alpha}}(\mathbf{w}^*) + T^{\frac{1}{1+\alpha}}\hat{\tau}_M^{\frac{\alpha}{1+\alpha}}\right). \quad (C.35)$$

Under the assumption that $\hat{\tau}_M \asymp m$, letting $\gamma = m^{\frac{1}{2}}n^{\frac{1}{2}}$ and $T \asymp m^2 n$, we have

$$T^{-\frac{1}{2}}\hat{\tau}_M^{\frac{1}{2}} \lesssim m^{-1}n^{-\frac{1}{2}}m^{\frac{1}{2}} = (mn)^{-\frac{1}{2}},$$

$$\gamma^{\frac{1+\alpha}{\alpha-1}} \lesssim \gamma^{-1} \leq (mn)^{-\frac{1}{2}},$$

$$\gamma T^{-\frac{\alpha}{1-\alpha}}\hat{\tau}_M^{\frac{\alpha}{1-\alpha}} \lesssim (mn)^{\frac{1}{2}-\frac{\alpha}{1-\alpha}} \lesssim (mn)^{-\frac{1}{2}},$$

$$\left(\gamma^{-1}T^{-1} + \gamma T^{-2}\hat{\tau}_M^{-1}(\frac{T}{mn})^2)\right)T \lesssim (mn)^{-\frac{1}{2}},$$

$$\left(\gamma^{-1}T^{-1} + \gamma T^{-2}\hat{\tau}_M^{-1}(\frac{T}{mn})^2)\right)T^{\frac{1}{1+\alpha}}\hat{\tau}_M^{\frac{\alpha}{1+\alpha}} \lesssim (mn)^{-\frac{1}{2}}.$$

Plugging the above inequalities into Eq. (C.35), we have

$$\mathbb{E}_{S,I}[F(A(S)) - F(\mathbf{w}^*)] \lesssim (mn)^{-\frac{1}{2}}(1 + F^{\frac{2\alpha}{1+\alpha}}(\mathbf{w}^*)).$$

**For the part 2,** taking $T \asymp m^{\frac{3}{1+\alpha}}n^{\frac{2-\alpha}{1+\alpha}}$, $\zeta = \frac{1+\alpha}{4-2\alpha}$, $\theta = \frac{3-3\alpha}{4-2\alpha}$ and $\gamma = m^{\frac{1}{2}}n^{\frac{1}{2}}$, the following inequalities holds:

$$T^{\theta-1}\hat{\tau}_M^{\zeta} \lesssim (mn)^{-\frac{1}{2}},$$

$$T^{-\theta}\hat{\tau}_M^{1-\zeta} \lesssim (mn)^{-\frac{3(1-\alpha)}{2(1+\alpha)}} \lesssim (mn)^{-\frac{1}{2}},$$

$$\gamma^{\frac{1+\alpha}{\alpha-1}} \lesssim \gamma^{-1} = (mn)^{-\frac{1}{2}},$$

$$\gamma T^{1-\frac{2\theta}{1-\alpha}}\hat{\tau}_M^{\frac{1+\alpha}{1-\alpha}-\frac{2\zeta}{1-\alpha}} \lesssim (mn)^{-\frac{1}{2}},$$

$$\frac{\gamma T^{-2\theta}\hat{\tau}_M^{-2\zeta}N_T^M}{n}\left(1 + \frac{N_T^M}{n}\right) \lesssim (mn)^{-\frac{1}{2}},$$

$$T^{\frac{1-\alpha+2\alpha\theta}{1+\alpha}}\hat{\tau}_M^{\frac{2\alpha\zeta}{1+\alpha}} + T^{\frac{1-\alpha+2\alpha(1-\theta)}{1+\alpha}}\hat{\tau}_M^{\frac{2\alpha(1-\zeta)}{1+\alpha}} \lesssim T.$$

Plugging above inequalities into Eq. (C.23) yields

$$\mathbb{E}_{S,I}[F(A(S)) - F(\mathbf{w}^*)] \lesssim (mn)^{-\frac{1}{2}}(1 + F^{\frac{2\alpha}{1+\alpha}}(\mathbf{w}^*)).$$

The proof is complete. □

# D. Extension to Heterogeneous Data

We first present the pseudocode of *random* ASGD proposed by Koloskova et al. (2022).

---

**Algorithm 2** *Random* ASGD

---

**Input:** Initial weights $\mathbf{w_1} \in \mathbb{R}^d$, iteration number $T$, learning rate scheme $\{\eta_t\}_{t=1}^T$.
**Initialization:** Initialize $\mathcal{R}_1 = \emptyset, \mathcal{A}_1 = \{(j,1)|j \in \{1,\ldots,m\}\}$. The server assigns $\mathbf{w}_1$ to all workers for gradient computation
**Server**
   **for** $t = 1, \ldots, T$ **do**
      Receive gradient from worker $m_t$: $\nabla f(\mathbf{w}_{t-\tau_t}; \mathbf{z}_{i_t,m_t})$, where $i_t \sim \text{Uniform}[1, \cdots, \text{n}]$
      Update work receiving set: $\mathcal{R}_{t+1} = \mathcal{R}_t \bigcup \{(m_t, t - \tau_t)\}$
      Update the model: $\mathbf{w}_{t+1} = \mathbf{w}_t - \eta_t \nabla f(\mathbf{w}_{t-\tau_t}; \mathbf{z}_{i_t,m_t})$
      Select a new worker $a_{t+1} \sim \text{Uniform}[1, \cdots, \text{m}]$ and assign $\mathbf{w}_{t+1}$ to worker $a_{t+1}$ for gradient computation
      Update work assigning set:$\mathcal{A}_{t+1} = \mathcal{A}_t \bigcup \{(a_{t+1}, t+1)\}$
   **end for**

---

## D.1. Proof of Lemma 6.1

*Proof.* We first consider the delay-adaptive step size. Recall the notations of work assigning set $\mathcal{A}_t$ and receiving set $\mathcal{R}_t$. For any $t \in \{2, 3, \cdots\}$, let $\mathcal{G}_t = \mathcal{A}_{t-1} \setminus \mathcal{R}_t$ denote the set of unfinished jobs during the update from $\mathbf{w}_{t-1}$ to $\mathbf{w}_t$. Thus, before training, $\mathcal{G}_1 = \emptyset$. We adopt the perturbed iterate technique (Mania et al., 2017) and introduce the following virtual sequence $\{\widehat{\mathbf{w}}_t\}$ as

$$\widehat{\mathbf{w}}_{t+1} = \begin{cases} \widehat{\mathbf{w}}_t - \sum_{j=1}^m \eta_{1+\widehat{\tau}(1,j)} \nabla f(\mathbf{w}_t; S_{i_{1+\widehat{\tau}(1,j)},j}), & t = 1, \\ \widehat{\mathbf{w}}_t - \eta_{t+\widehat{\tau}(t,a_t)} \nabla f(\mathbf{w}_t; S_{i_{t+\widehat{\tau}(t,a_t)},a_t}), & t \in \{2, 3, \cdots\}, \end{cases}$$

where $t + \widehat{\tau}(t, a_t)$ denotes the time when the gradients calculated by worker $a_t$ on $\mathbf{w}_t$ is applied[3]. Note that if we denote $k = t + \widehat{\tau}(t, a_t)$, then it holds $t = k - \tau_k$. In our notation system, $\mathbf{w}_1 = \widehat{\mathbf{w}}_1$. Note that for random ASGD, the following relationships hold:

$$t + \widehat{\tau}(t, a_t) \neq t' + \widehat{\tau}(t', a_{t'}), \quad \forall t, t' \in \{1, 2, \cdots\}$$

and for $t \in \{2, 3, \cdots\}$,

$$\widehat{\mathbf{w}}_t - \mathbf{w}_t = - \sum_{(a_k, k) \in \mathcal{G}_t} \eta_{k+\widehat{\tau}_k} \nabla f(\mathbf{w}_k; S_{i_{k+\widehat{\tau}_k}, a_k}). \tag{D.1}$$

According to above update formula, for $t \in \{2, 3, \cdots\}$, we have

$$\|\widehat{\mathbf{w}}_{t+1} - \mathbf{w}^*\|^2 = \|\widehat{\mathbf{w}}_t - \eta_{t+\widehat{\tau}_t} \nabla f(\mathbf{w}_t; S_{i_{t+\widehat{\tau}_t}, a_t}) - \mathbf{w}^*\|^2$$
$$= \|\widehat{\mathbf{w}}_t - \mathbf{w}^*\|^2 + \eta_{t+\widehat{\tau}_t}^2 \|\nabla f(\mathbf{w}_t; S_{i_{t+\widehat{\tau}_t}, a_t})\|^2 - 2\eta_{t+\widehat{\tau}_t} \langle \widehat{\mathbf{w}}_t - \mathbf{w}^*, \nabla f(\mathbf{w}_t; S_{i_{t+\widehat{\tau}_t}, a_t}) \rangle. \tag{D.2}$$

We introduce the following decomposition

$$\langle \widehat{\mathbf{w}}_t - \mathbf{w}^*, \nabla f(\mathbf{w}_t; S_{i_{t+\widehat{\tau}_t}, a_t}) \rangle = \langle \mathbf{w}_t - \mathbf{w}^*, \nabla f(\mathbf{w}_t; S_{i_{t+\widehat{\tau}_t}, a_t}) \rangle + \langle \widehat{\mathbf{w}}_t - \mathbf{w}_t, \nabla f(\mathbf{w}_t; S_{i_{t+\widehat{\tau}_t}, a_t}) \rangle. \tag{D.3}$$

**For the first term** in the RHS of Eq. (D.3), taking expectation w.r.t. $S, I, A$, we derive

$$\mathbb{E}_{S,I,A}\left[\langle \mathbf{w}_t - \mathbf{w}^*, \nabla f(\mathbf{w}_t; S_{i_{t+\widehat{\tau}_t}, a_t}) \rangle\right] = \mathbb{E}_{S,I,A}\left[\langle \mathbf{w}_t - \mathbf{w}^*, \nabla F_S(\mathbf{w}_t) \rangle\right],$$

where the equality holds since both $i_{t+\widehat{\tau}_t}$ and $a_t$ are independent of $\mathbf{w}_t$, and

$$\mathbb{E}_{i_{t+\widehat{\tau}_t}, a_t}[\nabla f(\mathbf{w}_t; S_{i_{t+\widehat{\tau}_t}, a_t})] = \frac{1}{mn} \sum_{j=1}^m \sum_{i=1}^n \nabla f(\mathbf{w}_t; S_{i,j}) = \nabla F_S(\mathbf{w}_t).$$

---

[3]For brevity in the following proof, we use the shorthand $\widehat{\tau}_t = \widehat{\tau}(t, a_t)$.

According to the convexity and smoothness of $f$, using Lemma A.6 with $\alpha = 1$, we obtain

$$\mathbb{E}_{S,I,A}\Big[\big\langle \mathbf{w}_t - \mathbf{w}^*, \nabla f(\mathbf{w}_t; S_{i_{t+\widehat{\tau}_t},a_t})\big\rangle\Big] \geq \mathbb{E}_{S,I,A}\Big[F_S(\mathbf{w}_t) - F(\mathbf{w}^*) + \frac{1}{2L}\big\|\nabla F_S(\mathbf{w}_t) - \nabla F_S(\mathbf{w}^*)\big\|^2\Big]. \tag{D.4}$$

Similarly, we have

$$\mathbb{E}_{S,I,A}\Big[\big\langle \mathbf{w}_t - \mathbf{w}^*, \nabla f(\mathbf{w}_t; S_{i_{t+\widehat{\tau}_t},a_t})\big\rangle\Big]$$
$$\geq \mathbb{E}_{S,I,A}\Big[F_S(\mathbf{w}_t) - F(\mathbf{w}^*) + \frac{1}{2L}\big\|\nabla f(\mathbf{w}_t; S_{i_{t+\widehat{\tau}_t},a_t}) - \nabla f(\mathbf{w}^*; S_{i_{t+\widehat{\tau}_t},a_t})\big\|^2\Big]. \tag{D.5}$$

**For the second term** in the RHS of Eq. (D.3), taking expectation w.r.t. $S, I, A$, we get

$$\mathbb{E}_{S,I,A}\Big[\big\langle \widehat{\mathbf{w}}_t - \mathbf{w}_t, \nabla f(\mathbf{w}_t; S_{i_{t+\widehat{\tau}_t},a_t})\big\rangle\Big] = \mathbb{E}_{S,I,A}\Big[\big\langle \widehat{\mathbf{w}}_t - \mathbf{w}_t, \nabla F_S(\mathbf{w}_t)\big\rangle\Big].$$

The above equality holds since $\widehat{\mathbf{w}}_t$ is determined by $\{i_{k+\widehat{\tau}_k}, a_k\}_{k=1}^{t-1}$ which is independent of $\{i_{t+\widehat{\tau}_t}, a_t\}$. Recall Eq. (D.1). The above equality implies that

$$\sum_{t=2}^{T} \mathbb{E}_{S,I,A}\Big[\eta_{t+\widehat{\tau}_t}\big\langle \widehat{\mathbf{w}}_t - \mathbf{w}_t, \nabla f(\mathbf{w}_t; S_{i_{t+\widehat{\tau}_t},a_t})\big\rangle\Big]$$

$$= \mathbb{E}_{S,I,A}\Big[-\sum_{t=2}^{T}\sum_{(a_k,k)\in\mathcal{G}_t} \eta_{t+\widehat{\tau}_t}\eta_{k+\widehat{\tau}_k}\big\langle \nabla f(\mathbf{w}_k; S_{i_{k+\widehat{\tau}_k},a_k}), \nabla F_S(\mathbf{w}_t)\big\rangle\Big]$$

$$= \mathbb{E}_{S,I,A}\Big[-\sum_{t=2}^{T}\sum_{(a_k,k)\in\mathcal{G}_t} \eta_{t+\widehat{\tau}_t}\eta_{k+\widehat{\tau}_k}\mathbb{E}_{i_{k+\widehat{\tau}_k},a_k}\Big[\big\langle \nabla f(\mathbf{w}_k; S_{i_{k+\widehat{\tau}_k},a_k}), \nabla F_S(\mathbf{w}_t)\big\rangle\Big]\Big]$$

$$= \mathbb{E}_{S,I,A}\Big[-\sum_{t=2}^{T}\sum_{(a_k,k)\in\mathcal{G}_t} \eta_{t+\widehat{\tau}_t}\eta_{k+\widehat{\tau}_k}\big\langle \nabla F_S(\mathbf{w}_k), \nabla F_S(\mathbf{w}_t)\big\rangle\Big]$$

$$\geq -\frac{1}{2}\sum_{t=2}^{T}\sum_{(a_k,k)\in\mathcal{G}_t} \mathbb{E}_{S,I,A}\Big[\eta_{k+\widehat{\tau}_k}^2\|\nabla F_S(\mathbf{w}_k)\|^2 + \eta_{t+\widehat{\tau}_t}^2\|\nabla F_S(\mathbf{w}_t)\|^2\Big], \tag{D.6}$$

where the third equality holds since $\mathbf{w}_t$ is independent of the element in $\mathcal{G}_t$ and we assume that the delays and the clients are independent, i.e. $a_k$ is independent of $\widehat{\tau}_k, \widehat{\tau}_t$.[4] Note that by definition $|\mathcal{G}_t| = m - 1$ and for any $t \in \{2, \cdots, T\}$, the number of appearance of $(a_t, t)$ in $\bigcup_{i=2}^{T}\mathcal{G}_i$ is less or equal to $\widehat{\tau}_t$. Thus, Eq. (D.6) indicates that

$$\sum_{t=2}^{T} \mathbb{E}_{S,I,A}\Big[\eta_{t+\widehat{\tau}_t}\big\langle \widehat{\mathbf{w}}_t - \mathbf{w}_t, \nabla f(\mathbf{w}_t; S_{i_{t+\widehat{\tau}_t},a_t})\big\rangle\Big] \geq -\mathbb{E}_{S,I,A}\Big[\sum_{t=2}^{T} \frac{\widehat{\tau}_t + m - 1}{2}\eta_{t+\widehat{\tau}_t}^2\|\nabla F_S(\mathbf{w}_t)\|^2\Big]. \tag{D.7}$$

Taking an expectation w.r.t. $S, I$ to the both sides of Eq. (D.2) followed by a summation over $t$, we have

$$2\sum_{t=2}^{T} \mathbb{E}_{S,I,A}\Big[\eta_{t+\widehat{\tau}_t}\big\langle \widehat{\mathbf{w}}_t - \mathbf{w}^*, \nabla f(\mathbf{w}_t; S_{i_{t+\widehat{\tau}_t},a_t})\big\rangle\Big]$$

$$\leq \mathbb{E}_{S,I,A}[\|\widehat{\mathbf{w}}_2 - \mathbf{w}^*\|^2] - \mathbb{E}_{S,I,A}[\|\widehat{\mathbf{w}}_{T+1} - \mathbf{w}^*\|^2] + \sum_{t=2}^{T} \mathbb{E}_{S,I,A}[\eta_{t+\widehat{\tau}_t}^2\|\nabla f(\mathbf{w}_t; S_{i_{t+\widehat{\tau}_t},a_t})\|^2]. \tag{D.8}$$

---

[4]Eq. (D.6) holds without such an assumption if we consider a constant step size, i.e. $\eta_t = \eta, \forall t \in \{1, 2, \cdots\}$.

Plugging Eq. (D.4), Eq. (D.5) and Eq. (D.7) into Eq. (D.8), we have

$$2\mathbb{E}_{S,I,A}\Big[\sum_{t=2}^{T}\eta_{t+\widehat{\tau}_t}\big(F_S(\mathbf{w}_t)-F(\mathbf{w}^*)\big)\Big] \leq \mathbb{E}_{S,I,A}[\|\widehat{\mathbf{w}}_2-\mathbf{w}^*\|^2]+\mathbb{E}_{S,I,A}[\sum_{t=2}^{T}\eta_{t+\widehat{\tau}_t}^2\|\nabla f(\mathbf{w}_t;S_{i_{t+\widehat{\tau}_t},a_t})\|^2]$$

$$+\mathbb{E}_{S,I,A}\Big[\sum_{t=2}^{T}\eta_{t+\widehat{\tau}_t}^2(\widehat{\tau}_t+m-1)\|\nabla F_S(\mathbf{w}_t)\|^2\Big]-\frac{1}{2L}\mathbb{E}_{S,I,A}\Big[\sum_{t=2}^{T}\eta_{t+\widehat{\tau}_t}\|\nabla F_S(\mathbf{w}_t)-\nabla F_S(\mathbf{w}^*)\|^2\Big]$$

$$-\frac{1}{2L}\mathbb{E}_{S,I,A}\Big[\sum_{t=2}^{T}\eta_{t+\widehat{\tau}_t}\|\nabla f(\mathbf{w}_t;S_{i_{t+\widehat{\tau}_t},a_t})-\nabla f(\mathbf{w}^*;S_{i_{t+\widehat{\tau}_t},a_t})\|^2\Big]. \tag{D.9}$$

Recall the definition of $\mathbf{w}^*$. The following inequality holds:

$$\mathbb{E}_S\Big[\|\nabla F_S(\mathbf{w}^*)\|^2\Big] = \mathbb{E}_S\Big[\langle\frac{1}{mn}\sum_{j=1}^{m}\sum_{i=1}^{n}\nabla f(\mathbf{w}^*;S_{i,j}),\frac{1}{mn}\sum_{j'=1}^{m}\sum_{i'=1}^{n}\nabla f(\mathbf{w}^*;S_{i',j'})\rangle\Big]$$

$$= \frac{1}{(mn)^2}\sum_{j=1}^{m}\sum_{i=1}^{n}\mathbb{E}_S\big[\|\nabla f(\mathbf{w}^*;S_{i,j})\|^2\big] \leq \frac{2LF(\mathbf{w}^*)}{mn}, \tag{D.10}$$

where the second equality holds since each sample in $S_j$ is drawn independently from $\mathcal{D}_j$ and for any $i \in [n]$,

$$\frac{1}{m}\sum_{j=1}^{m}\mathbb{E}_S[\nabla f(\mathbf{w}^*;S_{i,j})] = \frac{1}{m}\sum_{j=1}^{m}\mathbb{E}_{\mathbf{z}_j\sim\mathcal{D}_j}[\nabla f(\mathbf{w}^*;\mathbf{z}_j)] = 0.$$

The last inequality in Eq. (D.10) holds by Lemma A.4. We then know from Eq. (D.10) that

$$\mathbb{E}_{S,I,A}\big[\|\nabla F_S(\mathbf{w}_t)\|^2\big] \leq 2\mathbb{E}_{S,I,A}\big[\|\nabla F_S(\mathbf{w}_t)-\nabla F_S(\mathbf{w}^*)\|^2\big]+\frac{4LF(\mathbf{w}^*)}{mn}. \tag{D.11}$$

Plugging Eq. (D.11) into Eq. (D.9), we obtain

$$2\mathbb{E}_{S,I,A}\Big[\sum_{t=2}^{T}\eta_{t+\widehat{\tau}_t}\big(F_S(\mathbf{w}_t)-F(\mathbf{w}^*)\big)\Big] \leq \mathbb{E}_{S,I,A}[\|\widehat{\mathbf{w}}_2-\mathbf{w}^*\|^2]+4LF(\mathbf{w}^*)\sum_{t=2}^{T}\eta_{t+\widehat{\tau}_t}^2$$

$$+\mathbb{E}_{S,I,A}\Big[\sum_{t=2}^{T}\big(2(\widehat{\tau}_t+m-1)\eta_{t+\widehat{\tau}_t}-\frac{1}{2L}\big)\eta_{t+\widehat{\tau}_t}\|\nabla F_S(\mathbf{w}_t)-\nabla F_S(\mathbf{w}^*)\|^2\Big]+\frac{4LF(\mathbf{w}^*)}{mn}\sum_{t=2}^{T}\eta_{t+\widehat{\tau}_t}^2(\widehat{\tau}_t+m-1)$$

$$+\mathbb{E}_{S,I,A}\Big[\sum_{t=2}^{T}\big(2\eta_{t+\widehat{\tau}_t}-\frac{1}{2L}\big)\eta_{t+\widehat{\tau}_t}\|\nabla f(\mathbf{w}_t;S_{i_{t+\widehat{\tau}_t},a_t})-\nabla f(\mathbf{w}^*;S_{i_{t+\widehat{\tau}_t},a_t})\|^2\Big], \tag{D.12}$$

where we have used

$$\mathbb{E}_{S,I,A}[\|\nabla f(\mathbf{w}_t;S_{i_{t+\widehat{\tau}_t},a_t})\|^2] \leq 2\mathbb{E}_{S,I,A}\Big[\|\nabla f(\mathbf{w}_t;S_{i_{t+\widehat{\tau}_t},a_t})-\nabla f(\mathbf{w}^*;S_{i_{t+\widehat{\tau}_t},a_t})\|^2\Big]+2\mathbb{E}_{S,I,A}\Big[\|\nabla f(\mathbf{w}^*;S_{i_{t+\widehat{\tau}_t},a_t})\|^2\Big]$$

$$\leq 2\mathbb{E}_{S,I,A}\Big[\|\nabla f(\mathbf{w}_t;S_{i_{t+\widehat{\tau}_t},a_t})-\nabla f(\mathbf{w}^*;S_{i_{t+\widehat{\tau}_t},a_t})\|^2\Big]+4LF(\mathbf{w}^*), \tag{D.13}$$

where the second inequality holds by Lemma A.4. Recall that setting $t = k+\widehat{\tau}_k$ implies $k = t-\tau_t$. Under the assumption that $(\tau_t+m)\eta_t \leq 1/(4L)$, or equivalently $(\widehat{\tau}_k+m)\eta_{k+\widehat{\tau}_k} \leq 1/(4L)$, Eq. (D.12) indicates that

$$2\mathbb{E}_{S,I,A}\Big[\sum_{t=2}^{T}\eta_{t+\widehat{\tau}_t}\big(F_S(\mathbf{w}_t)-F(\mathbf{w}^*)\big)\Big] \leq \mathbb{E}_{S,I,A}[\|\widehat{\mathbf{w}}_2-\mathbf{w}^*\|^2]+4LF(\mathbf{w}^*)\sum_{t=2}^{T}(\frac{\widehat{\tau}_t+m-1}{mn}+1)\eta_{t+\widehat{\tau}_t}^2. \tag{D.14}$$

Recall the definition of $\widehat{\mathbf{w}}_2$. The following inequality holds

$$\mathbb{E}_{S,I,A}[\|\widehat{\mathbf{w}}_2 - \mathbf{w}^*\|^2] = \mathbb{E}_{S,I,A}\Big[\big\|\mathbf{w}_1 - \mathbf{w}^* - \sum_{j=1}^{m}\eta_{1+\widehat{\tau}(1,j)}\nabla f(\mathbf{w}_t; S_{i_{1+\widehat{\tau}(1,j)},j})\big\|^2\Big]$$

$$\leq 2\|\mathbf{w}_1 - \mathbf{w}^*\|^2 + 2m\sum_{j=1}^{m}\mathbb{E}_{S,I,A}\big[\eta_{1+\widehat{\tau}(1,j)}^2\|\nabla f(\mathbf{w}_1; S_{1+\widehat{\tau}_{1,j},j})\|^2\big]$$

$$\leq 2\|\mathbf{w}_1 - \mathbf{w}^*\|^2 + \frac{1}{4L^2m}\sum_{j=1}^{m}\mathbb{E}_{S,I,A}\Big[\big\|\nabla f(\mathbf{w}_1; S_{i_{1+\widehat{\tau}_{1,j}},j}) - \nabla f(\mathbf{w}^*; S_{i_{1+\widehat{\tau}_{1,j}},j})\big\|^2\Big] + \frac{1}{2L}F(\mathbf{w}^*)$$

$$\lesssim \|\mathbf{w}_1 - \mathbf{w}^*\|^2 + F(\mathbf{w}^*), \tag{D.15}$$

where the first inequality holds by $(\sum_{i=1}^{m}a_i)^2 \leq m\sum_{i=1}^{m}a_i^2$, the second inequality holds by the assumption $m\eta_t \leq 1/(4L)$ and a decomposition analogous to Eq. (D.13), and the third inequality holds due to the $L$-smoothness of $f$. Plugging Eq. (D.15) into Eq. (D.14) and ignoring the constants including $L$, we derive

$$\sum_{t=2}^{T}\eta_{t+\widehat{\tau}_t}\mathbb{E}_{S,I,A}\Big[F_S(\mathbf{w}_t) - F(\mathbf{w}^*)\Big] \lesssim \|\mathbf{w}_1 - \mathbf{w}^*\|^2 + F(\mathbf{w}^*)\Big(\sum_{t=2}^{T}(\frac{\widehat{\tau}_t + m - 1}{mn} + 1)\eta_{t+\widehat{\tau}_t}^2 + 1\Big).$$

Finally, we consider the case where the step size is constant and the delays are bounded by $\tau_M$. The number of appearance of $(a_t, t)$ in $\bigcup_{i=2}^{T}\mathcal{G}_i$ is less or equal to $\tau_M$, which implies that Eq. (D.7) is replaced by

$$\sum_{t=2}^{T}\mathbb{E}_{S,I,A}\Big[\eta\langle\widehat{\mathbf{w}}_t - \mathbf{w}_t, \nabla f(\mathbf{w}_t; S_{i_{t+\widehat{\tau}_t},a_t})\rangle\Big] \geq -\mathbb{E}_{S,I,A}\Big[\sum_{t=2}^{T}\frac{\tau_M + m - 1}{2}\eta^2\|\nabla F_S(\mathbf{w}_t)\|^2\Big]. \tag{D.16}$$

Taking $\eta_t = \eta \leq 1/(4L(\tau_M + m))$, we have

$$\sum_{t=2}^{T}\eta\mathbb{E}_{S,I,A}\Big[F_S(\mathbf{w}_t) - F(\mathbf{w}^*)\Big] \lesssim \|\mathbf{w}_1 - \mathbf{w}^*\|^2 + F(\mathbf{w}^*)\Big((\frac{\tau_M + m - 1}{mn} + 1)T\eta^2 + 1\Big).$$

The proof is complete. □

## D.2. Proof of Theorem 6.2

Before proving Theorem 6.2, we first present a more general result concerning the excess risk of random ASGD.

**Theorem D.1.** *Let the conditions of Theorem 4.1 and Lemma 6.1 hold. If we take a delay-adaptive step size, then for any $\gamma > 0$,*

$$\mathbb{E}_{S,I,A}[F(A(S))] - F(\mathbf{w}^*) \lesssim \Big(\frac{1+\gamma}{\gamma\sum_{t=2}^{T}\eta_{t+\widehat{\tau}_t}} + \frac{\gamma\eta_M}{mn}\big(1 + \frac{\mathbb{E}_A[N_T^M]}{n}\big)\Big)\|\mathbf{w}_1 - \mathbf{w}^*\|^2$$

$$+ \Big(\frac{1}{\gamma} + \frac{1+\gamma}{\gamma\sum_{t=2}^{T}\eta_{t+\widehat{\tau}_t}} + \frac{1}{mn}\Big)F(\mathbf{w}^*) + \frac{\gamma T\eta_M^2}{mn}\big(1 + \frac{\mathbb{E}_A[N_T^M]}{n}\big)\widehat{F}(\mathbf{w}^*), \tag{D.17}$$

*where $A(S) = \frac{\sum_{t=2}^{T}\eta_{t+\widehat{\tau}_t}\mathbf{w}_t}{\sum_{t=2}^{T}\eta_{t+\widehat{\tau}_t}}$.*

*Proof of Theorem D.1.* Consider the delay-adaptive step size. For any $t \in \{2, \cdots, T\}$, let $A(S) = \mathbf{w}_t$. We adopt Lemma 3.3 to the whole dataset, and consider the randomness of $I$ and $A$ and obtain that for any $\gamma > 0$,

$$\mathbb{E}_{S,I,A}[F(\mathbf{w}_t) - F_S(\mathbf{w}_t)] \leq \frac{L+\gamma}{2mn}\sum_{j=1}^{m}\sum_{i=1}^{n}\mathbb{E}_{S,\tilde{S},I,A}[\|\mathbf{w}_t^{(ij)} - \mathbf{w}_t\|_2^2] + \frac{L}{\gamma}\mathbb{E}_{S,I,A}[F_S(\mathbf{w}_t)].$$

Note that delays are independent to workers and therefore independent to $A$. Taking a weighted average of the above inequality, we derive

$$\frac{1}{\sum_{t=2}^{T}\eta_{t+\widehat{\tau}_t}}\sum_{t=2}^{T}\eta_{t+\widehat{\tau}_t}\mathbb{E}_{S,I,A}\Big[F(\mathbf{w}_t)-F_S(\mathbf{w}_t)\Big] \leq \frac{L}{\gamma\sum_{t=2}^{T}\eta_{t+\widehat{\tau}_t}}\sum_{t=2}^{T}\eta_{t+\widehat{\tau}_t}\mathbb{E}_{S,I,A}[F(\mathbf{w}_t)]+$$

$$\frac{1}{\sum_{t=2}^{T}\eta_{t+\widehat{\tau}_t}}\sum_{t=2}^{T}\eta_{t+\widehat{\tau}_t}\Big(\frac{L+\gamma}{2mn}\sum_{j=1}^{m}\sum_{i=1}^{n}\mathbb{E}_{S,\tilde{S},I,A}[\|\mathbf{w}_t^{(ij)}-\mathbf{w}_t\|_2^2]\Big). \quad \text{(D.18)}$$

**First**, we extend Theorem 4.1 to the case of heterogeneous data. We know that for any $j \in [m]$ and $t \in [T]$, $\mathbf{w}_t$ is $\epsilon_{stab,j}$-stable w.r.t. subset $S_j$ where

$$\epsilon_{stab,j}^2 = \frac{1}{n}\sum_{i=1}^{n}\mathbb{E}_{S,\tilde{S},I}[\|\mathbf{w}_t^{(ij)}-\mathbf{w}_t\|_2^2] \leq \frac{32L}{n}\Big(1+\frac{N_T^M}{n}\Big)\sum_{k=1}^{T}\eta_k^2\mathbb{I}_{[m_k=j]}\mathbb{E}_{S,I}\big[F_{S_j}(\mathbf{w}_{k-\tau_k})\big].$$

Taking summation over $j$ from 1 to $m$ to the above inequality, we obtain

$$\frac{1}{mn}\sum_{j=1}^{m}\sum_{i=1}^{n}\mathbb{E}_{S,\tilde{S},I}[\|\mathbf{w}_t^{(ij)}-\mathbf{w}_t\|_2^2] \leq \frac{32L}{mn}\Big(1+\frac{N_T^M}{n}\Big)\sum_{k=1}^{T}\eta_k^2\mathbb{E}_{S,I}\big[F_{S_{m_k}}(\mathbf{w}_{k-\tau_k})\big]. \quad \text{(D.19)}$$

**Second**, we consider Lemma 4.4. Compared to the homogeneous setting, two things are different when extending Lemma 4.4 to a heterogeneous data setting. First, we have $\mathbb{E}_{S,I}[f(\mathbf{w}^*;S_{i_t,m_t})] = F_{m_t}(\mathbf{w}^*)$ instead of $\mathbb{E}_{S,I}[f(\mathbf{w}^*;S_{i_t,m_t})] = F(\mathbf{w}^*)$. Second, we have $\mathbb{E}_{S,I}\big[\|\nabla F_{S_{m_t}}(\mathbf{w}^*)\|^2\big] \leq 2LF_{m_t}(\mathbf{w}^*)$ instead of $\mathbb{E}_{S,I}\big[\|\nabla F_{S_{m_t}}(\mathbf{w}^*)\|^2\big] \leq \frac{2LF(\mathbf{w}^*)}{n}$. Then, we follow the proof of Lemma 4.4 and get

$$\sum_{t=1}^{T}\eta_t\mathbb{E}_{S,I}[F_{S_{m_t}}(\mathbf{w}_{t-\tau_t})-F_{m_t}(\mathbf{w}^*)] \leq \frac{1}{2}\|\mathbf{w}_1-\mathbf{w}^*\|^2 + 2L\sum_{t=1}^{T}(\tau_t+m)\eta_t^2F_{m_t}(\mathbf{w}^*),$$

which further implies that

$$\sum_{t=1}^{T}\eta_t\mathbb{E}_{S,I}[F_{S_{m_t}}(\mathbf{w}_{t-\tau_t})] \leq \frac{1}{2}\|\mathbf{w}_1-\mathbf{w}^*\|^2 + \frac{3}{2}\sum_{t=1}^{T}\eta_t\hat{F}(\mathbf{w}^*), \quad \text{(D.20)}$$

where we denote $\hat{F}(\mathbf{w}^*) := \max_{j\in[m]}F_j(\mathbf{w}^*)$. Plugging Eq. (D.20) into Eq. (D.19) followed by an expectation w.r.t $A = \{a_t\}$ yields

$$\frac{1}{mn}\sum_{j=1}^{m}\sum_{i=1}^{n}\mathbb{E}_{S,\tilde{S},I,A}[\|\mathbf{w}_t^{(ij)}-\mathbf{w}_t\|_2^2] \leq \frac{32L\eta_M}{mn}\Big(1+\frac{\mathbb{E}_A[N_T^M]}{n}\Big)\Big(\frac{1}{2}\|\mathbf{w}_1-\mathbf{w}^*\|^2+\frac{3}{2}\sum_{t=1}^{T}\eta_t\hat{F}(\mathbf{w}^*)\Big), \quad \text{(D.21)}$$

where $\eta_M$ is the maximum step size. According to Lemma 6.1, we have

$$\frac{1}{\sum_{t=2}^{T}\eta_{t+\widehat{\tau}_t}}\sum_{t=2}^{T}\eta_{t+\widehat{\tau}_t}\mathbb{E}_{S,I,A}\Big[F_S(\mathbf{w}_t)-F(\mathbf{w}^*)\Big] \lesssim \frac{\|\mathbf{w}_1-\mathbf{w}^*\|^2}{\sum_{t=2}^{T}\eta_{t+\widehat{\tau}_t}} + \frac{F(\mathbf{w}^*)}{\sum_{t=2}^{T}\eta_{t+\widehat{\tau}_t}}\Big(\sum_{t=2}^{T}(\frac{\widehat{\tau}_t+m-1}{mn}+1)\eta_{t+\widehat{\tau}_t}^2+1\Big). \quad \text{(D.22)}$$

Plugging Eq. (D.21) and Eq. (D.22) into Eq. (D.18) yields the following generalization error bounds of $A(S) = \frac{\sum_{t=2}^{T}\eta_{t+\widehat{\tau}_t}\mathbf{w}_t}{\sum_{t=2}^{T}\eta_{t+\widehat{\tau}_t}}$

$$\mathbb{E}_{S,I,A}\big[F(A(S))-F_S(A(S))\big] \lesssim \frac{1}{\gamma\sum_{t=2}^{T}\eta_{t+\widehat{\tau}_t}}\Big(\|\mathbf{w}_1-\mathbf{w}^*\|^2 + \big(\sum_{t=2}^{T}\eta_{t+\widehat{\tau}_t}+1\big)F(\mathbf{w}^*)\Big)$$

$$+ \frac{(1+\gamma)\eta_M}{mn}\Big(1+\frac{\mathbb{E}_A[N_T^M]}{n}\Big)\Big(\|\mathbf{w}_1-\mathbf{w}^*\|^2+\sum_{t=1}^{T}\eta_t\hat{F}(\mathbf{w}^*)\Big), \quad \text{(D.23)}$$

where we have used the inequality $\sum_{t=2}^{T} (\frac{\hat{\tau}_t + m - 1}{mn} + 1) \eta_{t+\hat{\tau}_t}^2 \lesssim \sum_{t=2}^{T} \eta_{t+\hat{\tau}_t}$. Combining Eq. (D.22) and Eq. (D.23) and ignoring the constants including $L$, we obtain following excess risk bounds

$$\mathbb{E}_{S,I,A}[F(A(S))] - F(\mathbf{w}^*) \lesssim \Big( \frac{1+\gamma}{\gamma \sum_{t=2}^{T} \eta_{t+\hat{\tau}_t}} + \frac{\gamma \eta_M}{mn} \big(1 + \frac{\mathbb{E}_A[N_T^M]}{n}\big) \Big) \|\mathbf{w}_1 - \mathbf{w}^*\|^2$$
$$+ \Big( \frac{1}{\gamma} + \frac{1+\gamma}{\gamma \sum_{t=2}^{T} \eta_{t+\hat{\tau}_t}} + \frac{1}{mn} \Big) F(\mathbf{w}^*) + \frac{\gamma T \eta_M^2}{mn} \big(1 + \frac{\mathbb{E}_A[N_T^M]}{n}\big) \hat{F}(\mathbf{w}^*).$$

The proof is complete. $\qquad\square$

Theorem 6.2 is a direct corollary of Theorem D.1.

*Proof of Theorem 6.2.* **First, we consider the constant step size**. Following the proof of Eq. (D.17), one can show that

$$\mathbb{E}_{S,I,A}[F(A(S))] - F(\mathbf{w}^*) \lesssim \Big( \frac{1+\gamma}{\gamma T \eta} + \frac{\gamma \eta}{mn} \big(1 + \frac{\mathbb{E}_A[N_T^M]}{n}\big) \Big) \|\mathbf{w}_1 - \mathbf{w}^*\|^2$$
$$+ \Big( \frac{1}{\gamma} + \frac{1+\gamma}{\gamma T \eta} + \frac{1}{mn} \Big) F(\mathbf{w}^*) + \frac{\gamma T \eta^2}{mn} \big(1 + \frac{\mathbb{E}_A[N_T^M]}{n}\big) \hat{F}(\mathbf{w}^*).$$

Taking $\gamma = \sqrt{mn}$, $T \asymp mn$ and $\eta \asymp 1/\sqrt{T}$ yields Eq. (6.3).

**Second, we consider the adaptive step size.** As illustrated in the proof of Corollary 4.8, choosing $T \asymp mn$ and the delay-adaptive step size as

$$\eta_t = \begin{cases} \min\{\frac{1}{\sqrt{T}}, \frac{1}{12mL}\}, & \text{if } \tau_t \leq m, \\ \min\{\frac{1}{\sqrt{T}}, \frac{1}{4L((1+\frac{m}{n})\tau_t + m)}\}, & \text{if } \tau_t > m, \end{cases}$$

then it holds that $\sum_{t=1}^{T} \eta_t \geq \frac{\sqrt{T}}{2}$. Thus, we have

$$\sum_{t=2}^{T} \eta_{t+\hat{\tau}_t} \geq \sum_{t=1}^{T} \eta_t - \eta_1 - \sum_{(a_t,t) \in \mathcal{A}_1} \eta_{t+\hat{\tau}_t} \geq \frac{\sqrt{T}}{2} - \frac{m+1}{\sqrt{T}} \gtrsim \sqrt{T},$$

where the first inequality holds by the definition of $\hat{\tau}_t$, and the second inequality holds since $|\mathcal{A}_1| = m - 1$ and the last inequality holds by $\frac{m+1}{\sqrt{T}} \lesssim \frac{m+1}{m} \asymp 1$. Further taking $\gamma \asymp \sqrt{mn}$, Eq. (6.3) implies that

$$\mathbb{E}_{S,I,A}[F(A(S))] - F(\mathbf{w}^*) \lesssim \Big( \frac{1}{\sqrt{T}} + \frac{1}{\sqrt{mnT}} \big(1 + \frac{T}{n}\big) \Big) \|\mathbf{w}_1 - \mathbf{w}^*\|^2 + \Big( \frac{1}{\sqrt{mn}} + \frac{1}{\sqrt{T}} + \frac{1}{mn} \Big) F(\mathbf{w}^*)$$
$$+ \frac{1}{\sqrt{mn}} \big(1 + \frac{\mathbb{E}_A[N_T^M]}{n}\big) \hat{F}(\mathbf{w}^*) \lesssim \frac{1}{\sqrt{mn}} \Big( \|\mathbf{w}_1 - \mathbf{w}^*\|^2 + F(\mathbf{w}^*) + \big(1 + \frac{\mathbb{E}_A[N_T^M]}{n}\big) \hat{F}(\mathbf{w}^*) \Big).$$

The proof is complete. $\qquad\square$

# E. Experimental Details and Supplementary Experiments

## E.1. Experimental Details

All experiments were conducted on a single NVIDIA T4 GPU provided by Google Colab. We implemented the distributed training framework and executed ASGD using the Ray and Pytorch packages in Python. Experiments were carried out for smooth, non-smooth and non-convex optimization problems. All experiments were repeated over ten trials with different random seeds, and the results were averaged to reduce randomness. To ensure reproducibility, our code is publicly available at `https://github.com/xxyufeng/Stability-and-Generalization-of-ASGD`.

For the smooth setting, we trained a linear classifier with cross-entropy loss on the MNIST dataset, which contains 60,000 training samples of dimension $32 \times 32$ from 10 classes. For the non-smooth setting, we consider the $q$-norm hinge loss, whose

gradients are $(q-1, L)$-Hölder continuous, to train a linear model on the RCV1 dataset from the LIBSVM repository (Chang & Lin, 2011). The $q$-norm hinge loss is defined as

$$l_q(\mathbf{w}; \mathbf{z}) = \big( \max\{0, 1 - y\langle\mathbf{w}, \mathbf{x}\rangle\}\big)^q,$$

where $y \in \{-1, 1\}$. Here, for simplicity, we used 1-norm hinge loss for model training. The RCV1 dataset has dimension 47,236 and consists of binary-labeled samples. For the non-convex setting, we train a three-layer fully connected neural network with cross-entropy loss on MNIST.

To study the dependence of stability and excess risk on the number of workers, we varied the number of workers in $\{1, 2, \ldots, 2^6\}$. For MNIST, we assigned 500 samples to each worker and ran ASGD for 20,000 iterations with a constant step size $\eta = 2 \times 10^{-3}$, selected via grid search. For RCV1, ASGD was run for 30,000 iterations with step size $\eta = 1 \times 10^{-2}$. To estimate stability (left panel of Figure 1, Figure 2 and Figure 3), we constructed 5 neighboring datasets and computed the average model difference, approximating the on-average model stability defined in Definition 3.1. For the test error and accuracy results on RCV1 (middle and right panels of Figure 2), we randomly sampled 10,000 test samples due to computational constraints.

To evaluate the effect of worker participation (right panel of Figure 1, Figure 2 and Figure 3), we simulated straggling behavior by forcing all but one fast worker to sleep for $\{50, 150, 250, 350, 450, 550\}$ milliseconds (ms) after pulling gradients from the server. To simulate heterogeneous data settings, we partitioned the training dataset by labels, assigning each worker samples from a single class, which is common in horizontal federated learning scenarios (Xu et al., 2023).

### E.2. Supplementary Experiments

As shown in Figure 2 and Figure 3, we observed similar generalization behavior in both non-smooth and non-convex settings. The left and middle panels of Figure 2 and Figure 3 illustrate that increasing the number of workers improves the stability of ASGD during training and reduces the test error, which is consistent with the results of Section 5. The right panel of Figure 2 and Figure 3 show that as the imbalance measure $N_T^M$ increases, the test error of ASGD degrades due to uneven worker participation. In contrast, random ASGD maintains better worker balance and achieves superior generalization performance. These empirical results support the theoretical analysis presented in Sections 5 and extend the analysis in Section 6 to non-smooth problems.

We also compare the stability evolution of five different ASGD variants (Koloskova et al., 2022; Islamov et al., 2024): vanilla ASGD, random ASGD, vanilla ASGD with waiting, random ASGD with waiting, and shuffle ASGD. We train a linear classifier on MNIST with $m \in \{4, 8, 16\}$. As shown in Figure 4, all variants exhibit similar stability evolution. Also, We evaluate ASGD variants on heterogeneous data with stragglers. We use 10 workers and simulate stragglers by forcing 6 workers to sleep for $\{50, 150, 250, 350, 450, 550\}$ ms. The buffer size is set to 4 for variants with waiting. As shown in Figure 5b and Figure 5c, increasing sleep time raises the test error of vanilla ASGD and vanilla ASGD with waiting, while reducing the worker participation index $T/N_T^M$. In contrast, other variants remain stable. The alignment between test error and participation highlights the importance of balanced worker participation, especially in heterogeneous settings. In shuffle ASGD, jobs are assigned via random permutations, enforcing $T/N_T^M \asymp m$, i.e., $\sum_{t=1}^{T} \mathbb{E}[\mathbb{I}_{a_t=j}] \asymp T/m, \forall j \in [m]$, which explains its similar performance to random ASGD. These findings support our results in Section 6.

Finally, we provide empirical evidence for the minimax-optimal rate $O(1/\sqrt{mn})$ by showing that $\sqrt{mn} \cdot \mathbb{E}[F(A(S)) - F(\mathbf{w}^*)]$ remains nearly constant across different $m$ in Figure 5a. Here, we set $T \asymp mn$ as in Corollary 4.8 and select a best constant step size via grid search.

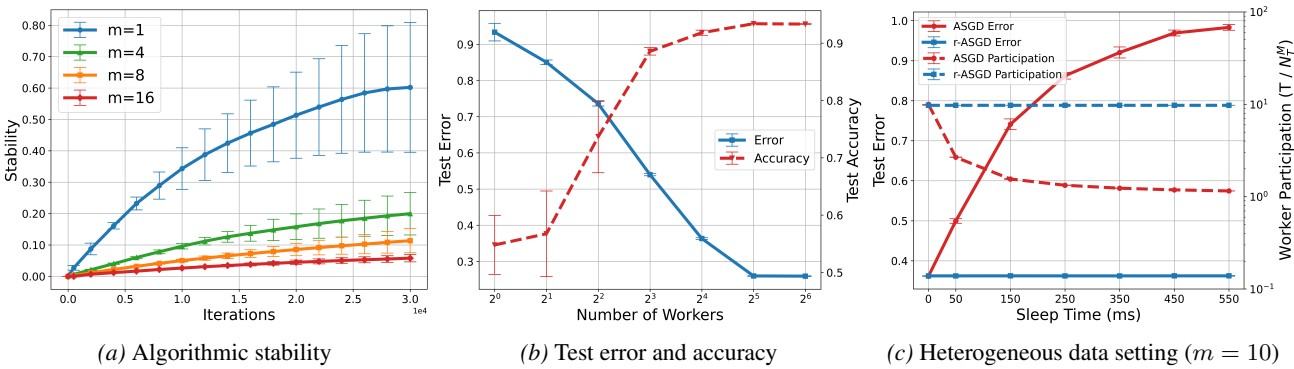

*(a)* Algorithmic stability      *(b)* Test error and accuracy      *(c)* Heterogeneous data setting ($m = 10$)

*Figure 2.* Experiment for non-smooth problems. Left panel: evolution of algorithmic stability of ASGD with varying numbers of workers. Middle panel: impact of worker numbers on testing error and accuracy. Right panel: effects of straggling behavior on worker participation and testing errors.

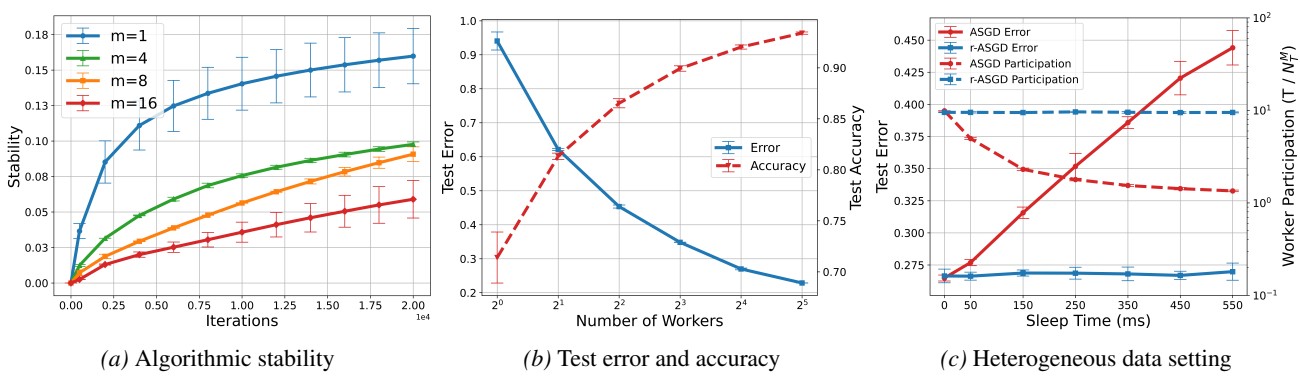

*(a)* Algorithmic stability      *(b)* Test error and accuracy      *(c)* Heterogeneous data setting

*Figure 3.* Experiment for nonconvex problems.

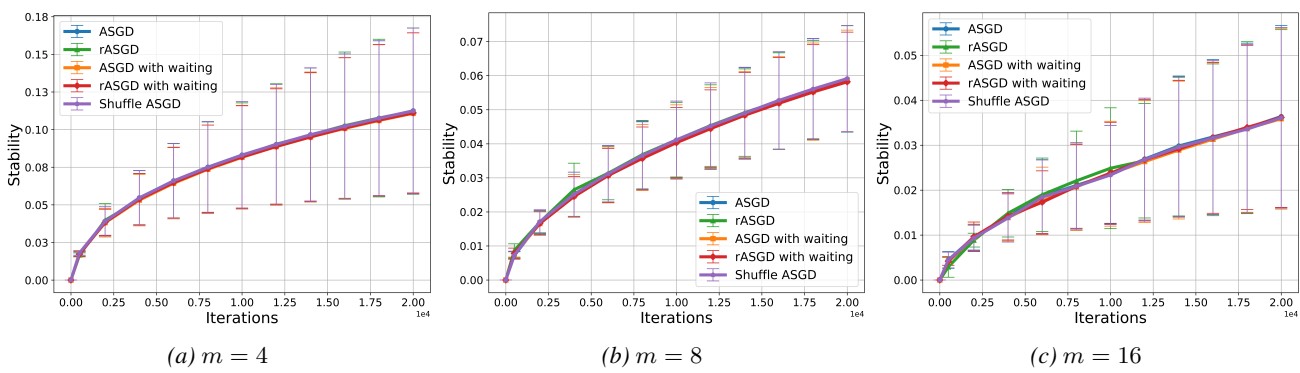

*(a)* $m = 4$      *(b)* $m = 8$      *(c)* $m = 16$

*Figure 4.* Stability evolution of ASGD vairants under a different number of workers ($m = 4, 8, 16$).

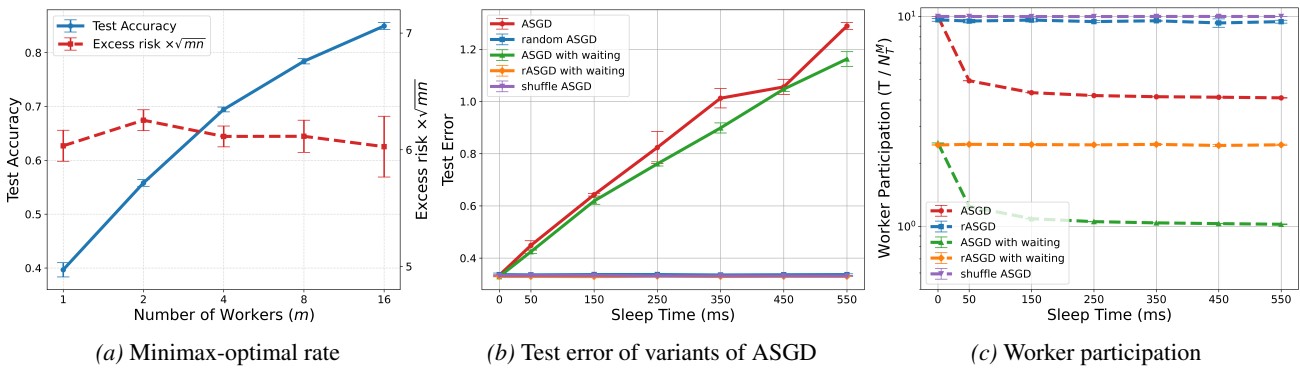

*(a)* Minimax-optimal rate      *(b)* Test error of variants of ASGD      *(c)* Worker participation

*Figure 5.* Experiment on MNIST. Left panel: effects of $m$ on test accuracy and excess risk; the flat red curve shows $\sqrt{mn}(\mathbb{E}[F(A(S)) - F(\mathbf{w}^*)]) \asymp 1$. Middle panel: effect of straggling on test error for ASGD variants. Right panel: effect of straggling on worker participation; $T/N_T^M$ reflects the constant $k$ defined in Corollary 4.8.

