# OpenReview forum: "Sharper Generalization Guarantees for Asynchronous SGD: Beyond Lipschitzness, Smoothness and Data Homogeneity"
_ICML.cc/2026/Conference — ICML 2026 regular_

### Official Review · Reviewer_4VXN · 2026-03-07

**Soundness:** 3
**Presentation:** 3
**Significance:** 3
**Originality:** 2
**Overall Recommendation:** 4
**Confidence:** 4

**Summary:**

The paper studies the generalization of asynchronous SGD through on-average model stability. Its main claim is that, for convex smooth objectives, ASGD admits sharper stability and excess-risk guarantees under weaker assumptions than prior analyses: no Lipschitzness, no bounded noise, no bounded parameter or data domain, and support for partitioned local datasets with arbitrary delays. It further extends the framework to non-smooth objectives via Holder-continuous gradients and to heterogeneous-data settings through a random-ASGD variant. The paper argues that the smooth-case bounds depend on worker participation rather than delay magnitude, recover the balanced-regime $O((mn)^{-1/2})$ excess-risk rate, and imply faster $O(1/n)$-type behavior under low noise. Experiments on MNIST in the main paper and RCV1 in Appendix E are used to support the qualitative trends predicted by the theory.

**Compliance With Llm Reviewing Policy:**

Affirmed.

**Final Justification:**

---
### Original justification:

I maintain 3 (weak reject). While the smooth partitioned-data result is technically solid, the rebuttal does not fully resolve my main concerns about the heterogeneous extension: random ASGD is still insufficiently motivated, and the discussion of bounded $\tau\_{max}$ remains shallow, leaving the result effectively constrained by worst-straggler behavior unless extra assumptions are added.

---
### After second-round rebuttal:
The authors have successfully addressed my concerns about the motivation of ASGD, and tightened the error bound by removing the dependency on $\tau\_{\max}$. Therefore, I raised the score from 3 to 4 (weak accept).

**Key Questions For Authors:**

1. The smooth-case contribution seems the main novelty. Can the authors isolate more clearly which part of the proof is fundamentally new relative to Deng et al. (2024), and which parts are extensions of the same on-average-stability machinery?
2. For the heterogeneous-data result, why is random ASGD the right extension point rather than the original ASGD formulation? Is the bounded-$\tau_{\max}$ assumption in Lemma 6.1 only a technical proof device, or is there a concrete obstruction preventing an arbitrary-delay heterogeneous-data theorem of the same flavor as Sec. 4?
3. The experiments mainly validate qualitative monotonic trends. Can the authors provide stronger empirical evidence for the specific rate claims, especially the balanced-regime $O((mn)^{-1/2})$ behavior and the low-noise fast-rate discussion? As currently written, the experiments do not directly test those asymptotic predictions.

**Limitations:**

No. The paper includes a brief impact statement, but the limitations discussion is minimal apart from noting nonconvex extension as future work. In particular, the authors should explicitly discuss: the restriction to convex objectives; the fact that the heterogeneous-data result is for random ASGD with bounded $\tau_{\max}$, not the full arbitrary-delay setting; the narrow empirical scope; and plausible deployment risks in distributed & federated systems, such as skewed resource access across workers and how system heterogeneity or data imbalance could worsen performance disparities.

**Strengths And Weaknesses:**

Strengths:
- The paper’s main technical value is the partitioned-data smooth analysis, where the authors explicitly identify the mismatch $\nabla F_{S_{m_t}}(\omega) \neq \nabla F_{S}(\omega)$ and introduce a new error decomposition to control the resulting bias. The smooth-case theory is interpretable: Theorem 4.1 and Corollary 4.7 tie stability and excess risk to worker participation $N_T^M$ and recover $O((mn)^{-1/2})$ in balanced regimes, with faster behavior in the low-noise case.
- Beyond the smooth case, the manuscript also covers non-smooth objectives and a heterogeneous-data extension via random ASGD, so the paper is broader than a one-result note. The paper is generally well typeset: heading hierarchy, theorem formatting, and caption placement are consistent, and Fig. 1 & 2 use parallel layouts that make the smooth and non-smooth experiments easy to compare.

---

Weaknesses:
- Novelty is incremental relative to the immediate works [1, 2]:
    -  The paper reads more as an incremental improvement over an existing on-average stability criterion beyond Lipschitzness than a new conceptual direction. The non-smooth Holder extension, in particular, overlaps strongly with themes already highlighted publication [1].
    -  The heterogeneous-data section looks closer to a synthesis of existing convergence results than to a fully new proof framework: the manuscript explicitly states that Lemma 6.1 *“extends the convergence results in [3]”*, *“retains some notations from [4]”*, and Appendix D adopts a perturbed-iterate analysis; these choices closely mirror Koloskova’s average-delay perspective and AsGrad’s formalism $(A_t, R_t)$ for random ASGD. As a result, beyond the partitioned-data bias decomposition in Sec. 4, a noticeable portion of the proof architecture reads as a recombination of prior tools rather than a fundamentally new analytical approach.
- Experimental support is less convincing:
    - The main paper’s experimental evidence is limited to MNIST with a linear classifier and cross-entropy loss; the non-smooth results appear only in Appendix E on RCV1 with a linear model and 1-norm hinge loss.
    - No direct empirical comparison against prior ASGD baselines, e.g., [2, 4].

---
[1]. Deng, X., Sun, T., Li, S., Li, D., and Lu, X. Stability and generalization of asynchronous sgd: Sharper bounds beyond Lipschitz and smoothness. Advances in Neural Information Processing Systems, 37:7675–7713, 2024.

[2]. Deng, X., Shen, L., Li, S., Sun, T., Li, D., and Tao, D. Towards understanding the generalizability of delayed stochastic gradient descent. IEEE Transactions on Pattern Analysis and Machine Intelligence, 2025.

[3]. Koloskova, A., Stich, S. U., and Jaggi, M. Sharper convergence guarantees for asynchronous sgd for distributed and federated learning. Advances in Neural Information Processing Systems, 35:17202–17215, 2022.

[4]. Islamov, R., Safaryan, M., and Alistarh, D. Asgrad: A sharp unified analysis of asynchronous-sgd algorithms. International Conference on Artificial Intelligence and Statistics, pp. 649–657. PMLR, 2024.

---

> ### Author Rebuttal · Authors · 2026-03-31
>
> We sincerely thank you for your constructive suggestions and valuable comments. Full references, tables, and experiments are available at: https://anonymous.4open.science/r/ICML_rebuttal-23152/Rebuttal.pdf.
>
> **Q1:** Novelty compared to [4].
> **A1:**
> *Techniques.*
> 1. We introduce a new stability w.r.t. subsets and a new decomposition (Remark 4.6), yielding risk bounds without data homogeneity assumptions.
> 2.  By better controlling the accumulation of errors caused by delays, we improve the stability bounds in [4] and obtain a minimax excess risk bound, which cannot be improved. The detailed technique can be seen in the **Rebuttal "A3" to Reviewer cYvC**.
> 3. We study a realistic heterogeneous setting with worker participation index $N_T^M$ capturing effect of asynchrony on generalization.
>
> *Results.*
> 1. Unlike [4], which requires Lipschitzness, bounded-space and low-noise assumptions, we remove Lipschitzness and bounded-space assumptions and allow noisy settings.
> 2. For smooth problems, [4] shows $O(1/(mn)+1/\bar\tau)$ with $\bar\tau\le (mn)^{1/4}$, implying at most $O((mn)^{-1/4})$. We prove a minimax $O(1/\sqrt{mn})$ when $N_T^M\asymp m$, and $O(1/\sqrt{kn})$ with $k=T/N_T^M$ in the imbalanced case, showing that the worker participation plays an important role in the effect of asynchrony on generealization performance rather than the delays itself.
> 3. For non-smooth problems, [4] gives worse than $O(n^{1/6})$, while we obtain $O(1/\sqrt{mn})$ under balanced training and extend to heterogeneous data.
>
> **Q2:** What is the novelty comparing to [6]? Can dependency on $\tau_{max}$ be removed?
>
> **A2:**
> 1. [6] only considered the optimization of ASGD, i.e., the convergence of training errors. We instead study the generalization behaviour of ASGD on heterogeneous data setting, i.e., how the training behavior generalizes to the test datasets. Since the generalization results (Lemma 3.3, Theorem 4.1) also hold for heterogeneous data, we only show the convergence result in Section 6.
> 2. For the convergence rate (Lemma 6.1), [6] made an assumption on data heterogeneity, i.e. $\\|\nabla F_{S_i}(\cdot)-\nabla F_{S}(\cdot)\\|\leq \zeta_i^2$. We, instead, do not make any assumption on data heterogeneity and still get a similar convergence rate of $O((1+\frac{\tau_{max}}{mn})/\sqrt{T})$ by taking $\eta\asymp 1/\sqrt{T}$.
> 3. The dependence on $\tau_{max}$ can be removed by additionally assuming bounded gradients together with data heterogeneity [5,6]. Another possible routeline is to do the following decomposition:
>     $$
> F(w_{t-\tau_t}) - F(w^\*)= (F(w_{t-\tau_t}) - F_{S_{m_t}}(w_{t-\tau_t})) + (F_{S_{m_t}}(w_{t-\tau_t})-F_{S_{m_t}}(w^\*)) + (F_{S_{m_t}}(w^\*)-F(w^\*)).
>     $$
>     With this decomposition, we can combine stability and optimization bound shown in Eq. (D.18) to derive an excess risk bound without dependency on $\tau_{max}$. When $F_{S_{m_t}}(w^\*)-F(w^\*)$ is tightly bounded, we can also get a non-vacuous excess risk bound.
>
> **Q3:** Experimental support.
>
> **A3:**  We've conducted the following supplemental experiments to support  theoretical findings. See the anonymous link for experimental results.
> 1. We compare the stability evolution of five ASGD variants [5,6]: pure ASGD, random ASGD, pure ASGD with waiting, random ASGD with waiting, and shuffle ASGD. We train a linear model on MNIST with $m \in \\{4,8,16\\}$. As shown in Figure R1, all variants exhibit similar stability evolution, and stability decreases as $m$ increases.
> 2. We evaluate ASGD variants on heterogeneous data with stragglers. We use 10 workers and simulate stragglers by forcing 6 workers to sleep for $\\{0,50,150,250,350,450,550\\}$ ms. The buffer size is 4 for variants with waiting. As shown in Figures R2(b)–\(c), increasing sleep time raises the test error of pure ASGD and pure ASGD with waiting, while reducing the worker participation index $T/N_T^M$. In contrast, other variants remain stable. The alignment between test error and participation highlights the importance of balanced worker participation, especially in heterogeneous settings. In shuffle ASGD, jobs are assigned via random permutations, enforcing $T/N_T^M \asymp m$, i.e., $\sum_{t=1}^T \mathbb{E}[\mathbf{1}_{\{a_t=j\}}] \asymp T/m$, which explains its similar performance to random ASGD. These findings support our results in Section 7.
> 3. We provide empirical evidence for the minimax-optimal rate $O(1/\sqrt{mn})$ by showing that $\sqrt{mn} \cdot \mathbb{E}[F(A(S)) - F(\mathbf{w}^*)]$ remains nearly constant across different $m$ (Figure R2(a)). Here, we set $T \asymp mn$ as in Corollary 4.7 and use a constant step size selected via grid search.
> 4. We also evaluate stability and excess risk for nonconvex problems. Due to resource constraints, we train a three-layer fully connected network on MNIST. As shown in Figure R3, we observe similar stability and test error trends as in the convex setting, suggesting that our theoretical insights extend to nonconvex cases.

---

> > ### Author Rebuttal · Reviewer_4VXN · 2026-04-02
> >
> > Thank the authors for the response. Some of my questions have been resolved, but several issues remain insufficiently addressed in the rebuttal, e.g., the motivation of using random ASGD, the discussion of $\tau\_{max}$ is not sharp enough to distinguish the effect of each worker, and the error bound is limited to the slowest worker (straggler). I will maintain my rating.
> >
> > ---
> > Update: Thank the authors for additional explanations of ASGD. The analysis of removing $\tau\_{max}$ strengthens the results. As a result, I will raise my score from 3 to 4.

---

> > > ### Author Response · Authors · 2026-04-04
> > >
> > > We sincerely thank the reviewer for further questions.
> > >
> > > **Q1**: The motivation of using random ASGD.
> > >
> > > **A1**: Random ASGD has better convergence performance on heterogeneous data. It's pointed out in Section 5 in [9] in the anonymous link that without assumptions on data heterogeneity, no useful convergence guarantee of ASGD can be obtained under arbitrary delays. With the assumption that $\\|\nabla F_{S_i}-\nabla F_S\\|^2\le \zeta^2$ and $\\|\nabla f(w;z)-\nabla F_{S_i}(w)\\|^2\le\sigma^2,\forall z\in S_i, i\in[m]$, the convergence rate of ASGD is of $O(m/T+\sqrt{\sigma^2/T}+\zeta^2)$ [5,9], where $O(\zeta^2)$ is unavoidable [9]. In comparison, random ASGD, controlling the job assignment sequence, can improve the dependency on $\zeta^2$ to $O(\sqrt{\zeta^2/T})$, and therefore has better convergence [4]. In our manuscript, we further show that random ASGD can generalize well without such data heterogeneity assumption. We will include this motivation in the revision.
> > >
> > > **Q2:** Further discussion on $\tau_{max}$.
> > >
> > > **A2:** We can introduce a delay-adaptive step size to totally remove the dependence on the maximum delay, beyond the discussion in the original rebuttal.
> > >
> > > Recall that $\\mathcal{G}\_t$ denotes the set of ongoing jobs at time $t$. Then the number of times $(a_t,t)$ appears in $\cup_{t=1}^T \\mathcal{G}\_t$ is at most $\\tau_{t+\hat{\tau}\_t}$. Hence, the bound in Eq (D.6) can be further improved as follows
> > >
> > > $$
> > > \\sum\_{t\in[T]}\\sum\_{(a_k,k)\\in\\mathcal{G}\_t}\\mathbb{E}[\\eta\_{k+\\hat{\\tau}\_k}\^2\\|\\nabla F\_S(w_k)\\|^2]\le\\sum_{t\in[T]}\\mathbb{E}[\tau_t\eta_t\^2\\|\nabla F_S(w_{t-\tau_t})\\|^2]\leq\sum_{t\in[T]}\\mathbb{E}[\tau\_{t+\\hat{\tau}\_t}\eta\_{t+\\hat{\tau}\_t}\^2\\|\nabla F_S(w_t)\\|^2].
> > > $$
> > >
> > > Assuming that $\\max\\{\tau_t+m-1,1\\}\eta_t \leq \frac{1}{4L},$ we can improve Eq. (6.1) to
> > > $$
> > > \\mathbb{E}[\sum_{t=1}\^T \eta_t(F_S(w_t)-F(w^\*))]\lesssim\\|w_1-w^\*\\|^2+\\mathbb{E}[\sum_{t=1}\^T (\frac{\tau_t+m}{mn}+1)\eta_t\^2 F(w^\*)].
> > > $$
> > > Accordingly, if we choose the adaptive step size $\eta_t$ as in Cor. 4.7, then Eq. (D.21) improves to
> > > $$
> > > \\mathbb{E}[\frac{1}{\sum_{t=1}\^T \eta_t}\sum_{t=1}\^T \eta_t(F(w_t)-F(w^\*))]\lesssim(\frac{1+\gamma}{\gamma\sqrt{T}}+\frac{\gamma}{\sqrt{T}mn})(1+\frac{\\mathbb{E}_A[N_T^M]}{n})\\|w_1-w^\*\\|^2
> > > $$
> > > $$
> > > +(\frac{1}{\gamma}+\frac{1}{mn}+\frac{1}{\sqrt{T}})F(w^\*)
> > > +
> > > \frac{\gamma}{mn}(1+\frac{\\mathbb{E}_A[N_T^M]}{n})\\hat{F}(w^\*).
> > > $$
> > > Setting $T \asymp mn$ and $\gamma=\sqrt{mn}$ gives an excess risk bound of $O(\frac{1}{\sqrt{mn}}(1+\frac{\\mathbb{E}\_A[N_T\^M]}{n})),$ without any assumption on $\tau\_{\max}$. We will include this discussion in the revision.

---

### Official Review · Reviewer_cYvC · 2026-03-12

**Soundness:** 3
**Presentation:** 3
**Significance:** 2
**Originality:** 2
**Overall Recommendation:** 3
**Confidence:** 3

**Summary:**

The paper derives generalization/excess risk bounds for asynchronous SGD, where it relaxes some assumptions from the previous works such as Lipschitz loss or homogeneity.

**Compliance With Llm Reviewing Policy:**

Affirmed.

**Final Justification:**

I thank the authors again for their response. I maintain my borderline score (weak reject). The authors addressed some of my concerns, however my concerns regarding the insights, the impact beyond the convex case and the step-size remain valid.

**Key Questions For Authors:**

Please see the "weaknesses" section.

**Limitations:**

yes

**Strengths And Weaknesses:**

Strengths:

- The results are sound and new. Most parts of the paper are clearly presented. The asynchronous SGD algorithm is widely studied/considered in distributed learning community.


Weaknesses:

- The major issue with this paper is the significance of the results and methodology. The paper does not provide new insights (for example on the impact of delay) or improvements for the asynchronous SGD algorithm. Studying on-average stability for ASGD or relaxing assumptions such as Lipschitz or extending smoothness to the Holder assumption do not provide new insights. Morevoer, while the approach includes heterogeneity, It is unclear how heterogeneity affects the generalization performance for ASGD.

- It appears that the step-size condition in Theorem 4.1. ($\left(\left(1+\frac{m}{n}\right) \tau_t+m\right) \eta_t \leq 2 / L,$)  scales rather unfavorably with $m$ and $\tau_t$, and it's far below its possible maximum value in practice. Can the authors please justify this bound on the learning rate?

- The authors do not discuss in detail why their analysis is novel compared to previous works. The discussion in Remark 4.2. does not mention the new steps in the proof compared to previous works.

- The considered convex setup, while is suitable for analysis with algorithmic stability, does not provide insights for the analysis of real-world models such as deep neural networks.

- It is unclear how the proposed approach improves upon the previous works. Can the authors comment on the analysis and how they improve the results of previous works, and whether they believe their results can be further improved?

---

> ### Author Rebuttal · Authors · 2026-03-31
>
> Thanks for the valuable comments. Full details are available at: https://anonymous.4open.science/r/ICML_rebuttal-23152/Rebuttal.pdf.
>
> **Q1:** Insights.
> **A1:** Asynchrony and heterogeneity have negligible impact on stability and generalization (see **Rebuttal A2 to Reviewer zsj6**).
>
> **Q2:** Step-size condition.
> **A2:** A delay-adaptive rule $(a+\tau_t)\eta_t\lesssim L$ is standard for achieving convergence [4,5,6,9]. For cyclic delay [1] ($\tau_t=m-1$), $m\eta_t\lesssim L$ is mild. For example, [9] uses $\eta_t\lesssim \min\\{1/(L\tau_t),1/(Lm),1/\sqrt{T}\\}$. Our choice is practical and consistent with prior work.
>
> **Q3:** Novelty.
> **A3:** See the link for detailed comparisons. Key idea:
> 1. **Remark 4.2 (Delay error control).** For homogeneous data [4] where all workers have access to the whole training dataset $S$ and therefore the index $m_t$ is ignored, the cross term in Eq. (B.1) becomes
> $$
> -2\eta_t\langle w_{t}-w^{(i)}\_t,\nabla f(w_{t-\tau_t};S_{i_t})-\\nabla f(w\^{(i)}\_{t-\tau_t};S_{i_t})\rangle,
> $$
> which decomposes into
> $$
> -2\eta_t\langle w_{t-\tau_t}-w^{(i)}\_{t-\tau_t},\nabla f(w_{t-\tau_t};S_{i_t})-\nabla f(w^{(i)}\_{t-\tau_t};S_{i_t})\rangle,
> $$
> and
> $$
> -2\eta_t\langle w_{t}-w_{t-\tau_t}-(w^{(i)}\_t-w^{(i)}\_{t-\tau_t}),\nabla f(w_{t-\tau_t};S_{i_t})-\nabla f(w^{(i)}\_{t-\tau_t};S_{i_t})\rangle. \qquad (\*)
> $$
> The first term is bounded by
> $$
> -2\eta_t\langle w_{t-\tau_t}-w_{t-\tau_t}\^{(i)},\nabla f(w_{t-\tau_t};S_{i_t})-\nabla f(w_{t-\tau_t}\^{(i)};S_{i_t})\rangle\leq -\frac{2}{L}\\|\nabla f(w_{t-\tau_t};S_{i_t})-\nabla f(w_{t-\tau_t}^{(i)};S_{i_t})\\|^2. \qquad (\*\*)
> $$
> The key is bounding Eq.(\*), which is mainly composed of
> $$
> e_t:=-2\sum^{\tau_t}\_{l=1}\eta_t\eta_{t-l}\langle\nabla f(w_{t-l-\tau_{t-l}};S_{i_{t-l}})-\nabla f(w^{(i)}\_{t-l-\tau_{t-l}};S_{i_{t-l}}),\nabla f(w_{t-\tau_t};S_{i_t})-\nabla f(w^{(i)}\_{t-\tau_t};S_{i_t})\rangle.
> $$
> In the previous work [4], this term is bounded as follows:
>     $$
>     \begin{align\*}
>     e_t&\leq 2\sum^{\tau_t}\_{l=1}\eta_t\eta_{t-l}\\|\nabla f(w_{t-l-\tau_{t-l}};S_{i_{t-l}})-\nabla f(w^{(i)}\_{t-l-\tau_{t-l}};S_{i_{t-l}})\\|\\|\nabla f(w_{t-\tau_t};S_{i_{t-\tau_t}})-\nabla f(w^{(i)}\_{t-\tau_t};S_{i_{t-\tau_t}})\\|\\\\
>     &\leq 2L\sum^{\tau_t}\_{l=1}\eta_t\eta_{t-l}\\|w_{t-l-\tau_{t-l}}-w^{(i)}\_{t-l-\tau_{t-l}}\\|\\|w_{t-\tau_t}-w^{(i)}\_{t-\tau_t}\\|\leq Lr^2\sum^{\tau_t}\_{l=1}\eta_t\eta_{t-l},
>     \end{align\*}
>     $$
>     where the last two inequalites hold by the smoothness and bounded space assumption. Instead, we point out that this term can be eliminated together with Eq. (\*\*). Notice that
> $$
> e_t\leq\sum^{\tau_t}\_{l=1}\eta_{t-l}^2\\|\nabla f(w_{t-l-\tau_{t-l}};S_{i_{t-l}})-\nabla f(w^{(i)}\_{t-l-\tau_{t-l}};S_{i_{t-l}})\\|^2+\sum^{\tau_t}\_{l=1}\eta_t^2\\|\nabla f(w_{t-\tau_t};S_{i_t}) -\nabla f(w^{(i)}\_{t-\tau_t};S_{i_t})\\|^2. \qquad (\*\*\*)
> $$
> Summing over $T$ and applying Lemma B.1, the first term of Eq.(\*\*\*) is bounded by$(m-1)\sum^{T}\_{t=1}\eta_{t}\^2\\|\nabla f(w_{t-\tau_t};S_{i_t}) - \nabla f(w^{(i)}\_{t-\tau_t};S_{i_t})\\|^2$. Thus, if $(\tau_t+m-1)) \leq 2/L$, $e_t$ is absorbed with Eq.(**), removing the bounded-space assumption and tightening bounds.
>
> 2. **Definition 3.1 & Remark 4.6.** We introduce new definition of stability w.r.t. subsets and a new decomposition of excess risk. The motivations are:
>     1. In both data-partitioned and heterogeneous settings, the local objectives differ, i.e., $F_{S_{i}}(\cdot)\neq F_{S}(\cdot)$. Existing convergence analyses typically rely on heterogeneity assumptions, i.e. $\\|\nabla F_{S_i}(\cdot)-\nabla F_{S}(\cdot)\\|^2 \leq \zeta^2$, to control the full empirical risk. In contrast, restricting the evaluation to the optimization trajectory ($F_{S_{m_t}}(w_{t-\tau_t})$) yields tighter bounds without requiring such assumptions.
>     2. Second, stability w.r.t. subsets can be controlled by this trajectory-based empirical risk.
>
> **Q4:** Convexity restriction.
> **A4:** It is possible to extend our analysis to DNNs and nonconvex problems. The recent work shows that the loss function for DNNs satisfies a weak convexity, and the weak convexity parameter decays with the order of $O(1/\sqrt{q})$ [7], where $q$ is the width of DNNs. Based on this, [11] shows that gradient methods can achieve optimal excess risk bounds for DNNs by stability analyses. We foresee our stability and generalization analyses can also imply optimal excess risk bounds for sufficiently wide DNNs. We will add discussions in the revision.
>
> Also, we add experiments on non-convex problems (see link and **Rebuttal A3 to Reviewer 4VXN**).
>
> **Q5:** Improvements.
> **A5:** We improve the excess risk from $O((mn)^{-1/4})$ [4] to the optimal $O((mn)^{-1/2})$, which can not be further improved. We also provide the first bounds for ASGD without heterogeneity assumptions, and improved results in convex non-smooth settings (see Tables in the link and Remarks 4.3, 4.8, 5.2 and 5.5 in manuscript for details).

---

> > ### Author Rebuttal · Reviewer_cYvC · 2026-04-01
> >
> > I thank the authors for their response, especially regarding the novelty and insights from the analysis. After reading the reviews and the response, I maintain my score.

---

> > > ### Author Response · Authors · 2026-04-05
> > >
> > > Thank you once again for your thoughtful comments and valuable feedback. We summarize our previous responses and respond to your remaining concerns in detail. We will certainly incorporate all following discussions into the revised version. All the references can be founded in the anonymous link.
> > >
> > > **Q1** Concerns on insights.
> > > **A1:**
> > > 1. Asynchrony has a negligible effect on the stability and generalization behavior of SGD. Instead, worker participation plays a key role in the generalization of ASGD. Corollary 4.7 shows that, under a balanced regime, ASGD attains the minimax-optimal risk of $O(1/\sqrt{mn})$ for convex and smooth problems, demonstrating generalization comparable to vanilla SGD. In the imbalanced case, e.g., $T/N^M_T=k, k\in[1,m)$, we prove a generalization bound of $O(1/\sqrt{kn})$. As $k\to 1$, training is dominated by the fastest worker, and ASGD reduces to SGD on a dataset of size $n$, whose minimax-optimal risk is $O(1/\sqrt{n})$, showing the tightness of our analysis. Experimental results (Figure 1(a),(b)) also confirm the negligible effect of asynchrony: as the number of workers (and thus the size of training dataset and asynchrony) increases, both stability and excess risk decrease. In fact, our claim holds since the error accumulation from delayed updates is well controlled, as explained in Rebuttal A3 to Reviewer cYvC. More generally, for given hyperparameters and $N_T^M$, tuning $\gamma > 0$ yields the optimal rate by balancing optimization and stability in Theorem 4.5.
> > >
> > > 2. The heterogeneity assumption is not needed for the generalization analysis of ASGD. As shown in Theorem 4.1, stability is controlled by the optimization error along the trajectory, $\sum\_{t=1}\^T \eta\_t\^2 \\mathbb{E}[F_{S_{m_t}}(\mathbf{w}\_{t-\tau_t})]$, which can be tightly bounded without any heterogeneity assumption. In some cases, such as random ASGD, the full empirical risk can also be bounded without this assumption. Specifically, random ASGD controls the job assignment sequence $\{a_t\}$. Since $a_t$ is sampled uniformly from $[m]$, we have $\\mathbb{E}[\mathbb{1}\_{\{a_t = j\}}] = 1/m$, implying $\\mathbb{E}\_{a_t}[F_{S_{a_t}}(w_t)] = F_S(w_t)$. This is the core reason why random ASGD can generalize well on heterogeneous data even under large asynchrony.
> > >
> > > **Q2:** Impact beyond the convex setting.
> > > **A2:**
> > > We already emphasized the impact in Conclusion (Section 8). It is possible to extend our analysis to nonconvex objectives under additional conditions such as the PL condition (i.e. $\mathbb{E}[F_S(w)-F_S(w^*_s)]\leq \frac{1}{2\mu}\mathbb{E}[\\|\nabla F_S(w)\\|^2$) or weak convexity. Under the PL condition, [A1] characterizes the relationship between generalization and optimization, allowing us to derive non-vacuous excess risk bounds from optimization error bounds derived similarly to those in our analysis. Moreover, neural networks are known to satisfy weak convexity, and the weak convexity parameter decays as $1/\sqrt{q}$ [7], where $q$ is the width. Based on this observation, [11] shows that gradient methods can achieve optimal excess risk bounds for DNNs via stability analysis. Therefore, our techniques may also be used to derive optimal risk bounds for ASGD on neural networks. We also added experiments on non-convex problems, and the results align well with the insights from the convex setting. Detailed results are provided in the anonymous link and in the Rebuttal to Reviewer 4VXN.
> > >
> > > **Q3:** Choice of step size.
> > > **A3:**
> > > We believe that the step-size choice is indeed practical and aligned with existing studies. For large-scale distributed training, $m\lesssim n$ is a mild assumption [A2]. For example, ImageNet has roughly 16 million images, so typically $mn \lesssim 10^7$ and $m\lesssim 10^3$. Under this assumption, our choice implies $(\tau_t+m)\eta_t\lesssim 1/L$. As discussed in existing convergence and generalization analyses of ASGD [4,5,6,9], a delay-adaptive step size satisfying a constraint of the form $(a+\tau_t)\eta_t\lesssim 1/L$, where $a$ is a constant, is standard for mitigating the effect of stale gradients and improving convergence. In the cyclic delayed architecture [1], $\tau_t=m-1$, so the condition $m\eta_t\lesssim 1/L$ is not restrictive. More concretely, [9] uses a step size of the form $\eta_t\lesssim \min\\{1/(L\tau_t),1/(Lm),1/\sqrt{T}\\}$ and [4] require $\eta_t\lesssim 1/(\tau_t\sqrt{T})$.
> > >
> > > [A1] Lei, Yunwen, and Yiming Ying. "Sharper generalization bounds for learning with gradient-dominated objective functions." International Conference on Learning Representations. 2021.
> > >
> > > [A2] Dean, Jeffrey, et al. "Large scale distributed deep networks." Advances in neural information processing systems 25 (2012).

---

### Official Review · Reviewer_xJvX · 2026-03-13

**Soundness:** 3
**Presentation:** 3
**Significance:** 3
**Originality:** 3
**Overall Recommendation:** 4
**Confidence:** 4

**Summary:**

This paper develops a fine-grained stability-based generalization analysis for Asynchronous SGD (ASGD). The paper provides risk bounds for convex and smooth objectives under minimal assumptions, removing requirements on Lipschitz continuity, bounded noise, and bounded parameter/data domains. The analysis is extended to non-smooth objectives via Holder-continuous gradients and to heterogeneous data settings via random ASGD.

**Compliance With Llm Reviewing Policy:**

Affirmed.

**Final Justification:**

I believe the authors have addressed all my questions. I maintain my positive score.

**Key Questions For Authors:**

The majority of the concerns are outlined in the 'Weaknesses' section.

**Limitations:**

Yes

**Strengths And Weaknesses:**

Strengths:
* The paper has a significant theoretical contribution by removing several assumptions, such as Lipschitz
  continuity, bounded noise, bounded parameter/data domain that appear in most prior work on ASGD.
* The analysis accommodates arbitrary delay sequences without additional strict assumptions, and the bounds are explicitly characterized in terms of worker participation $N_T^M$, providing a useful connection between system architecture and generalization.
* Moreover, to the authors' knowledge, Theorem 6.2 provides the first non-vacuous excess risk bound for ASGD under heterogeneous
   data, which is an important open problem in federated learning theory.

Weaknesses:

* I am concerning that the step size conditions are impractical. Theorems 4.1, 4.5, and 5.1 all require conditions of the form
  $\bigl((1+\frac{m}{n})\tau_t + m\bigr)\eta_t \leq C/L$, where $\tau_t$ is the random delay at step $t$. Since $\tau_t$
  is revealed only at the time the gradient arrives, I think this condition would be hard to verify.
* The current analysis is restricted to convex objectives. ASGD and asynchronous FL is predominantly used to train deep neural networks, where non-convexity is the norm. The paper acknowledges this
  in the conclusion but does not provide detailed analysis.

---

> ### Author Rebuttal · Authors · 2026-03-31
>
> We sincerely appreciate you for your valuable comments and constructive suggestions. This anonymous link contains all the references, supplementary tables and experimental results: https://anonymous.4open.science/r/ICML_rebuttal-23152/Rebuttal.pdf.
>
> **Q1:** Whether the step size is impractical?
>
> **A1:** The choice of step size is practical. Under the assumption $m\leq n$, our choice implies that $(\tau_t+m)\eta_t\lesssim 1/L$. As discussed in most existing convergence and generalization analyses of ASGD [4,5,6,9], an delay-adaptive step size satisfying a constraint of the form $(a+\tau_t)\eta_t\lesssim L$, where $a$ is a constant, is a standard choice to achieve better convergence. In the case of cyclic delayed architecture [1], we have $\tau_t=m-1$, so the condition $m\eta_t\lesssim L$ is not restrictive. More concretely, [9] adopts a step size of the form $\eta_t\lesssim \min\\{1/(L\tau_t),1/(Lm),1/\sqrt{T}\\}$.
>
> **Q2:** The current analysis is restricted to convex objectives.
>
> **A2:** It is possible to extend our discussions to nonconvex objectives by imposing some other conditions such as the PL condition and the weak-convexity assumption. It is shown that neural networks satisfy the weak convexity and the weak convexity parameter decays as the order of $1/\sqrt{q}$ [7], where $q$ is the width. Based on this observation, the recent work [11] shows that gradient methods can achieve optimal excess risk bounds for DNNs by stability analyses. Therefore, it is possible to use our techniques to derive optimal risk bounds for ASGD on neural networks. The main contribution of this manuscript is that we remove the assumptions on Lipschitzness, data homogeneity, bounded space, bounded variance, and relax the smoothness assumption to Holder continuity assumption.
>
>  Also, we add experiments on non-convex problems. The results matches well with the insights we obtain from convex problems. The detailed experimental results can be seen in the anonymous link and the **Rebuttal A3 to Reviewer 4VXN**.

---

> > ### Author Rebuttal · Reviewer_xJvX · 2026-04-02
> >
> > Thank the authors for the response. I believe the authors have addressed all my questions. I would encourage them to explicitly incorporate these clarifications in the revised version. I maintain my positive score.

---

> > > ### Author Response · Authors · 2026-04-05
> > >
> > > We are pleased to hear that your concerns have been fully addressed. Thank you once again for your thoughtful comments and valuable insights. We will certainly incorporate all these clarifications into the revised version.

---

### Official Review · Reviewer_zsJ6 · 2026-03-13

**Soundness:** 3
**Presentation:** 2
**Significance:** 3
**Originality:** 2
**Overall Recommendation:** 4
**Confidence:** 3

**Summary:**

This paper studies the generalization properties of Asynchronous SGD (ASGD) through the lens of on-average model stability. The authors first derive stability and excess risk bounds under the assumptions of homogeneous data and smooth loss functions. They then relax the smoothness assumption by considering a Hölder-type condition. Finally, the homogeneous data assumption is relaxed, leading to an excess risk analysis for random ASGD. The paper concludes with experimental results illustrating the impact of the number of workers, the number of iterations, and worker participation on the test error.

**Compliance With Llm Reviewing Policy:**

Affirmed.

**Final Justification:**

The authors have addressed most of my concerns in their rebuttal, and I have increased my score accordingly. That said, I still believe the paper would benefit from a later resubmission after incorporating the reviewers’ comments and suggestions.

**Key Questions For Authors:**

## Questions

1. Could a fast rate of $\mathcal{O}(1/(mn))$ also be obtained in the regime where $N_t = T/m$?

2. What would you consider the main takeaway of your theoretical analysis of ASGD? In particular, does your analysis provide practical guidance for practitioners on how to use ASGD efficiently?

3. Could your analysis be adapted to recover the results of synchronous distributed SGD in the special case where the delay is zero?

4. Could you further elaborate on Remark 4.6? In particular, I did not fully understand the claim that, unlike previous work, your approach can derive excess risk bounds without evaluating the full empirical risk.

5. How do your results for heterogeneous data compare with the homogeneous case? It seems that the delay plays a more significant role in the heterogeneous setting, but this aspect is not discussed in detail.

I would be willing to increase my score if the authors address these questions satisfactorily.

**Limitations:**

Yes

**Strengths And Weaknesses:**

## Strengths

The main strength of the paper lies in its theoretical contributions, which are rather comprehensive, appear technically sound, and, most importantly, lead to non-vacuous bounds. These results clearly improve upon previous work and allow the recovery of the optimal risk rate $\mathcal{O}\left(\frac{1}{\sqrt{mn}}\right)$ in general settings, as well as a fast rate $\mathcal{O}\left(\frac{1}{n}\right)$ in low-noise regimes. The authors also make a commendable effort to compare their results with prior work throughout the paper.

## Weaknesses

- **Presentation and clarity.**
  Overall, the presentation could be improved. The paper is quite dense and sometimes difficult to follow. In particular, some discussions and remarks contain substantial theoretical material, which occasionally breaks the flow of the exposition. For example, Remark 4.6 is rather long and somewhat difficult to parse. While many of these discussions provide useful comparisons with prior work, which is a positive aspect, they are spread across the paper. A summary table comparing the main results with those in previous works would greatly improve readability.

- **Lack of a clear conceptual insight about ASGD.**
  Although the analysis appears technically sound, the paper does not clearly highlight a key conceptual insight regarding the impact of asynchrony on stability and generalization. Beyond recovering classical rates, the analysis does not provide a strong intuitive explanation of how asynchrony affects (or does not affect) stability and excess risk. If I understood correctly, the results suggest that asynchrony has a provably limited impact compared to previous analyses, which would be a strong and interesting message. However, this point is not sufficiently emphasized or explained. Without a clearer takeaway, the contribution may appear as a direct extension of Lei & Ying (2020), which could reduce the perceived originality of the work.

## Other remarks

- Lemma 4.4 could be moved earlier in the paper to better connect with Eq. (4.2).
- On line 266 (right column), Lemma 3.5 should be Lemma 3.3.
- The use of random ASGD in Section 6 is not strongly motivated, and the algorithm itself is not very clearly explained. Including a pseudocode description of the algorithm would improve clarity.

---

> ### Author Rebuttal · Authors · 2026-03-30
>
> We sincerely thank you for your constructive suggestions and valuable comments.
>
> **Q1:** Presentation and clarity.
> **A1:** We agree the presentation is dense and will clarify the remarks. We will also add summary tables for stability and excess risk bounds.  Full tables, references, and experiments are available at: https://anonymous.4open.science/r/ICML_rebuttal-23152/Rebuttal.pdf.
> Our result achieves the optimal excess risk of $O(1/\sqrt{mn})$, improving over [4], which gives $O((mn)^{-1/4})$ when $\bar{\tau}_T \asymp (mn)^{1/4}$.
>
> **Q2:** Conceptual insight about ASGD.
> **A2:** We highlight two insights:
> 1. Asynchrony has negligible impact on stability and generalization. Under a balanced regime, ASGD achieves the minimax-optimal rate for convex smooth problems. Intuitively, the error accumulation from delayed updates are well controlled (see **Rebuttal A3 to Reviewer cYvC**). The optimal rate follows by balancing optimization and stability (Theorem 4.5) via tuning $\\gamma>0$ based on hyperparameters and $N_T^M$.
> 2. Heterogeneity assumptions are not needed for generalization analysis of ASGD. By Theorem 4.1, stability is governed by the optimization error along the trajectory $\\sum_{t=1}\^T \\eta_t\^2 \\mathbb{E}[F_{S_{m_t}}(w_{t-\\tau_t})]$,  which can be tightly bounded without such assumptions. For random ASGD, we further bound full empirical risk by utilizing the relationship between $\\{m_t\\}$ and $\\{a_t\\}$  as shown in the proof of Lemma 6.1. Since $a_t\\sim \\text{Unif}([m])$, we have $\\mathbb{E}[\\mathbb{1}\_{\\{a_t=j\\}}]=1/m$, which implies $\\mathbb{E}\_{a_t}[F_{S_{a_t}}(w_t)]=F_S(w_t)$.
>
> **Q3:** Other remarks.
> **A3:** We will improve our presentation by fixing the typos and describing pseudocode of random ASGD.
>
> **Q4:** Can fast rate $\\mathcal{O}(1/(mn))$ be obtained?
> **A4:** Yes. Under the setting of Cor. 4.7, set $T\\asymp m^2n$, $N_T^M\\asymp mn$, and choose $\\eta_t$ that satisfies $\\sum_{t=1}^T \\eta_t\\ge T/(24mL)$ (shown in Line 1235). Then Eq. (4.6) yields a fast rate $O(1/mn)$ under low noise.
>
> **Q5:** Does the analysis recover synchronous SGD when delay is 0?
> **A5:** Since we do not consider gradient aggregation in Algorithm 1, when the delay is zero, the algorithm reduces to standard SGD rather than synchronous SGD (mini-batch SGD). As specified in Remark 4.3, our analysis recovers the results of standard SGD. Extending to aggregation or semi-synchronous methods (e.g., FedBUFF) is an interesting question for future work.
>
> **Q6:** Explanation of Remark 4.6.
> **A6:** Prior work [5,6,9] requires heterogeneity assumptions (e.g., $\\|\\nabla F_{S_i}-\\nabla F_S\\|^2\\le \\zeta^2$) to evaluate the full empirical risk since in both data-partitioned and heterogeneous settings, the local objectives differ, i.e., $F_{S_{i}}(\cdot)\neq F_{S_{j}}(\cdot)$ for $i\neq j$. We instead use  following new decomposition:
> $$
> \\mathbb{E}[F(w_{t-\\tau_t})-F(w^\*)]
> =\\mathbb{E}[F(w_{t-\\tau_t})-F\_{S_{m_t}}(w_{t-\\tau_t})]+\\mathbb{E}[F\_{S_{m_t}}(w_{t-\\tau_t})-F(w^\*)],
> $$
> where the emprical risk along the optimization trajectory, $F_{S_{m_t}}(w_{t-\tau_t})$, can be tighter bounded since the stochastic gradient of ASGD satisfies $\mathbb{E}\_{i_t}[\nabla f(w_{t-\tau_t};z_{i_t,m_t})] = \nabla F_{S_{m_t}}(w_{t-\tau_t})$.
> By convexity of $f$,
> $$
> \begin{align\*}
> \mathbb{E}[F(A(S))-F(w^*)]&\le \frac{1}{\\sum_{t=1}^T\eta_t}\sum_{t=1}^T\eta_t\mathbb{E}[F(w\_{t-\tau_t})-F(w^\*)]\\\\
> &\\le \\frac{1}{\\sum_{t=1}^T\\eta_t}(\\sum_{t=1}^T\\eta_t\\mathbb{E}[F(w\_{t-\tau_t})-F\_{S_{m_t}}(w_{t-\\tau_t})]+ \\sum_{t=1}^T\eta_t\\mathbb{E}[F\_{S_{m_t}}(w\_{t-\\tau_t})-F(w^\*)]),
> \end{align\*}
> $$
> where the first term can be bounded using Lemma 3.3 and the second term can be controlled by Lemma 4.4.
>
> **Q7:** How do your results for heterogeneous data compare with the homogeneous case?
> **A7:** Our stability analysis (Thm. 4.1) holds under heterogeneous data and does not impose any assumption on delays. This suggests that the main gap between heterogeneous and homogeneous settings arises in the convergence analysis rather than in stability and generalization. The $\\tau_{\\max}$-dependence comes from perturbed iterate technique and the virtual sequence (Lemma 6.1). As in [5,6], it can be removed with additional bounded gradients and heterogeneity assumptions. Alternatively, we can introduce following decomposition:
> $$
> F(w_{t-\\tau_t})-F(w^\*)=(F(w_{t-\\tau_t})-F_{S_{m_t}}(w_{t-\\tau_t}))+(F_{S_{m_t}}(w_{t-\\tau_t})-F_{S_{m_t}}(w^\*))+(F_{S_{m_t}}(w^\*)-F(w^\*)).
> $$
> Then we can combine stability (Thm. 4.1) and optimization bound shown in Eq. (D.18) to derive an excess risk bound without dependency on $\tau_{max}$. When $F_{S_{m_t}}(w^\*)-F(w^\*)$ is tightly bounded, or in low-noise regimes, we can also get a non-vacous excess risk bound. In this work, we take a step toward weakening the heterogeneity assumption. Reducing $\\tau_{\\max}$ dependence without additional assumptions remains open.

---

> > ### Author Rebuttal · Reviewer_zsJ6 · 2026-04-03
> >
> > I thank the authors for their response. They have addressed most of my concerns, and I have increased my score accordingly. That said, I still believe the paper would benefit from a resubmission after more thoroughly incorporating the reviewers’ comments.

---

> > > ### Author Response · Authors · 2026-04-05
> > >
> > > We are pleased to hear that most of your concerns have been addressed. Thank you once again for your thoughtful comments and valuable insights. We will carefully incorporate all of the reviewers' suggestions in the revision.

---

### Decision · Program_Chairs · 2026-04-30

**Decision:**

Accept (regular)

**Comment:**

This paper investigates the generalization guarantees for asynchronous SGD, removing the assumptions on Lipschitz continuity, bounded noise, and bounded parameter or data domains. The reviewers agreed that this paper appears to be technically sound. However, they also pointed out that the technical contributions against the existing works are unclear, and that the analysis does not provide a strong intuitive explanation of how asynchrony affects stability. Besides, the impact of data heterogeneity is not adequately investigated in the analysis. For these reasons, although this paper has its merits, I have to recommend reject.